# Typhoon Haiyan's sedimentary record in coastal environments of the Philippines and its palaeotempestological implications

Dominik Brill[1*], Simon Matthias May[1], Max Engel[1], Michelle Reyes[2], Anna Pint[1], Stephan Opitz[1], Manuel Dierick[3], Lia Anne Gonzalo[4], Sascha Esser[1], Helmut Brückner[1]

[1] Institute of Geography, Universität zu Köln, Germany
[2] Marine Science Institute, University of the Philippines, Philippines
[3] Department of Physics and Astronomy, Universiteit Gent, Belgium
[4] Nationwide Operational Assessment of Hazards (Project NOAH), Department of Science and Technology, Philippines

*Correspondence to:* Dominik Brill, Albertus-Magnus-Platz, 50923 Köln, Germany, brilld@uni-koeln.de

**Abstract:** On November 8th 2013, category 5 Supertyphoon Haiyan made landfall on the Philippines. During a post-typhoon survey in February 2014, Haiyan-related sand deposition and morphological changes were documented at four severely affected sites with different exposure to the typhoon track, and different geological and geomorphological settings. Onshore sand sheets reaching 100-250 m inland are restricted to coastal areas with significant inundation due to amplification of surge levels in embayments or due to accompanying long-wave phenomena at the most exposed coastlines of Leyte and Samar. However, localized washover fans with a storm-typical laminated stratigraphy occurred even along coasts with limited inundation due to waves overtopping or breaching coastal barriers. On a recent reef platform off Negros in the Visayan Sea, storm waves entrained coral rubble from the reef slope and formed a several 100 m long, intertidal coral ridge when breaking at the reef edge. As these sediments and landforms were generated by one of the strongest storms ever recorded, they not only provide a recent reference for typhoon signatures that can be used for palaeotempestological and palaeotsunami studies in the region, but might also increase the general spectrum of possible cyclone deposits. Although a rather atypical example for storm deposition due to the influence of infra-gravity waves, it nevertheless provides a valuable reference for an extreme case that should be considered when discriminating between storm and tsunami deposits in general. Even for sites with low topography and high inundation levels during Supertyphoon Haiyan, the landward extent of the documented sand sheets seems significantly smaller than typical sand sheets of large tsunamis. This criterion may potentially be used to distinguish both types of events.

**Keywords:** typhoon; tropical cyclone; storm deposit; storm versus tsunami; sand sheet; washover fan; coral ridge; Philippines

## 1. Introduction

On November 8th 2013, Typhoon Haiyan (local name: Yolanda) made landfall on the Philippines reaching category 5 on the Saffir-Simpson Hurricane scale. By crossing the archipelago Haiyan caused more than 6000 casualties, affected more than 16

million people, and damaged more than 1 million houses (NDRRMC, 2014; Lagmay et al., 2015). Its destructive power resulted from exceptional surface winds reaching sustained velocities of up to 315 km/h (1-minute averaged data), gusting even up to 380 km/h, in combination with massive storm surge flooding with water levels up to 9 m above tide level (IRIDeS, 2014). Based on recorded wind speeds and core pressure, Haiyan was not only an exceptional event for the Philippines but one of the

most powerful tropical cyclones ever recorded. Against the background of the ongoing controversial discussion on the influence of climate change on cyclone frequencies and magnitudes (Knutson et al., 2010; Pun et al., 2013), Typhoon Haiyan could be both an exceptional low-frequency event and/or a precursor of a new normality.

The need for robust data that provide information about long-term typhoon risk in affected areas is highlighted by the lack of awareness and preparedness of many inhabitants regarding Haiyan (Engel et al., 2014; Lagmay et al., 2015). Similar to other

recent coastal disasters, such as the 2004 Indian Ocean Tsunami (Brückner and Brill, 2009) or Cyclone Nargis in 2008 (Fritz et al., 2009), studies of occurrence patterns and effects of past flooding events with exceptional magnitude have been lacking in the Central Philippines. Although previous catastrophic typhoons with similar tracks have been historically documented, e.g. in 1897 or 1984 (PAGASA, 2014; Soria et al., 2016), their disastrous effects have not been taken into account properly. This was at least partly due to limited comprehension of the term "storm surge", which people mainly associated with the

moderate flooding of typical typhoons in the area (Engel et al., 2014; Mas et al., 2015). This discrepancy between real risk and perceived risk results from an underrepresentation of local high category typhoons in instrumental and historical records. This discrepancy is great, because cyclones usually follow an inverse power law (Corral et al., 2010).

Geological imprints of cyclones may be used to enhance existing historical and instrumental records, as they potentially cover periods of several millennia and, thus, document even large events in statistically significant numbers (Hippensteel et al., 2013;

May et al., 2013, 2015a). However, using geological archives to extend tropical cyclone histories requires the identification of tropical cyclone traces, reliable dating of event layers, as well as a careful consideration of potential changes in paleogeography and/or sea level. In particular the discrimination of similar depositional events such as tsunamis is challenging (Shanmugam, 2012). In this regard, modern event deposits offer the possibility to establish (locally valid) event-specific criteria that indicate prehistoric events in the geological record (Nanayama et al., 2000; Kortekaas and Dawson, 2007). Although all potential

discrimination criteria have been observed for both tsunami and storm deposits on a global perspective (Shanmugam, 2012), local comparisons assign several decimetre thick, laminated deposits with inland extents of a few tens to hundreds of meters that often show foreset bedding and tend to thin and fine landwards to typical storm signatures, while tsunami deposits tend to be thinner, are composed of a few layers with massive or normally graded structure, and may extend inland for several kilometres (Tuttle, 2004; Morton et al., 2007; Switzer and Jones, 2008; Goto et al., 2011). Likewise, modern analogues may

help to evaluate reliable age (Bishop et al., 2005; Brill et al., 2012) and magnitude (Brill et al., 2014) reconstructions of palaeoevents. Since sedimentary signatures of cyclones are influenced by local factors (Schwartz, 1982; Matias et al., 2008), Haiyan's geological footprint will primarily be useful to establish criteria for inferring past typhoons from geological records found in the same areas. However, as one of the strongest storms ever recorded, Haiyan might increase the spectrum of possible cyclone features and, by this, add to the discussion on discriminating between tsunamis and extreme cyclones in general.

Here, we report on sandy sediment data from four different Haiyan-affected coastal areas, collected during a post-typhoon field survey on the Philippines in February 2014, i.e. three months after Haiyan made landfall. The investigated areas comprise different geologies with carbonate and volcanoclastic coasts, as well as different geomorphological settings including steep cliff platforms, low coastal plains and coral reefs. The major aims of this study are to (i) document Haiyan's impact on the sedimentology and geomorphology of heavily affected coastal areas by recording onshore and intertidal sedimentation, coastal erosion and geomorphological changes. Based on these data, sedimentary and geomorphological typhoon signatures typical for the study area shall be established. In addition, (ii) the spatial variability of these typhoon signatures due to site-specific characteristics such as the local topography, bathymetry, geology, and hydrodynamics as well as the exposure to the typhoon track shall be investigated. Finally, (iii) the potential of these modern typhoon deposits will be evaluated in respect of possible implications for the identification of prehistoric tropical cyclones in the geological record.

## 2. Study area

The Philippine archipelago is located in the western Pacific Ocean between 5° and 20° northern latitude. Field work was carried out in four different areas (Fig. 1): (i) at the east coast of Samar between Llorente and General MacArthur (Fig. 1b); (ii) at the northeastern coast of Leyte between Tacloban and Dulag (Fig. 1b); (iii) on several islands northeast of Northern Negros (Sagay); and (iv) on Bantayan (Fig. 1c). All four study areas were significantly affected by Typhoon Haiyan, but are characterized by a different exposure to the typhoon track and by particular geomorphological and geological settings.

### 2.1 Climate and typhoon activity

### 2.1.1 General climatic and hydrodynamic conditions

The Philippine islands are characterized by a subtropical climate influenced by monsoon winds and the ENSO system. While the west of the archipelago experiences seasonal variations in rainfall and temperature with a dry period from November to April and a wet season during summer monsoon, seasonal variability gradually decreases towards the east reaching all-season wet conditions along the eastern coast (PAGASA, 2011). Hence, the study areas on Leyte and Samar show no pronounced annual rainfall variations. Northern Negros and Bantayan are influenced by weak monsoon seasonality. The highest swell waves occur during the winter monsoon and along the Pacific coast, while wave action during the summer monsoon and within the Sea of Visayas is generally moderate (Fig. 1e, Navy METOC, 2014). Tidal variations are in the range of 0.8–1.8 m. In addition, the entire Philippine territory is located in the corridor of east-west moving typhoons. On average, 21 tropical cyclones hit the Philippines annually, whereas the occurrence probability increases markedly towards the north (Fig. 1d, PAGASA, 2014).

## 2.1.2 Haiyan's path over the Philippines

Starting as a tropical depression on November 3$^{rd}$ 2013 over the northwestern Pacific, Haiyan continuously gained intensity, turning into a tropical storm on November 4$^{th}$, and a typhoon on November 5$^{th}$. On November 6$^{th}$ Haiyan reached category 5 on the Saffir-Simpson Hurricane scale and finally made landfall on Samar on November 8$^{th}$ (IRIDeS, 2014) (Fig. 1a). After its first landfall near Guiuan (Eastern Samar) at 4:40 am, Haiyan crossed the archipelago in a western direction without reducing its strength below category 5 (Fig. 1a). On its path over the Philippines, Haiyan made landfall on northern Leyte at 7:00 am, on northern Cebu at 10:00 am, on Bantayan at 10:40 am, and later on Panay and Palawan (NDRRMC, 2014).

Depending on the exposure to the typhoon track, wind speeds, storm surge levels and wave heights varied significantly between the study sites. Additional variations are determined by differences in local bathymetry, fetch and shape of the coastline (see section 2.2). In general, coastal flooding rapidly reached peak levels that lasted for approximately two hours and was characterized by inflowing waves with periods of several seconds (Mas et al., 2015; Morgerman, 2014). The resulting flooding levels at the affected coastlines could mostly be reconstructed by means of combined storm surge and phase-averaged wave models (Bricker et al., 2014; Mori et al., 2014; Cuadra et al., 2014) (details concerning the specifications of each model presented in this paper are provided in the respective references). However, around Tacloban, significant amplification of storm surge levels to values of up to 8 m were measured (Mas et al., 2015), and three distinct flooding pulses were observed by eyewitnesses (May et al., 2015). Furthermore, along the coast of Eastern Samar surprisingly high values of run-up and flow depth were documented during post-typhoon surveys (Tajima et al., 2014; May et al., 2015b). While the exceptional water levels in the semi-enclosed basin of the San Pedro Bay can be explained by the funnel-shaped topography and seiches using combined storm surge and phase-averaged wave models (Mori et al., 2014; Bricker et al., 2014; Soria et al., 2016), the inundation pattern along the open coast of Eastern Samar can be attributed to the impact of infra-gravity waves due to an interaction of wind waves with the coral reef, if phase-resolving wave models are applied (Roeber and Bricker, 2015; Kennedy et al., 2016).

The exposed coast of Eastern Samar is characterized by a large fetch and a steep offshore bathymetry. Hence, it experienced maximum wind speeds with the highest model-predicted storm waves of up to 20 m, but only a limited wind-driven surge (Bricker et al., 2014). Field evidence documents run-up of approximately 12 m above mean sea level, and up to 800 m inundation distance (PAGASA, 2014; Tajima et al., 2014; May et al., 2015b). Also the northeastern coast of Leyte was still exposed to the full strength of the storm winds, since the landmasses to the east are too narrow to significantly reduce Haiyan's intensity. However, due to the shallow water and the resonance effects in the enclosed embayment (Mori et al., 2014), storm-water levels were dominated by a surge setup, while model-predicted wave heights were <5 m (Bricker et al., 2014). Field evidence shows that the storm surge reached run-up levels of nearly 8 m a.s.l., and inundation of several hundred meters inland (Mas et al., 2015, Tajima et al., 2014). On the other hand, smaller surge levels and moderate wave heights were modelled for Northern Negros and Bantayan (Bricker et al., 2014), due to a shallow bathymetry, low tide at the time of landfall and the

sheltering landmasses of Samar, Leyte and Cebu to the east. Likewise, eyewitness accounts document maximum onshore flooding levels of only ~2 m a.s.l. (Cuadra et al., 2014).

## 2.2 Geology and geomorphology

The Philippine islands are formed by a complex geological structure of north-south running volcanic arcs, tectonic basins and fragments of continental crust reflecting subduction and collision processes between the Eurasian and the Philippine plates (Rangin et al., 1989). As the main tectonic structures, the Philippine and Manila trenches confine offshore subduction zones to the east and west of the archipelago, while the Philippine Fault crosses the archipelago in an axial position from north to south (Fig. 1a, Rangin et al., 1989). The associated volcanoes and ophiolites form islands dominated by steep, cliff-lined coasts that are bounded by fringing reefs and are occasionally intersected by flat alluvial lowlands and pocket beaches. The clastic beaches are characterized by either white sand of coral reef origin, or darker and denser minerals derived from volcanic provinces (Bird, 2010).

The rocky carbonate coast of Eastern Samar is characterized by steep headlands with cliffs formed of Pleistocene coral limestone and occasional pocket beaches bordered by fringing reefs (HER, Fig. 1b). Since the fetch over the Pacific is not restricted, Eastern Samar is exposed to high swell waves especially during the winter monsoon (Fig. 1e) (Bird, 2010). On the other hand, the northeastern coast of Leyte is dominated by alluvial plains, sandy beaches and beach-ridge plains (TOL, Fig. 1b, Dimalanta et al., 2006), while rocky promontories and fringing coral reefs are scarce. Water depths in the shallow, funnel-shaped San Pedro Bay between Leyte and Samar do not exceed 30 m (Bird, 2010). In contrast to this, the northeastern coast of Negros is characterized by wide coastal plains bordered by mangroves and extensive coral reefs (Rangin et al., 1989). Within the shallow water of the Visayan Sea to the north (water depths mainly <50 m) numerous coral islands occur (CAR and MOL, Fig. 1c). The island of Bantayan represents the northeastern limit of the coral islands. The raised limestone formations making up the island are completely surrounded by fringing reefs that partly border steep rocky coastlines, but are also associated with sandy beaches (BAN A and B in Fig. 1c, Bird, 2010).

## 3. Methods

The topography of all studied locations was documented along transects by means of a Topcon HiPer Pro differential global positioning system (DGPS). To reconstruct flooding characteristics of Haiyan's storm surge, indicators for onshore flow depth and run-up height were documented by measuring elevations of debris lines, grass or floated debris in trees and bushes, as well as impact marks in the bark of palm trees relative to the sea level. All altitudes are given in meters above mean sea level (msl), and – in case of flood marks – additionally in meters above ground surface. Onshore flow directions were deduced from oriented grass and trunks using a compass. In addition, comparison of rectified pre- and post-Haiyan satellite images was used to estimate inundation areas. Sediments and depositional landforms generated by Typhoon Haiyan were documented both onshore and in the intertidal zone. This included sandy deposits as well as coral-rubble ridges. Shore-perpendicular trenches

were used to describe and document the stratigraphy of sandy deposits in the field. For detailed sedimentary and faunal analyses, sediment from Haiyan's deposits and reference environments was taken in the form of bulk samples from each stratigraphical unit at different distances from the shoreline, as well as push cores from at least one storm deposit with representative sedimentary structure at each investigated location.

To deduce transport processes, mode of deposition, and source environments of the storm deposits, we analysed fine-grained samples in terms of their granulometry, geochemistry, mineralogy and faunal composition at the Institute of Geography, University of Cologne. Grain-size analyses were performed with a Laser Particle Analyser (Beckman Coulter LS 13320) for material <2 mm after pre-treatment with $H_2O_2$ and $Na_4P_2O_7$ to remove organic carbon and to avoid aggregation. In case of samples with grain size >2 mm, the granulometry was determined on dried sediment using a Camsizer (Retsch Technology).

It should be noted that both approaches do not consider differences in particle shape and density, which may influence the settling velocity of the grains significantly (Woodruff et al., 2008). This is particularly important for the interpretation of granulometric variations in storm deposits with a significant percentage of shells. Statistical parameters (mean, sorting, skewness) of grain-size distributions were calculated with the GRADISTAT software (Blott and Pye, 2001) using the formulas of Folk and Ward (1957). Further information about flow directions and mode of deposition was obtained by visually

interpreting sedimentary structures in µCT scans with 50 µm voxel resolution (using myVGL 2.1 software) performed on two selected push cores (BAN 4 and TOL 8) at the University of Ghent, Belgium.

The chemical characterization of deposits includes determination of organic carbon by means of loss on ignition (LOI) measured after oven-drying at 105 °C for 12 h and ignition in a muffle furnace at 450 °C for 4 h. Carbonate contents were measured gas-volumetrically using the Scheibler method. The bulk-mineralogical composition was determined by X-ray

diffractometry (XRD) on powder compounds performed on a Siemens D5000 using a step interval of 0.05° and a dwell time of 4 seconds. A fixed 1° divergence and antiscatter slit was used at diffraction angles from 5 to 75° 2theta. The Cu K-alpha radiation source was operated at 40 keV and 40 mA. The data were analysed with DiffracPlus Eva software package (Bruker AXS, Berlin, Germany).

Magnetic susceptibility (MS) was measured with a Geotec MSCL core sampler. To apply microfaunal composition as an

25 indicator for sediment source and mode of transportation (e.g., Pilarczyk et al., 2016; Quintela et al., 2016), samples were sieved to isolate fractions >100 and <100 µm. At least 100 foraminifers were identified to species level, if possible, and counted under a binocular microscope. States of reworking were assessed semi-quantitatively on the basis of test taphonomy and classified as no, low, medium, or strong reworking. To select appropriate parameters for interpreting transport processes and sediment source areas, we carried out principal component analysis (PCA) using PAST software (Hammer et al., 2001) after

30 removing foraminifer species with insignificant abundance (i.e. <5 individuals in all samples) and correlating parameters based on Spearman's rank correlation coefficient (≥0.95).

## 4. Results – Sedimentary and geomorphological structures

### 4.1 Eastern Samar – Hernani (HER)

At Barangay Batang, municipality of Hernani (HER, Fig. 1b) the Pleistocene reef retreats to a position more than 200 m from the shoreline, resulting in a gently inclined sandy coast (Fig. 2). For Haiyan, local residents report complete inundation of the lower coastal plain with water levels of several meters above msl, while Agaton and Basyang – the two tropical storms/depressions hitting the Philippines on January 19th and February 1st, respectively (Fig. S3), which is after Haiyan and before this field survey – caused no significant flooding (Tab. S1 in online supplement).

Starting at the shoreline behind a 550 m wide intertidal reef lagoon and destroyed mangrove stands, the landward transect crosses a sandy beach ridge with a crest height of 2.2 m above msl at 50 m from the shoreline. Landwards, the tree-covered back-barrier depression is crossed by a shore-parallel channel before shallow reef outcrops occur at 120 m from the shoreline. After a second depression cultivated with rice fields (160–220 m), the transect ends at the inactive cliff of the elevated Pleistocene reef platform at 230 m inland and 2.8 m above msl. While swash lines document maximum inundation that exceeded the inactive cliff, floated grass and litter in trees indicate water levels of at least 2.5 m above msl (equal to a flow depth of 1.5 m above surface) at 100 m, up to 5.0 m above msl (3.7 m flow depth) at 170 m, and heights of 2.6 m above msl (1.4 m flow depth) at 220 m (Fig. 2).

Erosion was the dominant process seaward of the beach ridge, which is marked by an erosive scarp. Behind the barrier, sedimentation of sand and coral rubble was documented (HER 3–10), reaching its landward limit at the foot of the inactive cliff (Fig. 2). A sharp contact separates the light brown sandy layer covering the dark brown, densely rooted pre-Haiyan soil (Figs 4b, c). The thickness of the deposit decreases rapidly landwards, from 20 cm directly behind the beach ridge (HER 9) to only 2 cm at 110 m (HER 7) and a few mm at 210 m (HER 3). Coarse coral clasts at the surface (up to ~20 cm) occur as far as 100 m inland. While mean grain size does not show any fining trend in the deposit, modal grain size decreases from 1.3 cm to 220 µm along the same section (Figs 3, 4a).

Two different sedimentary facies units can be distinguished: Unit 1 was not encountered in HER 9 but comprises the entire typhoon deposit in HER 3–8 and 10 (Figs. 4a, b). It is characterized by normally graded to massive (HER 3-7) or slightly laminated (HER 8, 10), poorly sorted (3.4-7.3), unimodal fine-medium sand (mean of 85-230 µm). While sorting and mean grain size remain more or less constant, unit 1 is thinning landward from 8 cm at 90 m from the shoreline (HER 8) to only 3 mm at 210 m (HER 3). At the same time, the modal grain size decreases from 570 µm to 220 µm (Fig. 3). Unit 2 is characterized by a bimodal grain-size distribution that is significantly coarser than unit 1 (modes at 1.2 mm and 2.7 cm). It is only present at HER 9 (Fig. 4c), where it forms a 15 cm thick layer with steeply landward inclined layering and numerous angular coral fragments (constituting 77% of total mass). Both units are dominated by carbonates (nearly 100% Mg-calcite and aragonite). The foraminifer assemblage (determined for HER 10) is dominated by *Calcarina* sp. (41%), *Amphistegina* sp. (21%) and *Baculogypsina sphaerulata* (12%) that show significant reworking (<8% fresh tests, >50% strongly reworked). Vertical

variations of species composition and taphonomy within HER 10 follow no clear trend. They rather roughly correlate with changes in grain size, showing a higher percentage of abraded and broken tests for coarser sediment sections (Fig. 4d).

## 4.2 Northeast Leyte (Tolosa)

Flood levels in Tacloban and in the coastal barangays to the south (Fig. 1b) reached >6 m above msl and locally inundation extended up to 1 km inland (Tajima et al. 2014). The typhoon was accompanied by strong wind damage with uprooted trees, heavy beach erosion and onshore sedimentation. Due to the position to the typhoon centre – winds and surge are strongest to the right of the typhoon track – water levels, beach erosion and sedimentation continuously decreased towards the south.

Although – according to surge levels – thickest onshore deposits are to be expected in Tacloban and directly south of it, we report on the storm impact on the nearshore area of Tolosa's beach-ridge plain (TOL, Fig. 1b), since the northern areas are more densely populated and unaltered typhoon sediments were hard to find three months after Haiyan. Floated grass found in trees and bushes are evidence that Haiyan overtopped the 2.2 m high beach ridge (above msl) and inundated the marshy back-barrier depression with water levels of at least 4.7 m above msl (3.5 m flow depth) at a distance of 70-90 m from the shoreline (Fig. 5). At the landward end of the transect, 300 m from the shoreline, water levels of at least 3.2 m above msl (1.8 m flow depth) have been recorded and satellite images for the Tolosa area document a landward flooding extent of up to 800 m during Haiyan (Fig. 5a). However, since sandy deposition stopped even more seaward, any further documentation was beyond the scope of this survey. The direction of onshore flooding is indicated by grass bent down in a NW-NNW direction, a westward collapsed north-south running wall structure, and the NW orientation of floated palm trunks (Figs 5b, c).

An erosive scarp at the seaward slope of the beach ridge indicates coastal erosion by Typhoon Haiyan that had already partially recovered in February 2014. Pre- and post-typhoon satellite images show a shoreline retreat of 25–30 m (inset in Fig. 5b). Between 30 and 130 m inland, a sheet of dark grey sand was deposited (TOL 3–14, Fig. 6). While 5–10 cm of sediment were accumulated on top of the beach ridge (TOL 3–4), the maximum thickness of 10–20 cm is reached directly leeward of the barrier (TOL 5–8). Further landward, the thickness rapidly declines, reaching 2–5 cm at 70–90 m from the shoreline (TOL 9–11) and only a few millimetres landwards of TOL 11 (TOL 12–14) (Fig. 6).

Investigations of trenches TOL 5 and 7 reveal the complete absence of carbonate components and microfauna in the storm layer. Instead, the composition is dominated by a mixture of feldspar (30–70%), amphibole (20–40%) and a minor percentage of quartz (5–30%). Magnetic susceptibility and LOI help to discriminate typhoon deposits from the underlying soil, but have constant values within the storm layer (Fig. 6c). However, in the stratigraphy of TOL 7 (Fig. 6b) and in the µCT scan of TOL 8 (Fig. 7), a normally graded unit 1 with scour marks at the base is clearly discriminated from the successive horizontally laminated unit 2 within the Haiyan deposits.

Unit 1 forms a slightly normally graded to massive layer of unimodal fine to medium sand at the base of trenches TOL 7–14. It covers the pre-Haiyan soil and, at several places, bended stems of grass. In trenches TOL 9–14, unit 1 constitutes the entire

event deposit and is covered by a thin mud cap (Fig. 6a). While it is well-sorted (1.5–1.7) throughout the entire transect, its thickness decreases in a landward direction from 3–5 cm behind the beach ridge (TOL 7–10) to <1 cm at TOL 11–14. A landward fining trend from 240 µm to 130 µm has been noted as well (Fig. 3). Unit 2 comprises a well-laminated layer of unimodal medium sand (Figs 6, 7) and makes up the upper part of the storm layer in the proximal part of the transect (TOL 3–8). Sorting is similar to unit 1 (1.5–1.6), but the mean grain size (245–328 µm) is slightly coarser. The lamination is due to alternating concentrations of pyroxene (dark) and quartz (light) (XRD results, Fig. 8a). In TOL 5, the laminae are slightly inclined landwards (10-15°) and show repeated coarsening and fining sequences that superimpose a vertical coarsening upwards trend from ~260 µm to 320 µm (Fig. 6c). Unit 2 forms a washover fan at the landward slope of the beach ridge. While no clear trend in the thickness of the deposit was found – values range between 5 cm at the top of the beach ridge (TOL 4) and the landward edge of unit 2 (TOL 8), and ~20 cm directly behind the ridge (TOL 5) – it slightly thins landward (Fig. 3).

## 4.3 Northern Negros

All visited coral islands north of Negros (Carbin, Molocaboc, Suyac, Fig. 1c) were affected by moderate flooding reaching a few tens of meters inland and water levels less than 2-3 m above msl. Wind and wave directions changed from NNE before to WSW after the passage of Typhoon Haiyan's centre (Tab. S1, online supplement). Substantial beach erosion associated with onshore transport of sediment was observed directly after Haiyan. While coral rubble ridges were formed in the intertidal zone of Suyac and Carbin (CAR), onshore deposition was restricted to thin sand patches (few centimetre) and small coral boulders or parts of sea walls (main axis <2 m) in the proximal coastal zones of Molocaboc (MOL) and Suyac.

### 4.3.1 Carbin Reef (CAR)

On Carbin Reef, a coral rubble ridge along the western edge of the reef platform is visible at low tide (Fig. 9), but entirely submerged at high water. According to local fishermen (personal communication), the ridge did not exist in its present form prior to Haiyan (Tab. S1, online supplement), which means that it had either been absent or significantly lower (Reyes et al., 2015). The average crest height of the more than 300 m long section of the ridge measured during the field survey is 0.20 m below msl (boulders reach heights of 0.45 m above msl) without a significant trend in crest elevation. The basal width is 10–20 m, and lobe structures at its landward side cover corals in living position that must have been alive prior to the typhoon. Morphology and sedimentary structure – the ridge is mainly composed of centimetre to decimetre large coral fragments – allow the discrimination of two units: lobes of greyish, algae-covered coral rubble extending up to 20 m landward (unit 1), and light, freshly broken coral branches that form steep lobes or patchy coverings extending not more than 10 m from the reef edge, on top (unit 2). While both units are present along the entire 300 m long section, they significantly broaden from 10 m to 20 m width along the northern section (Fig. 9).

### 4.3.2 Molocaboc Island (MOL)

At Molocaboc (Fig. 10), the coastal zone behind the 550 m wide intertidal platform of the fringing reef is formed by a beach ridge with a crest height of 2.4 m above msl, followed by a shallow back-barrier depression densely covered by acacia shrubs at 2.0 m above msl. Behind the beach ridge, fresh sandy deposits that rapidly thin landwards from 10 cm at 20 m shoreline distance to only 1 cm at a distance of 45 m have been recorded. The unimodal, medium to coarse (mean = 600–680 µm), moderately sorted (1.7–1.8) sand shows no apparent sedimentary structures and a calcareous composition (>90% Mg-calcite and aragonite); the foraminifer assemblage is dominated by *Calcarina* sp. (58%) and strongly reworked tests (<18% fresh). The storm deposit contrasts with the underlying soil which is likewise unimodal, but slightly finer (mean of 390-410 µm), poorly sorted (3.4–4.7) and shows elevated contents of organic matter (4–7%) and reduced carbonate concentrations (70–80%). A reference sample from the shallow subtidal ($R_{subt}$) at 0.5 m below msl is unimodal, slightly finer (mean = 310 µm), moderately sorted (2.4), and dominated by different foraminifers (*Quinqueloculina* spp. [18%], *Peneroplis pertusas* [17%], *Elphidium* sp. [14%], *Ammonia beccarii* [12%]) with moderate reworking (74% fresh).

### 4.4 Bantayan

Eyewitnesses report limited storm surge elevations with moderate waves and peak flooding arriving at low tide for the entire east coast of Bantayan (Tab. S1, online supplement). However, while onshore deposition is restricted to sand sheets of a few centimetres in coastal areas directly behind active beach ridges, beach erosion of several meters did occur. More pronounced flooding happened only locally at the mouths of estuaries and resulted in the formation of small washover fans (BAN A and B, Fig. 1c).

### 4.4.1 Bantayan A (BAN A)

At BAN A, the NE-exposed beach section (Figs 11a, b) is separated from an estuary river mouth by outcrops of the Pleistocene coral reef limestone with 1–2 m high cliffs. Cross sections reveal a succession of shore-parallel beach ridges with elevations of 1.5-1.8 m above msl that are interrupted by ~1 m deep swales (Fig. 12a). While recent flooding levels were indicated by debris lines at 1.4 m above msl along the seaward and landward slopes of the second beach ridge, as well as on top of the 2.9 m high (above msl) reef platform to the NW, the maximum wave height during Haiyan was estimated to have reached 3.0–3.4 m above msl on the basis of eyewitness accounts (Tab. S1, online supplement). Furthermore, residents report flooding of approximately 50 m inland and strong beach erosion during Haiyan. However, for this section of the coast (E1–E5, Fig. 1c) similar effects were observed after tropical storm Basyang in February 2014 which had been less intense, but had struck at high tide (Fig. S3 and Tab. S1, online supplement).

Storm erosion created a steep shoreface with an erosive scarp and uprooted palm trunks at the seaward slope of the first beach ridge. Residents report lateral erosion of more than 5 m due to the combined influence of Haiyan and Basyang along large sections of the beach (Tab. S1, online supplement). However, the formation of lobate washover fans in the back-barrier

depression is already visible on satellite images from November 2013 (Fig. 11b). The washover fans are restricted to a 10 m wide section behind the beach, but form prominent landforms with two distinct stratigraphical units (Figs 11c, 12a).

The basal unit 1 is related to flat washover lobes characterized by planar lamination. It overlies a weakly developed soil with a sharp contact and extends slightly further inland compared to the subsequent unit 2, which is characterized by landwards inclined beds with a steep terminal front. The internal structure is illustrated by the sedimentology (Fig. 12b) and µCT scans (Fig. 7) of sediment core BAN 4. Below the pre-Haiyan soil, a thin sheet of medium sand (PE) was found along the erosive cliff in the ridge (Fig. 12a) as well as in BAN 4 (Figs. 7, 12b). The sand layer covers an older palaeosol, composed of brown, slightly loamy sand. The basal Haiyan deposit is composed of a laminated section of unimodal, moderately sorted (1.8) medium-coarse sand (mean of 600-770 µm), rich in strongly reworked foraminifer tests (35-60%) dominated by *Calcarina* sp. and *Amphistegina* sp. (unit 1). The uppermost 19 cm (unit 2) are composed of coarser (mean of 730-1300 µm) and slightly less sorted (2.0) unimodal sand with similar species composition but a higher percentage of fresh or only slightly reworked foraminifers (60-80%) between 19 and 10 cm below the surface (Fig. 12b). A reference sample collected at the present beach at 0.5 m above msl (BAN $R_1$) is slightly finer than unit 1 (mean of 400 µm) and better sorted (1.6).

### 4.4.2 Bantayan B (BAN B)

Site BAN B is located in direct vicinity of an estuary mouth in northern Bantayan, lined by a 500 m wide intertidal reef platform (Fig. 11d). While the coast south of the estuary is characterized by a 2.8 m high cliff (above msl) in the Pleistocene coral reef limestone, a c. 100 m wide sand spit backed by mangroves forms the section to the north (Fig. 11e). In a shore-perpendicular direction, a beach ridge at 2.4 m above msl is followed by a flat mud plain with mangroves and reef outcrops at 0.9–1.1 m above msl (Figs. 13a, S1). Eyewitnesses report significant flooding by Typhoon Haiyan, while Basyang – different from the southern part of Bantayan – caused no marked inundation (E6–10 in Tab. S1, online supplement). Flood marks in the form of floated debris on top of the cliff or trapped in bushes and mangroves document inundation of at least 200 m inland. Minimum water levels decrease landwards from 3.1 m above msl directly at the coastline, to 2.1–2.3 m above msl at 60 m from the shoreline and only 1.4 m above msl at 160 m (Fig. 11e).

Haiyan-induced onshore sand deposition is less than 1–2 cm in the back-barrier mangroves. Thicker deposits occur in the form of up to 30 cm thick and 50 m wide washover fans at two sections behind breaches in the barrier (Fig. 11e). Coast-perpendicular transects (T2 and T3, Fig. 11e) reveal landward thinning trends from 28 cm at 10 m behind the barrier to only 5 cm at a distance of 40 m (T2, Fig. 3), and from 15 cm at 10 m behind the ridge to 1 cm at a distance of 40 m respectively (T3 in Fig. S1a, online supplement). Similar to site BAN A, the pre-Haiyan soil had formed in a thin sand sheet, which has a composition similar to the modern storm deposit and covers a second palaeosol at the base of the trenches (PE in Fig. 13a). It might be the deposit of a former storm.

In both washover fans, two successive sedimentary units were distinguished and investigated in detail for cores BAN 1–3 (transect 2; Figs 13b, S2). Unit 1 is massive to slightly normally graded, composed of unimodal, moderately sorted (1.5–2.6)

medium sand (mean = 250–320 µm) and shows a sharp boundary to the pre-Haiyan soil. It constitutes the entire typhoon layer in BAN 3, but is topped by a markedly thicker unit 2 in BAN 1 and 2. Unit 2 is composed of several planar to slightly inclined, normally or inversely graded beds of well-sorted (1.5–1.9) medium sand (mean = 300–480) with shell and coral fragments as well as pieces of litter. Both units contain >90% carbonates (aragonite and Mg-calcite). The Foraminifer assemblage is

dominated by *Calcarina* sp. (32%), *Amphistegina* sp. (19%) and *Ammonia beccarii* (19%). Most of the tests are strongly reworked (<17% fresh), though test preservation is much poorer (<1% fresh) in the palaeosol which is mainly composed of *Amphistegina* sp. (22%), *Ammonia beccarii* (21%), *Elphidium craticulatum* (12%) and *Quinqueloculina* spp. (12%). While unit 1 thins landwards in T3, neither a clear thinning tendency nor a fining trend is detectable in T2. Unit 2 is restricted to the proximal part of the washover fans (BAN 1–2, Figs 13, S2 and A–B, Fig. S1a) and rapidly thins landwards. For comparison

with modern environments, reference samples were collected at 0.6 m above msl (BAN $R_2$) and at msl (BAN $R_3$). Both samples show a similar granulometry of unimodal, moderately sorted (1.9-2.0) medium sand (mean of 280-550 µm).

### 4.5 Inter-site comparison of sediment characteristics

#### 4.5.1 Geochemistry, mineralogy and foraminifers

The comparison of XRD data from all four sites reveals two general types of mineralogical compositions. While sediments

from the carbonate environments (MOL, HER, BAN) are dominated by calcite, Mg-calcite and aragonite (at least 80%) with minor percentages of feldspar or quartz, the samples from the siliciclastic coast (TOL) are mainly composed of feldspar and amphibole with a minor percentage of quartz (Fig. 8a).

A principal component analyses (PCA) on foraminifer data from the carbonate coasts indicates three PCs which explain 60% of the total variability. PC1 (31%) is characterized by positive loadings for the genera *Rosalina* sp., *Quinqueloculina* spp.,

*Peneroplis* sp., *Milionella* sp., *Globigerinoides* sp. and *Coscinospira* sp., while *Calcarina* and *Amphistegina* are negatively correlated. PC2 (18%) shows positive scores for *Amphistegina* sp. and strong reworking, and negative ones for *Heterostegina* sp. PC3 (11%) is positively correlated with *Challengerella* sp. and negatively with *Spirillina* sp. and *Schlumbergerella* sp. Plotting of PC1 against PC2 (Fig. S4, online supplement) reveals differences between sediment from the foreshore at Molocaboc (MOL $R_{subt}$) on the one hand, and beach reference samples and storm deposits from all locations on the other.

Differences between beach and storm deposits from different sites are less pronounced.

#### 4.5.2 Granulometry

PCA combining granulometric data from all sites reveals three PCs that explain 82% of the total variability. Plotting of PC1 (positive loadings for mean, sorting, gravel and mode, negative ones for skewness and medium sand) versus PC2 (positive loadings for mean, skewness and mode, negative ones for sorting and mud) reveal both site-dependent properties as well as

distinct clusters for samples from units 1, units 2 and the palaeosols (Fig. 8b1).

In addition, PCAs on grain-size parameters were performed for each site separately. For TOL, PC1 (62.8%) and PC2 (33.5%) explain 96% of the total variability. Plotting of PC1 (positive loadings for skewness, mode and medium sand, negative ones for sorting and mud) versus PC2 (positive scores for sorting, mode and medium sand, negative ones for skewness) reveals clusters for unit 1, unit 2 and the pre-Haiyan soil (Fig. 8b3). Samples from HER reveal three PCs explaining 84% of the total variability. Plotting of PC1 (positive loadings for sorting, mud and fine sand, negative ones for skewness and coarse sand) versus PC2 (positive scores for fine and medium sand, negative ones for mode and sorting) reveals clustering of unit 1 and the palaeosol, while unit 2 forms two separate sample groups (HER 10 and all other trenches) (Fig. 8b2). At BAN, three PCs explain 86% of the total variation. Plotting of PC1 (positive loadings for mean, skewness and medium sand, negative ones for sorting and mud) versus PC2 (positive scores for mean, sorting, mode and gravel, negative ones for medium sand) reveals clusters for unit 1, unit 2 and the palaeosol, whereas reference samples from the upper and lower beach plot into the cluster of unit 2 (Fig. 8b4), pointing to this area as the main sediment source.

## 5. Discussion

### 5.1 Sedimentary footprint of Typhoon Haiyan on the Philippines

Based on eyewitness accounts and the interpretation of satellite images, most of the documented storm deposits can unambiguously be related to Typhoon Haiyan. Even at BAN A, where eyewitnesses report strong coastal erosion by tropical storm Basyang for the period between Haiyan and the field survey, satellite images from November 11[th] clearly document the formation of the washover fans by Typhoon Haiyan. Only at TOL, where both satellite images and eye witnesses cannot unambiguously relate all of the documented onshore deposits with Haiyan, the proximal parts of the washover sediments (unit 2 at TOL) cannot be excluded to be associated with post-Haiyan storm waves of Basyang. In addition to the fine-grained storm deposits reported in this study and summarized in figure 14, Haiyan moved block- and boulder-sized reef-rock clasts at the coast of Eastern Samar, which have been discussed elsewhere (May et al., 2015b).

### 5.1.1 Sandy onshore deposits

Based on sedimentary and morphological criteria, the here presented sandy onshore deposits of extreme wave events are classified into sand sheets and washover fans. The classification reflects different flooding regimes and, therefore, is assumed to represent hydrodynamic processes that lead to different sediment characteristics (Fig. 14).

Sand sheets: separated from the underlying soil by a layer of bended grass, the base of the typhoon deposits at TOL, HER, BAN B and MOL (the local units 1) is formed by slightly normally graded to massive layers of sand (some of the layers were too thin to prove potential grading without laboratory analyses). All these sand sheets extend at least 100 m inland, are relatively thin (<10 cm), and exhibit clear landward thinning and slight fining trends (Fig. 3). Their appearance is similar to

storm deposits formed under inundation regimes related to extensive flooding of back-barrier marshes described by Donnelly et al. (2006) or Wang and Horwitz (2007). Indeed, complete inundation of coastal barriers and back-barrier areas at HER, TOL, BAN B and MOL is documented by flood marks. While at MOL and BAN B flooding is facilitated by the absence of a pronounced beach ridge or by a nearby river mouth, flow depths of nearly 4 m above surface and the complete submergence of barriers with crests more than 3 m above msl at TOL and HER are either due to the high storm surge levels (TOL) or due to a combination of storm surge, high storm waves and related infra-gravity waves such as surf beat (HER) (Bricker et al., 2014; Roeber and Bricker, 2015; Kennedy et al., 2016). The associated overland flow generally followed shore-perpendicular directions, if not re-directed by shore-parallel wall structures (TOL, Fig. 5). As indicated by scour marks (μCT scan of TOL 8, Fig. 7) and a mostly normally graded structure of these sand sheets, sediment dynamics related to this inundation regime (i.e., inundation overwash; cf. Donnelly et al., 2006) are assumed to be characterized by turbulent flow conditions and deposition from suspension. Even if clear suspension grading as described by Jaffe et al. (2011) could not be detected, the deposits are very similar to suspension-settled sediments described by Williams (2009) for Hurricane Rita at the US coast. At least at TOL and HER the deposition of these sand sheets was influenced by few tsunami-like flooding pulses due to infra-gravity waves or seiches that amplified peak inundation (Mori et al., 2014; Roeber and Bricker, 2015). Comparison of granulometry as well as foraminifer taphonomy and species composition with reference samples point to the beach as the dominant sediment source (BAN $R_{1-3}$) rather than foreshore (MOL $R_{subt}$) or other environments (Figs 8b, S4). Nevertheless, obvious differences in the granulometry and faunal composition of the sand sheets and modern beach sand (Fig. 8b, S4) may indicate also minor contributions of sediments from other source areas (the foreshore, deeper water, or landward areas), as reported by Pilarczyk et al. (2016) for deposits of Typhoon Haiyan from Tanauan (Leyte) and Basey (Samar). Alternatively, at least the differences in foraminifer taphonomy may reflect alteration of the sediments due to wearing and fracturing of foraminifer tests during transport in high energy flows (Quintela et al., 2016).

Washover fans: Unit 2 at TOL, HER, and BAN B, as well as the entire storm deposits at BAN A (the local units 1 and 2) are formed by lobate landforms of several decimetres thickness directly behind the barrier. As they (a) are restricted to the proximal part of back-barrier depressions (within <50 m from the shoreline), (b) form lobes with a gently inclined upper surface (1−5°) and a steep landward front, (c) show multiple grading within horizontally to inclined laminated sections, and (d) reveal landward thinning and fining trends, these features resemble typical storm-induced washover fans, e.g. described by Sedgwick and Davis (2003), Phantuwongraj et al. (2013), or Williams (2015). They formed due to wave-induced sediment transport over the coastal barriers. This took place after the first flooding pulses inundated the coastal barriers, most likely during the peak of the storm surge when the largest storm waves occurred (Williams, 2009). Where coastal barriers were already inundated due to the wind induced storm surge (at TOL and BAN B), or due to the impact of infra-gravity waves (at HER, May et al., 2015b), deposition is assumed to be caused by waves breaking at the inundated barrier. At site BAN A on the other hand, where a basal sand sheet (unit 1 at TOL, BAN B and HER) is absent, flood marks indicate maximum water levels lower than the coastal barrier, and deposition was related to confined overwash (i.e., run-up overwash; cf. Donnelly et al., 2006). The internal

stratification and washover morphology allow for a discrimination of a basal section with horizontal bedding associated with flat lobes (unit 1) and a section with steeply inclined layers with a steep avalanching front on top (unit 2). Horizontal lamination is interpreted as the result of initial barrier overtopping into dry back-barrier depressions associated with high flow velocities, whereas steep lobes with inclined bedding are associated with delta-front sedimentation into already flooded back-barrier

depressions (Sedgwick and Davis, 2003; Switzer and Jones, 2008), e.g. due to high levels of the preceding storm surge or intensive rainfall.

Anyway, the washover fans either form isolated structures such as at TOL, HER and BAN B, or coalescing washover terraces (BAN A). Since breaching is associated with barrier erosion and radial spread of water and sediment into back-barrier depressions, granulometry, foraminifer assemblages and taphonomy indicate that storm sediments were mainly derived from

the beach (Figs. 8b, S4). The swash of multiple individual waves generates successions of laminae which are probably caused by density separation in mixtures of heavy minerals, quartz and shell fragments during transport as traction load (Komar and Wang, 1984). Vertical changes of granulometry and faunal composition within individual washover deposits – such as the upward trend towards coarser and stronger reworked deposits at BAN A (Fig. 12) – could be related to slightly different sediment sources as a result of a successively changing beach profile. On the other hand, the coarsening trend could just be an

artefact of the reduced settling velocity of platy shell fragments, which are particularly abundant in this section of BAN 4 (Fig. 7), compared to more spherical grains (Woodruff et al., 2008).

### 5.1.2 Intertidal coral-rubble ridges

Considering the interviews with local fishermen and the fact that parts of the intertidal coral ridge at Carbin Reef in February

2014 covered reef organisms which must have been alive shortly before, formation or at least significant heightening can unambiguously be attributed to Typhoon Haiyan. Coral rubble ridges have repeatedly been reported to be a typical cyclone signature (Scheffers et al., 2012), e.g. on Funafuti Atoll, where Maragos et al. (1973) describe a cyclone-generated, intertidal coral ridge dominated by sand to boulder-sized rubble derived from the foreshore. Diving surveys before and after Haiyan revealed significant impact of the typhoon at the seaward reef slope while organisms on the intertidal platform were nearly

untouched (Reyes et al., 2015), pointing to the entrainment of sediment from the foreshore and its deposition at the reef edge. Since ridge formation requires the repeated impact of breaking waves and wave swash is attenuated by the high porosity of rubble ridges (Spiske and Halley, 2014), ridge generation during Haiyan is mainly due to the impact of multiple storm waves breaking on top of the inundated reef platform.

The two morpho-sedimentary ridge units at Carbin allow for two possible explanations: A first interpretation is that the entire

ridge was formed by Typhoon Haiyan, whereas the distinct units would reflect changing wind and wave directions. In accordance with a rotation of wind directions reported for the passage of Haiyan, unit 2 is present only at the W-exposed section of the ridge and may be interpreted as the result of stronger wind waves connected with increased destruction of the

living reef at Carbin's slope as indicated by the fresh, angular coral fragments incorporated in unit 2. On the other hand, the more pronounced algae cover on coral rubble of unit 1 and the eyewitness accounts of ridge occurrence after Haiyan might also be explained by the impact of several typhoons. In this case, formation of the initial ridge (unit 1) that was not pronounced enough to be recognized by local fishermen, could have taken place during former storms. Afterwards, Haiyan increased its height by adding fresh coral rubble on top (unit 2), which made the ridges widely recognizable.

## 5.2 Spatial variability of fine-grained typhoon signatures

The spatial distribution and sedimentary structure of deposits formed by Typhoon Haiyan was critically influenced by both local setting and the hydrodynamic characteristics during inundation. Although onshore transport of sand is rather ubiquitous, widespread sand sheets with a significant inland extent >100 m in the study area are restricted to locations with exceptional surge and/or inundation levels. This includes the exposed coastlines of Eastern Samar (HER) and the funnel-shaped San Pedro Bay (TOL), where a remarkable amplification of surge levels to values >8 m above msl occurred during Haiyan (Mori et al., 2014). However, pressure and wind driven surge alone do not explain the high inundation levels for Eastern Samar, since numerical storm-surge models combined with phase-averaged wave models infer rather low water levels of ~2 m (Bricker et al., 2014). While surge levels in the San Pedro Bay were additionally heightened by wave reflection in the enclosed embayment (Mori et al., 2014), phase-resolved wave models imply that the surprisingly high flooding levels along Eastern Samar result from infra-gravity waves, caused by non-linear wave interactions with the fringing reef (Roeber and Bricker, 2015; May et al., 2015b; Kennedy et al., 2016).

On the other hand, washover features were even formed in areas with limited flooding levels and restricted landward inundation such as Bantayan and Northern Negros. Since washover deposits require high waves capable to overtop coastal barriers, their local occurrence seems to be predetermined by bathymetry (river estuaries) and coastal morphology (pre-existing gaps or depressions in sandy barriers) as observed on Bantayan. Hence, they are limited to small sections of the coast even in heavily affected areas. For the localized occurrence of coral ridges on the reefs north of Negros, obligatory requirements seem to be the presence of intertidal reef platforms with coral rubble in the foreshore zone, exposure towards the main direction of the storm waves, and waves high enough to entrain the foreshore sediment and to lift it onto the reef platform.

Local geology, geomorphology and sediment source are also the main factors determining the composition of the investigated typhoon deposits (Fig. 8). On the one hand, variations of total sediment composition (mineralogy and granulometry) between different sites are more significant than variations between different units at individual sites (Fig. 8a, b). The granulometry of storm-transported sediment varies with the sediment availability at the beach and the foreshore zone so that dominant grain size varies between HER, TOL and BAN (Fig. 8b). Geochemistry and mineralogy basically reflect differences between siliciclastic and carbonate coasts (Fig. 8a). Varying site-specific compositions have also been reported for foraminifer assemblages of Haiyan deposits by Pilarczyk et al. (2016). Likewise, clearly laminated washover deposits (alternation of dark and light laminae) are linked to the presence of heavy minerals at siliciclastic coastlines (TOL), while bedding structures are

less prominent in carbonate environments (BAN and HER). On an intra-site level, on the other hand, mineralogy and geochemistry (due to insignificant differences) and microfauna (due to a limited dataset) only allow for a separation between storm deposits and underlying palaeosols, while sedimentary structures and grain-size data enable further discrimination of distinct subunits (Fig. 8b). Although the number of reference samples from recent environments is very limited in this study,

the granulometric differences between internal sublayers of the same storm deposit seem to be related to varying sediment sources (beach vs. foreshore) or to different hydrodynamic transport conditions as it has been described by Switzer and Jones (2008). This is the case for the units 1 and 2 at sites TOL, HER and BAN (Fig. 8b). The large scatter within the unit 2 deposits at HER might be explained by deposition of the HER 10 deposits by backwash, since the core was taken close to a fluvial channel.

**5.3 Implications for palaeotempestology**

Since the sedimentary characteristics of Typhoon Haiyan's deposits show significant site-specific variations, it is not possible to infer one particular storm signature type. Nevertheless, storm deposits can clearly be distinguished from most other depositional processes on the basis of granulometry, internal structures, mineralogy and faunal composition. Only tsunamis might be capable to produce similar features, due to comparable hydrodynamic characteristics that potentially allow for barrier

overwash, landward transport of sand for hundreds of meters, and the generation of waves strong enough to entrain and lift subtidal coral rubble onto intertidal reef platforms (Shanmugam, 2012). Although numerous features have been established to discriminate between tsunami and storm deposits, including among others thickness, lateral extent, granulometry, source areas and sedimentary structures of the event deposits (e.g., Morton et al., 2007; Switzer and Jones, 2008), most of these sedimentary indicators seem to be of local value only and cannot serve as universal discrimination criteria (Shanmugam, 2012).

Unfortunately, only very few tsunami signatures that might facilitate discrimination by providing typical site-specific features (e.g., Kortekaas and Dawson, 2007) have been described for the Philippines (Imamura et al., 1995), making it hard to evaluate which of the sedimentary characteristics documented for Typhoon Haiyan in this study are unique for the impact of local typhoons.

So far, coral ridges have been reported to be exclusively generated by cyclones (Scheffers et al., 2012), which seems

straightforward. Ridge formation requires repeated breaking waves as observed during cyclones (Nott, 2006), while the small number of inundation pulses characteristic for tsunamis tends to produce randomly scattered boulder fields (Richmond et al., 2011; Weiss, 2012). However, preservation of coral ridges is often limited (Baines and McLean, 1976), and age determination remains challenging due to potential reworking of the components (Scheffers et al., 2014). On the other hand, suspension-settled, normally graded sand sheets with large inland extents as described at TOL or HER are typical signatures of tsunamis

as well (e.g., Jankaew et al., 2008), and also washover fans with internal lamination as present at BAN or TOL have been reported for both storms (Switzer et al., 2012) and tsunamis (Atwater et al., 2013). Likewise, most features described for the sand sheets and washover fans in the study area have already been observed for tsunami deposits: The composition and

granulometry of the storm-induced sand sheets and washover fans presented here are mainly controlled by local geology rather than transport processes typical for cyclones. With most sediment derived from the littoral zone, the main sediment origin of the Haiyan deposits is similar to that of typical tsunami deposits (e.g., Brill et al., 2014), and apart from that not analysed with sufficient detail to determine secondary source areas that might enable better differentiation from tsunami deposits. Finally,

the documented landward thinning and fining trends in the Haiyan deposits are similar to those observed in many tsunami deposits (e.g., Goto et al., 2008).

Nevertheless, by adding local data to the knowledge on the manifold expressions of storm deposits, the sediments accumulated by Haiyan offer some valuable considerations regarding the interpretation of palaeoevent deposits in geological records during future studies both in the Philippines (e.g. for interpreting the young palaeoevents documented on Bantayan) and in general.

On the one hand, with not more than ~250 m the inland extent of the Haiyan-laid sand sheets presented here seems to be limited compared to sandy deposits of many recent tsunamis with comparable inundation levels in settings with flat topography that extend landwards for several kilometres (e.g., Jankaew et al., 2008; Goto et al., 2011). This is true even at site TOL that experienced onshore flooding levels >5 m above msl, and where the flat topography did not hinder lateral inundation and sediment transport. Although the landward limit of coastal inundation may be indicated by mud deposits rather than by sand

layers (e.g., Williams, 2010; Abe et al., 2012; Goto et al., 2014), and our comparison is not based on tsunami deposits from the same location (e.g., Kortekaas and Dawson, 2007), our findings seem to corroborate previous conclusions that the landward extent of tsunami sand sheets in low lying coasts tends to be larger in comparison with sandy storm deposits (cf. Morton et al., 2007). On the other hand, the combined occurrence of (i) a thin (i.e., a few cm), massive to slightly normally graded basal unit formed by suspension-settling due to surge-related extensive coastal flooding (unit 1 at TOL and HER), and (ii) a laminated,

swash-induced, spatially limited washover unit on top (unit 2 at TOL, BAN B and HER, units 1 and 2 at BAN A) that in case of HER can both be unambiguously related to Haiyan and therefore a single storm event, might be rather indicative for cyclone-generated deposition. However, at TOL we cannot exclude that part of this succession (i.e. the washover unit on top) is the result of deposition by Basyang, since its time of formation could not be proven by eyewitnesses or satellite images. In this case, the association of the two units might also be explained by several storm events in a quick sequence, whereas the strong

beach erosion during a major event such as Supertyphoon Haiyan would only increase the susceptibility of the sedimentary coastline towards storm overwash during follow-up events.

## 6. Conclusions

The deposits of Typhoon Haiyan are strongly influenced by local factors causing a wide variety of site-specific sedimentary and morphological characteristics. Nevertheless, in spite of their spatial variability, the deposits exhibit several storm-related

depositional patterns – including washover fans, sand sheets and coral rubble ridges – that can be related to specific hydrodynamic processes and resemble typical storm features from all over the world (Fig. 14).

Massive to normally graded onshore sand sheets extend 100–250 m inland, tend to fine and thin landwards, and are related to suspension settling during initial surge pulses that cause widespread inundation of coastal lowlands and complete submergence of the coastal barrier. They have formed in areas of significant amplification of the storm surge. In contrast, washover fans are composed of small (10–50 m inland) sand lobes with steep landward fronts and a distinct internal stratification that occur

localized behind breaches or depressions in coastal barriers as the result of traction transport by overtopping waves and repeated overwash. Coral rubble ridges were formed on intertidal reef platforms by storm waves entraining coral fragments from the reef slope and breaking at the reef edge.

Since Haiyan was one of the most powerful cyclones ever recorded, these and other findings from the Philippines particularly add to the general knowledge of extreme wave deposits and, ultimately, may contribute to discriminating sediments of strong

cyclones and tsunamis. While coral rubble ridges so far seem to be a unique feature of strong storms, the sandy onshore deposits left by Haiyan – both sand sheets and washover fans – resemble those generated by tsunamis in terms of sedimentary structure, granulometry and sediment sources (Fig. 14). However, the inland extents of sand sheets documented in this study are significantly smaller compared to those of large tsunamis with comparable flooding levels in similar topographical settings. Although more extensive sand layers have been reported for other TCs, and moderate tsunamis generate sand sheets in the

range of the Haiyan deposits, the inland extent of onshore sand sheets at least provides a valuable tendency for the discrimination of cyclones and strong tsunamis. In addition, the combined occurrence of basal sand sheets and overlying confined and well-stratified sand units may be indicative for cyclones, since they result from the succession of surge-related extensive coastal inundation and subsequent wave overwash during Haiyan at HER, which is not known to be typical for tsunamis. But the ambiguous origin of the respective units at TOL, where they might as well result from two successive storms,

prevents a more definite conclusion.

**Acknowledgements**

This research was financially supported by the Faculty of Mathematics and Natural Sciences, University of Cologne (UoC), and a UoC Postdoc Grant. Invaluable logistic support was provided by Karen Tiopes and Verna Vargas (Department of Tourism, Leyte Branch). We are very thankful for the great hospitality throughout the Visayas archipelago and the first-hand

insights provided by local interviewees, which is even more admirable considering the trauma after the disaster. Ramil Villaflor is acknowledged for guiding us safely through the islets north of Negros.

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

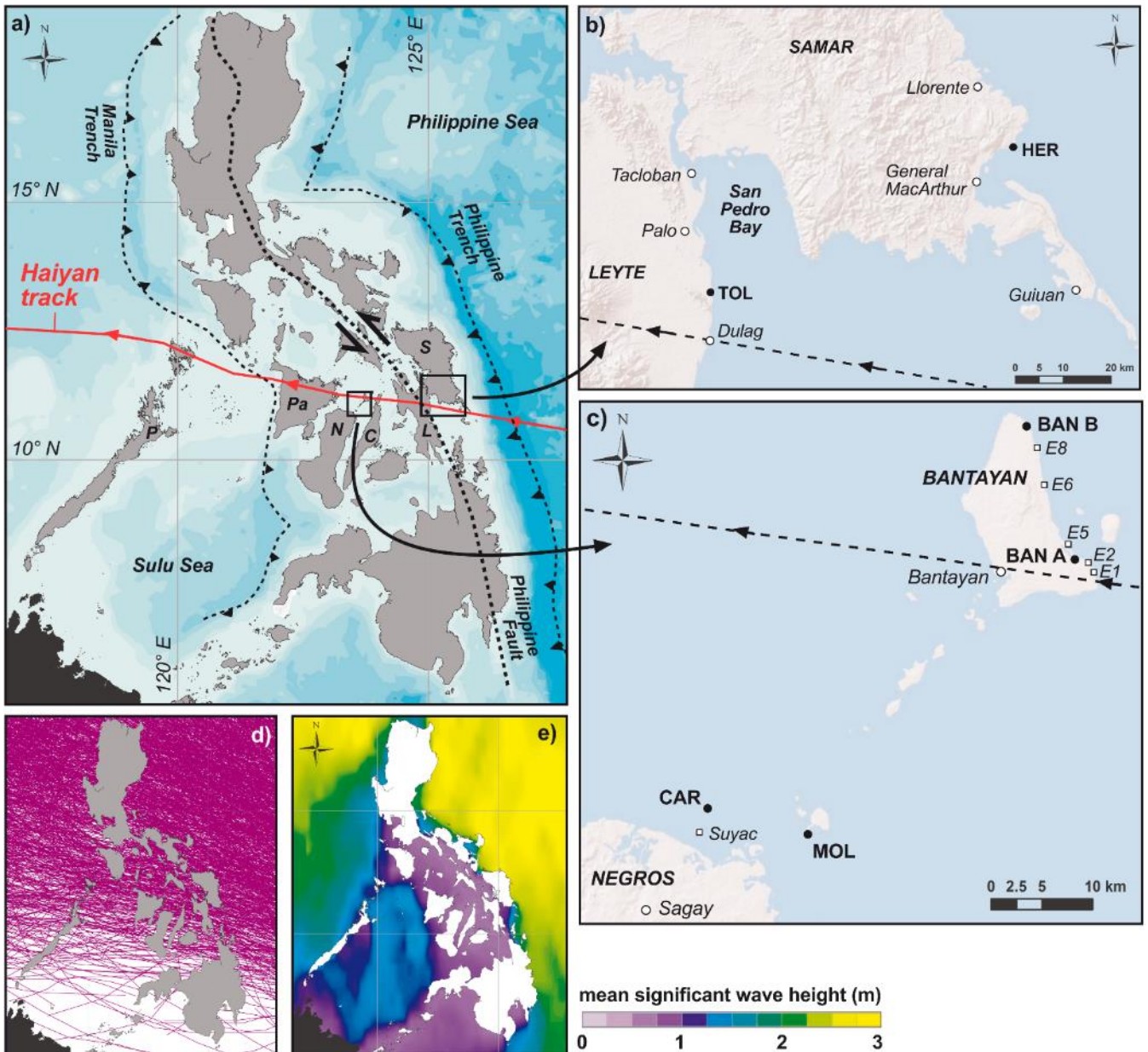

**Fig. 1: Overview of the study area. a) Philippine archipelago with main tectonic structures (Rangin et al. 1989), track of Typhoon Haiyan (NDRRMC 2014), and position of research areas on Samar and Leyte (b), as well as northern Negros and Bantayan (c) (based on ESRI basemaps). S – Samar, L – Leyte, C – Cebu, N – Negros, Pa – Panay, P – Palawan. d) Historical cyclone tracks crossing the Philippines since the beginning of weather recording (NOAA Historical Hurricane Track Pool) illustrate a decreasing cyclone frequency from north to south. e) Mean significant wave heights (Navy METOC 2014) indicate highest wave energy during winter monsoon along the exposed east coasts and usually calm conditions in the interior parts of the archipelago.**

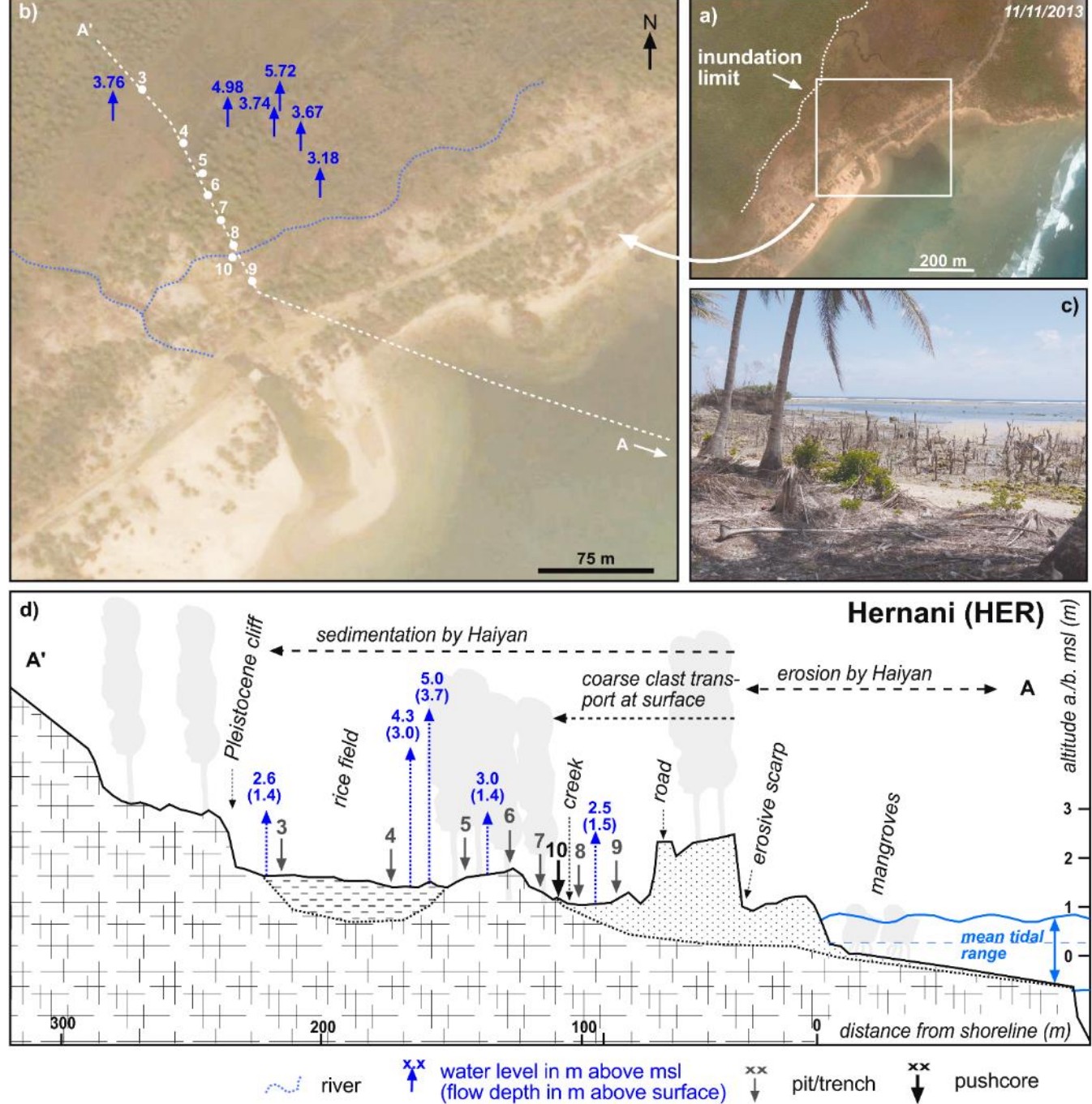

**Fig. 2: Hernani (HER) study site with documented Haiyan flood marks and positions of sampled typhoon deposits. a)** Local Typhoon Haiyan inundation limit (based on Google Earth/Digital Globe 11/11/2013). **b)** Onshore sediments were investigated along a coast-perpendicular transect (A–A') crossing the flooded area. **c)** Destroyed mangroves in front of the beach (view from the shore-parallel main road, photography: February 2014). **d)** Topographical cross section (A–A') with sampling sites. With water levels of at least 5 m above msl flooding during Haiyan overtopped the sandy coastal barrier, destroyed the road on its top, and transported sand nearly 250 m inland to the foot of the former cliff.

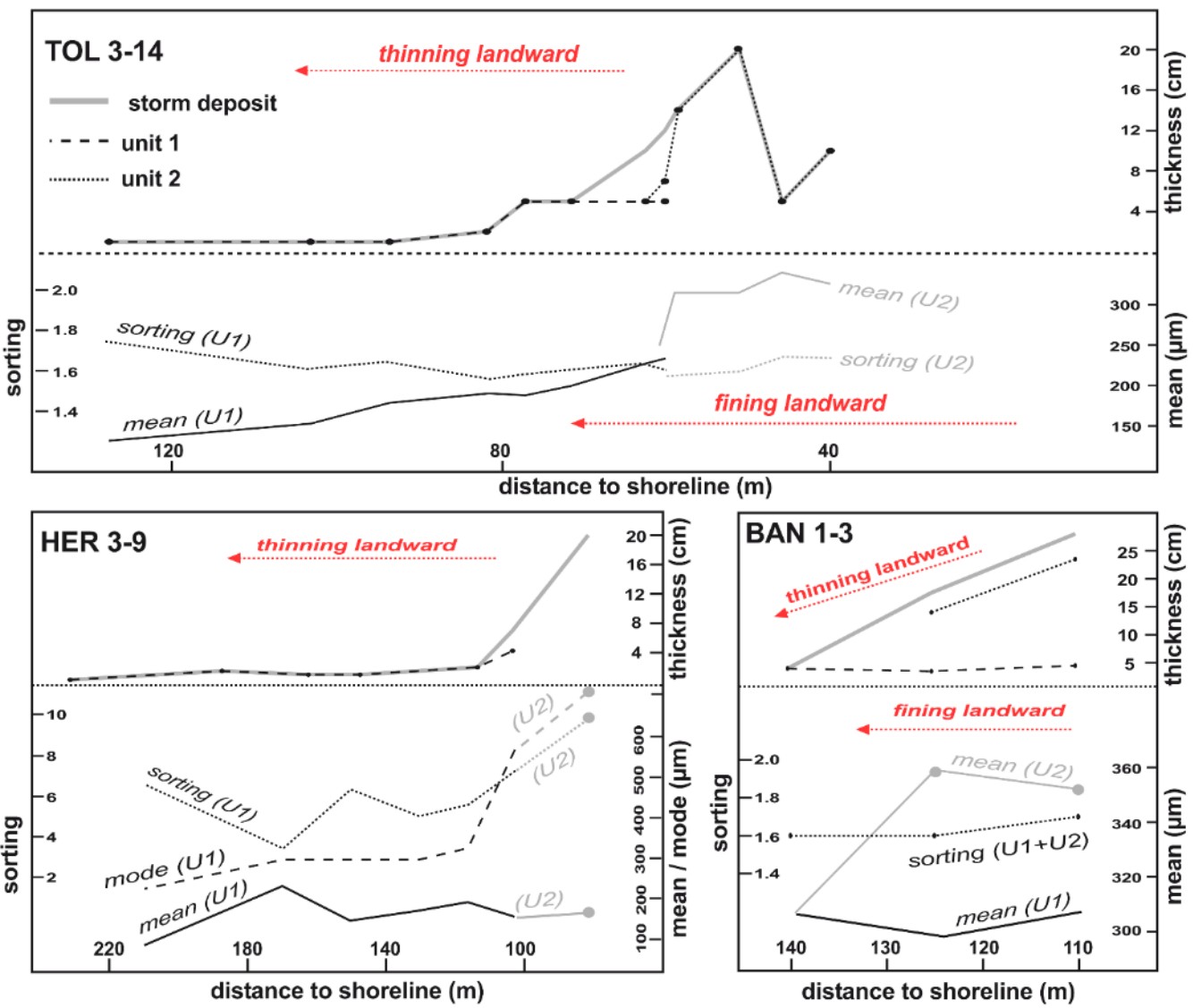

**Fig. 3: Landward trends of grain size and thickness in typhoon deposits. Onshore deposits at sites TOL, HER and BAN B tend to thin landward. While the deposits at HER and BAN are also characterized by fining trends of the mean grain size, a monotonic landward fining is not observed at TOL. Clear trends in sorting are not detected at all sites. Note the different scales.**

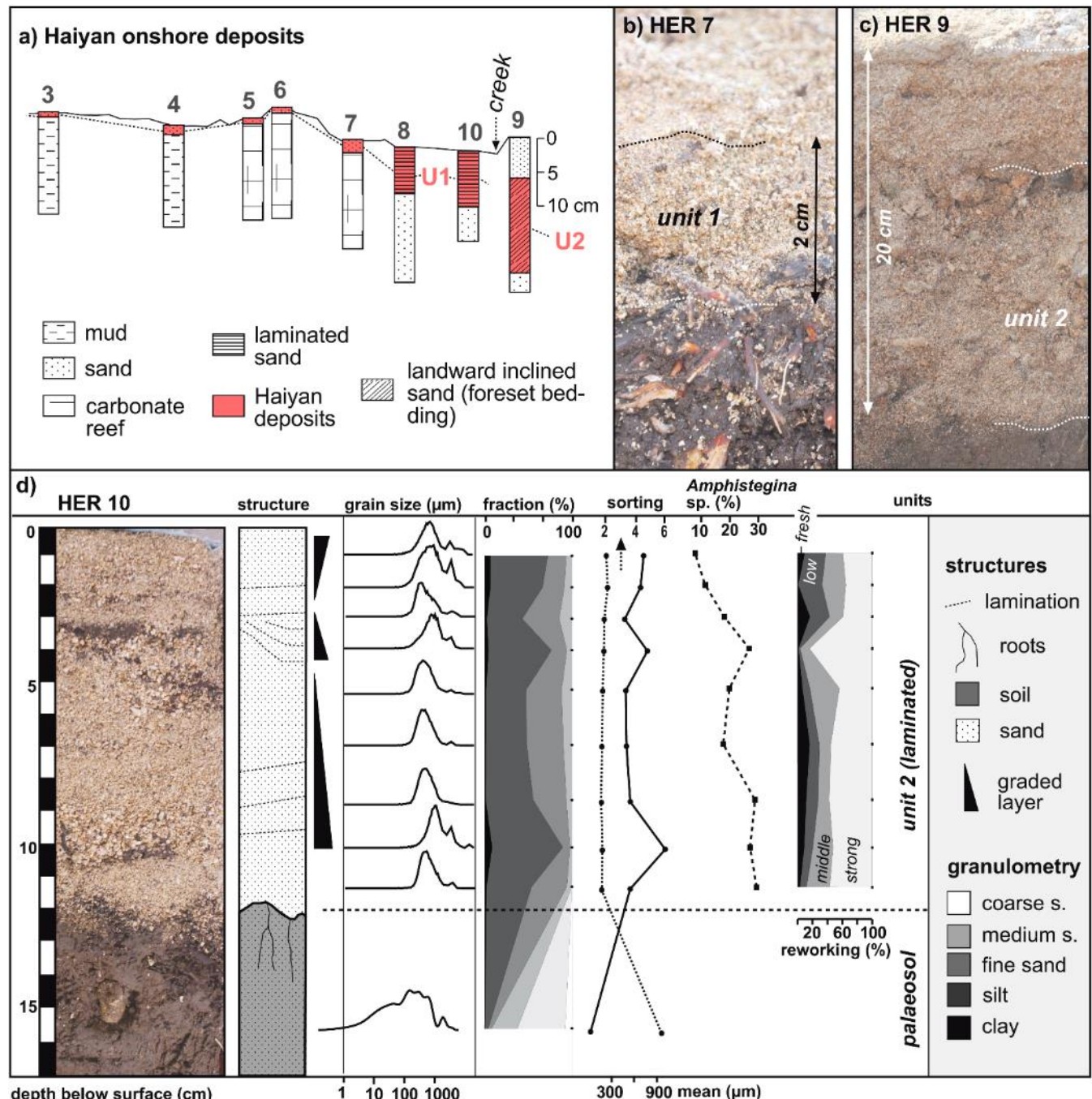

**Fig. 4: Onshore deposits of Typhoon Haiyan at site HER. a)** Landward transect illustrating the succession of typhoon deposits (see Fig. 2 for location). The basal unit 1 (U1) continuously thins landwards, while unit 2 (U2) is restricted to the proximal part of the coastal plain (HER 9). **b)** The 2 cm thick unit 1 at HER 7 (photography: February 2014). **c)** The 20 cm thick unit 2 at HER 9 (photography: February 2014). **d)** Sedimentary characteristics of HER 10. A 12 cm thick typhoon layer can clearly be separated from the palaeosol. The storm layer reveals repeated normally and inversely graded laminae, in which the amount of strongly reworked foraminifera is highest within the coarse basal sections.

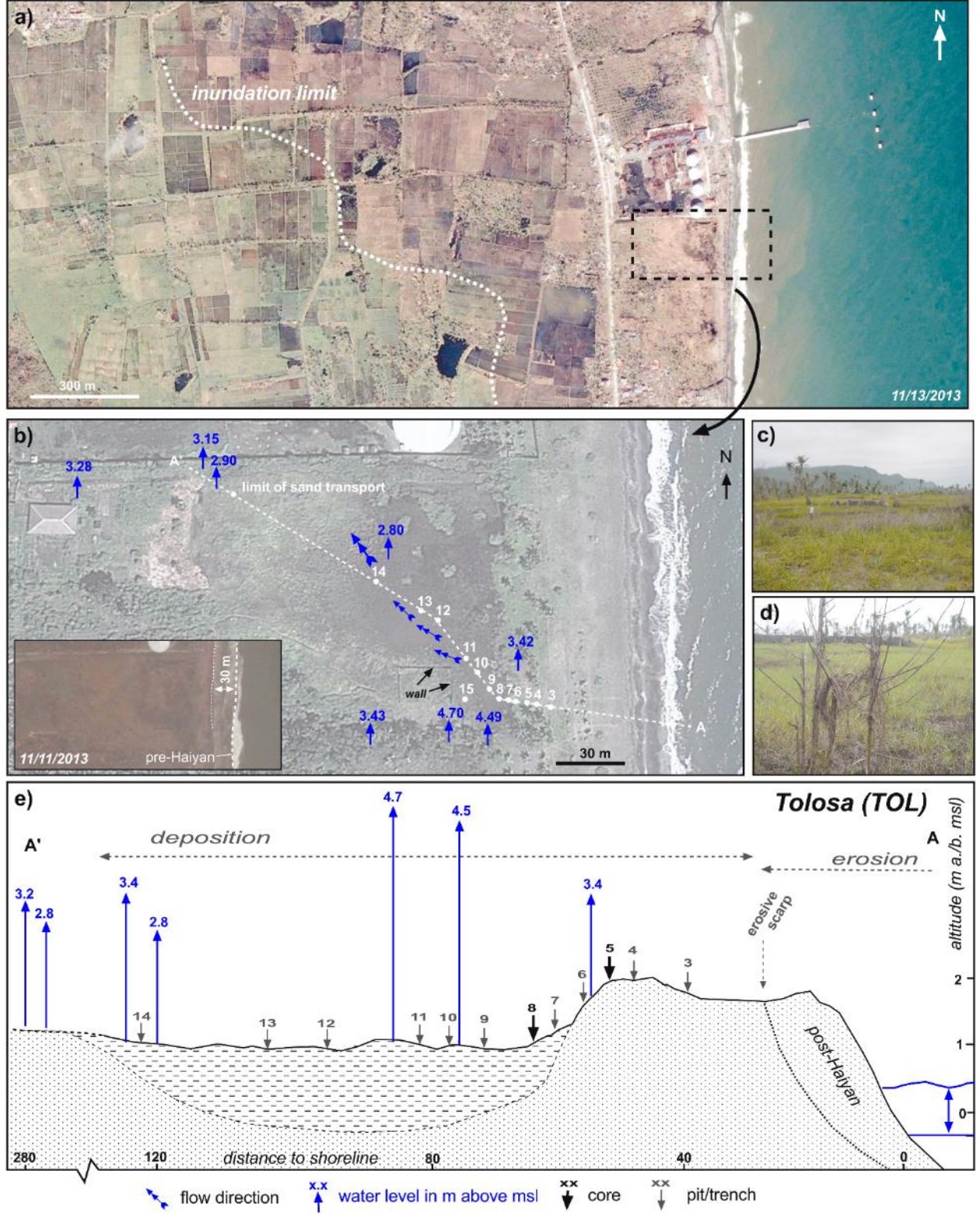

**Fig. 5: Tolosa (TOL) study site with documented Haiyan flood marks and positions of sampled typhoon deposits. a) Local inundation limit reached several 100 metres inland (based on Google Earth/Digital Globe 11/11/2013, i.e. immediately after Haiyan). b) Onshore sediments were investigated along a transect (A–A') crossing the inundated area (based on Google Earth/Digital Globe 02/23/2012; insert: Google Earth 11/11/2013). c) Westward view from the beach ridge over the back-barrier marsh. d) Flood debris in bushes served as an indicator for flow depth. e) With flow levels of at least 4.5 m above msl, Haiyan flooded the beach ridge entirely and transported sand about 150 m inland.**

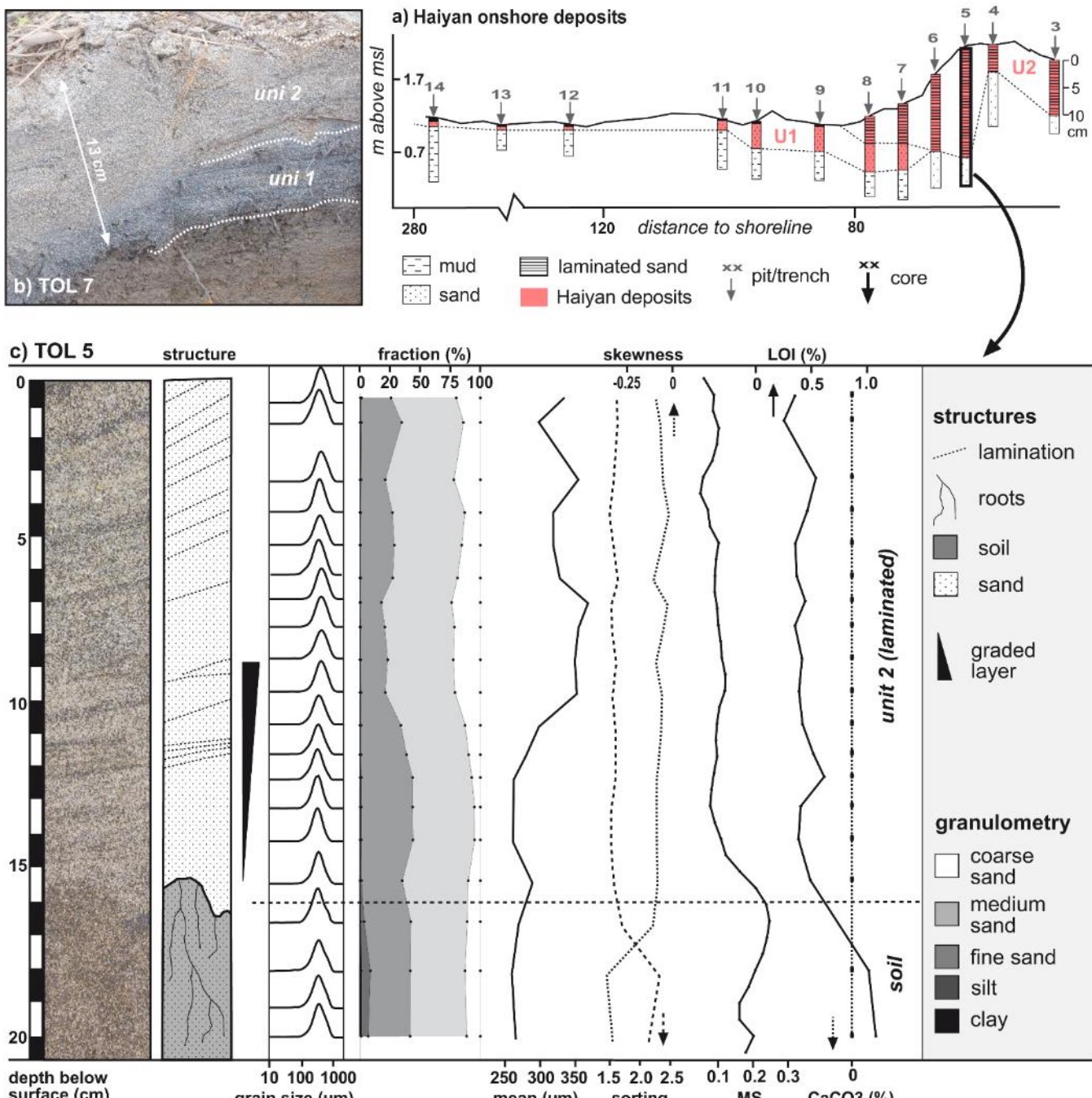

**Fig. 6: Typhoon Haiyan onshore deposits at site TOL. a) Transect illustrating the succession of typhoon deposits in landward direction. A graded to massive basal unit (U1) is covered by a laminated unit (U2) in the proximal part of the coastal plain (TOL 3–8). b) The 10 cm thick deposit at TOL 7 consists of massive sand at the base (unit 1) and laminated sand on top (unit 2). c) Sedimentary characteristics of TOL 5. The 16 cm thick, laminated typhoon layer can clearly be separated from the palaeosol due to lower LOI and magnetic susceptibility (MS).**

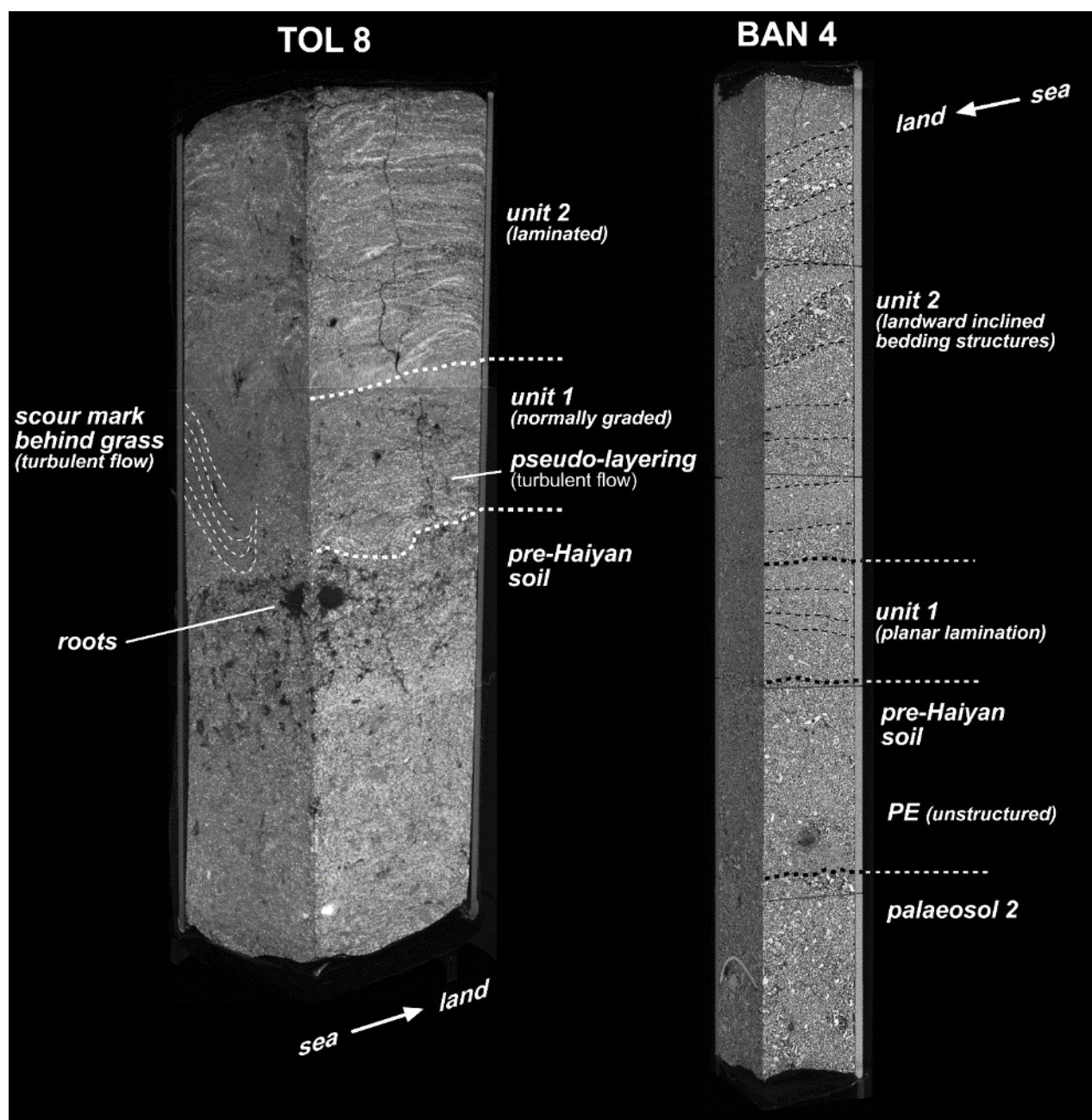

**Fig. 7: μCT scans of sediment cores TOL 8 and BAN 4 (as for locations see figures 5d and 10f, respectively). The 3D data support the documentation and interpretation of sedimentary features and different units within the typhoon deposits. The basal unit of TOL 8 is slightly normally graded and characterized by scour structures behind bended grass stems (unit 1). The upper part is horizontally laminated (unit 2). In BAN 4 the deposit of Typhoon Haiyan clearly contrasts the unstructured pre-Haiyan sediments composed of two palaeosols separated by an older sand sheet. It is divided into a planar bedded unit 1 at the base, and steeply landward inclined laminae in unit 2.**

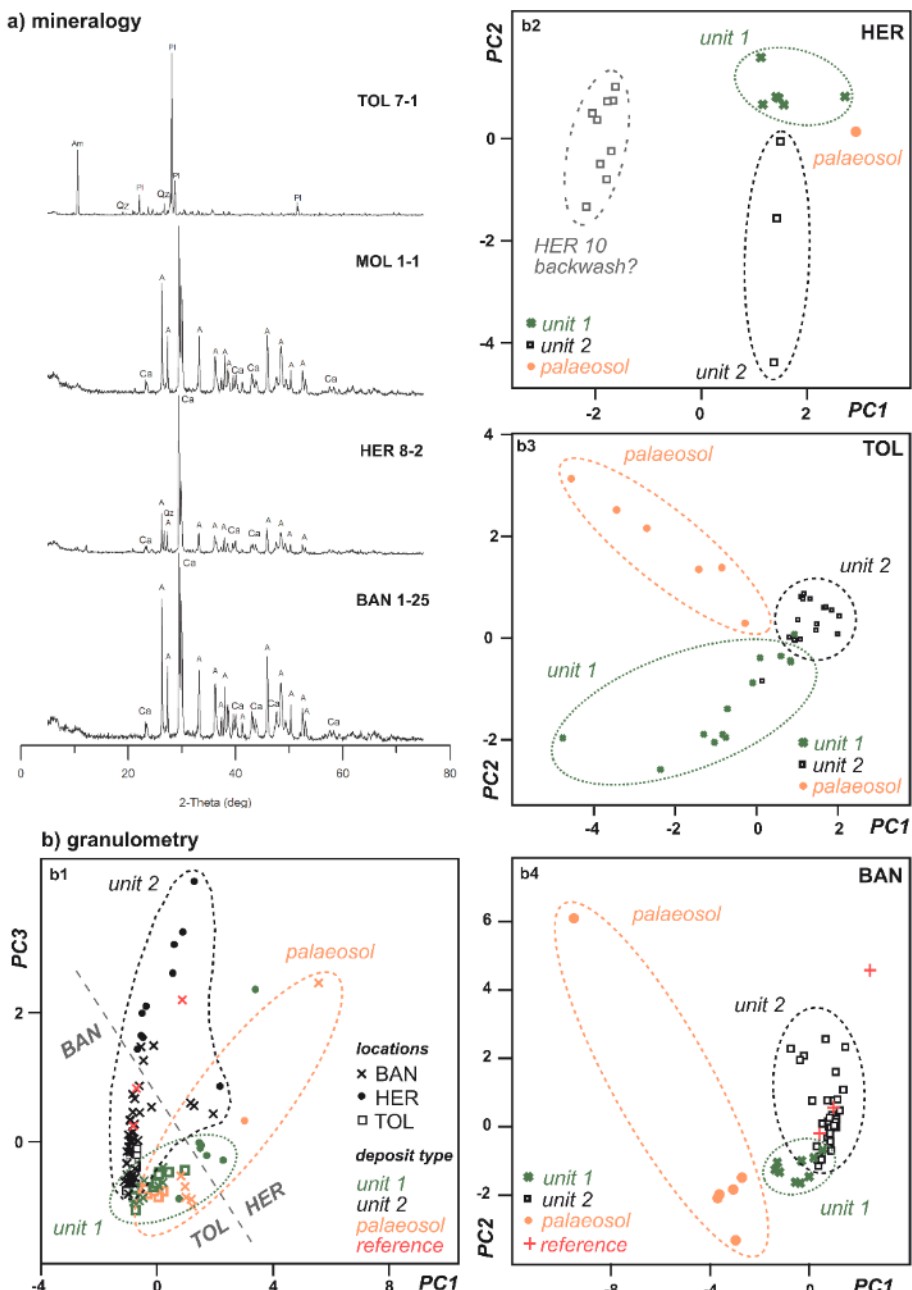

**Fig. 8: PCA results for sandy deposits of Typhoon Haiyan. a) XRD studies allow for a separation of deposits from carbonate coasts (MOL, HER, BAN) and the siliciclastic coast (TOL) on the basis of bulk-mineralogy. Ca = calcite, A = aragonite, Qz = quartz, Pl = plagioclase, Am = amphibole. b) Grain-size data reflect both site-specific characteristics (TOL, BAN, HER) and differences in formation (unit 1, unit 2, palaeosol) (b1). A discrimination of sediment formation (unit 1, unit 2, palaeosol) due to particular clusters is even more pronounced on the intra-site level (b2-4).**

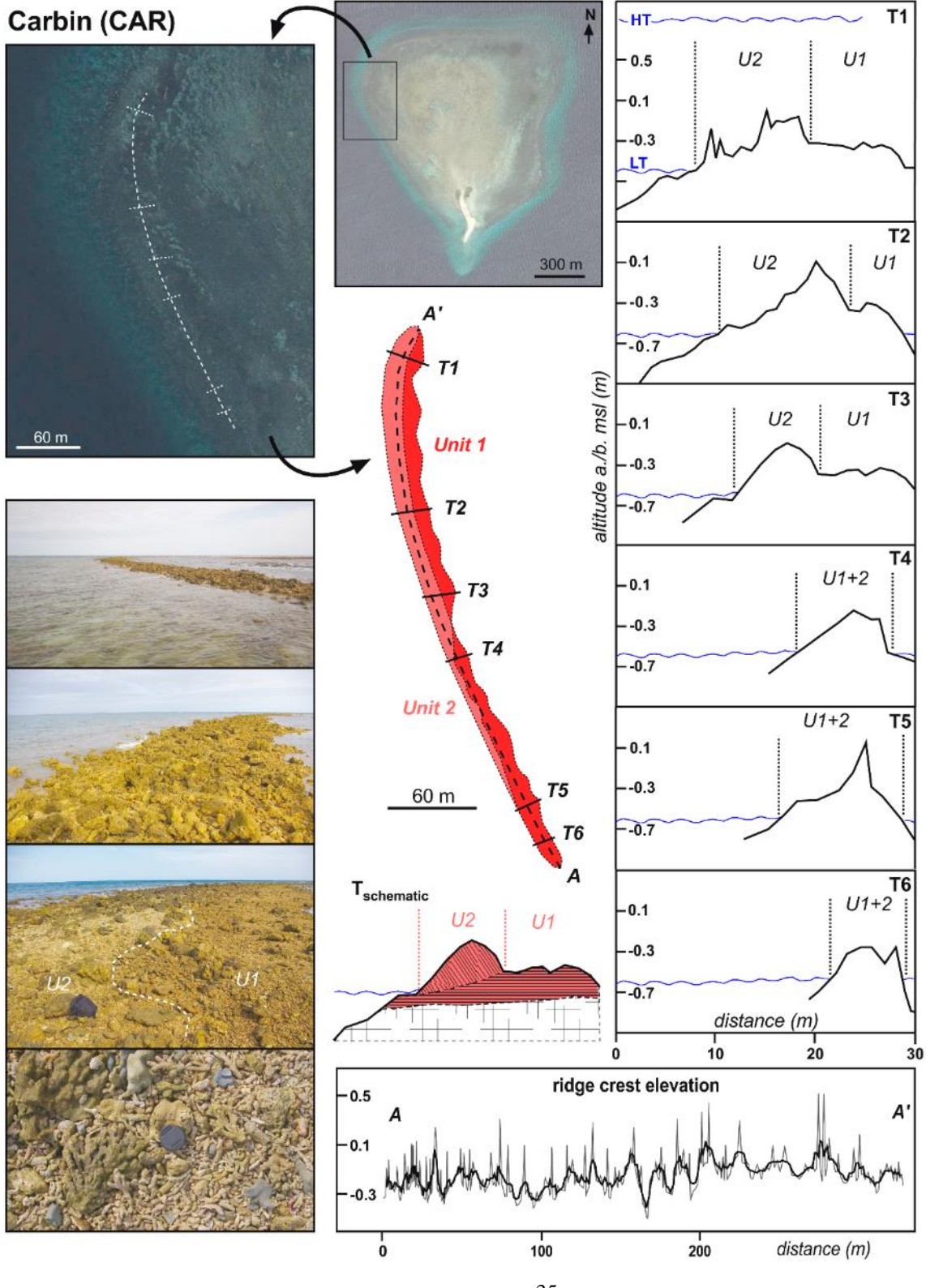

# Carbin (CAR)

60 m

300 m

A'

T1

Unit 1

T2

T3

T4

Unit 2

T5

T6

A

60 m

T_schematic

U2  U1

U2  U1

HT

T1

U2      U1

0.5
0.1
-0.3

LT

T2

U2      U1

0.1
-0.3
-0.7

T3

U2      U1

0.1
-0.3
-0.7

T4

U1+2

0.1
-0.3
-0.7

T5

U1+2

0.1
-0.3
-0.7

T6

U1+2

0.1
-0.3
-0.7

*altitude a./b. msl (m)*

*distance (m)*

0    10    20    30

ridge crest elevation

0.5
0.1
-0.3

A                                          A'

0          100          200    *distance (m)*

**Fig. 9: Carbin Reef (CAR) study site with typhoon-generated coral rubble ridge. The ridge at the western edge of the reef platform shows no clear trend of crest elevation, but significant differences in width and shape between six coast-perpendicular transects (T1– T6). In the northern part of the ridge, flat and up to 30 m wide lobes composed of algae-covered, greyish coral fragments form the basal unit of the ridge (U1). Unit 1 is topped by steeper lobes with a significant percentage of freshly broken coral fragments that occur along the whole ridge section but reach widths of only 10 m (U2). The presence of two morpho-stratigraphical units might be due to either changing wind directions during formation entirely by Haiyan, or due to the impact of successive storms where Haiyan only deposited the uppermost unit.**

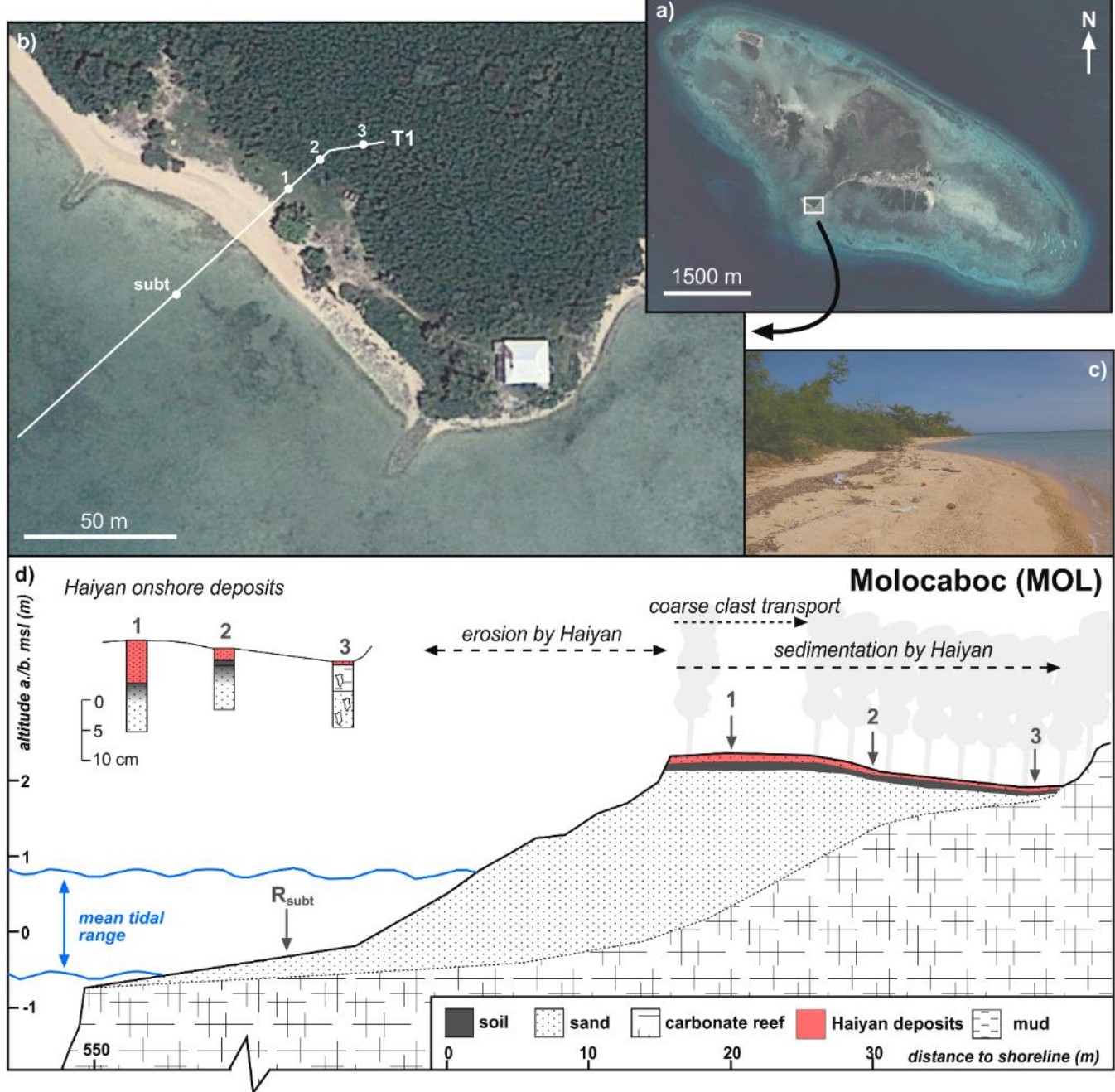

**Fig. 10: Molocaboc study site (MOL) with depositional and geomorphological effects of Typhoon Haiyan. a) Location of studied beach section on Molocaboc. b) Onshore sediments of Haiyan were investigated in three trenches (MOL 1–3) along a landward transect (T1). c) Beach profile after the Haiyan impact (view towards SE; date: 20 Feb 2014). d) Topographical cross section: the pre-Haiyan beach was substantially eroded and Haiyan generated a landward thinning sand sheet reaching up to 40 m inland in the back-barrier depression.**

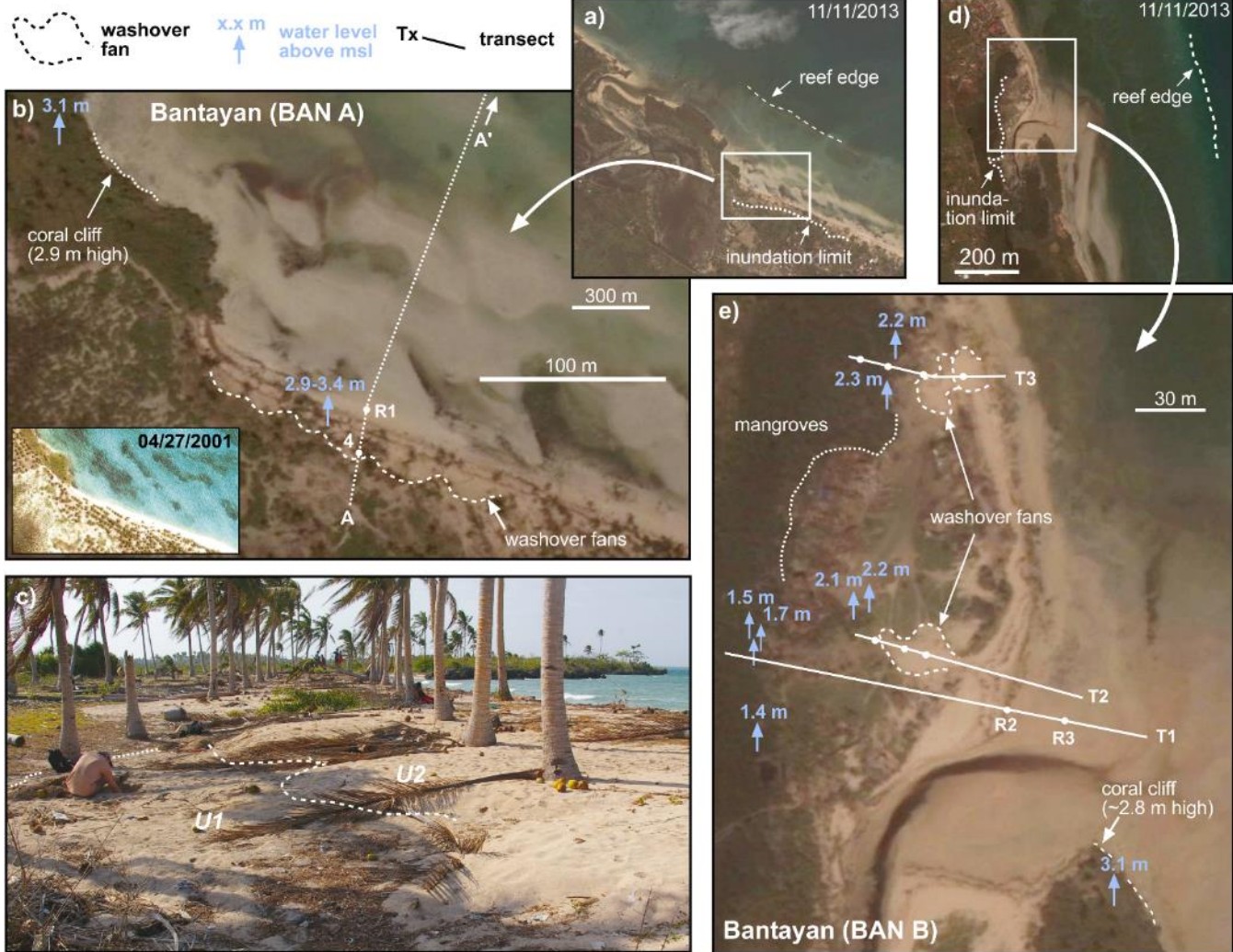

**Fig. 11: Bantayan study sites (BAN A and BAN B) with documented flood marks and geomorphological impact of Typhoon Haiyan. a–b) At BAN A, Haiyan flooded the proximal part of the back-barrier plain and formed a series of overlapping, small washover fans directly behind the beach ridge (based on Google Earth/Digital Globe 11/11/2013 [inset in b: 04/27/2001]). c) The washover fans reveal flat lobes at the base (U1), and steep lobes on top (U2). d–e) At BAN B, flooding levels of more than 3 m above msl generated several distinct washover fans (based on Google Earth/Digital Globe 11/11/2013).**

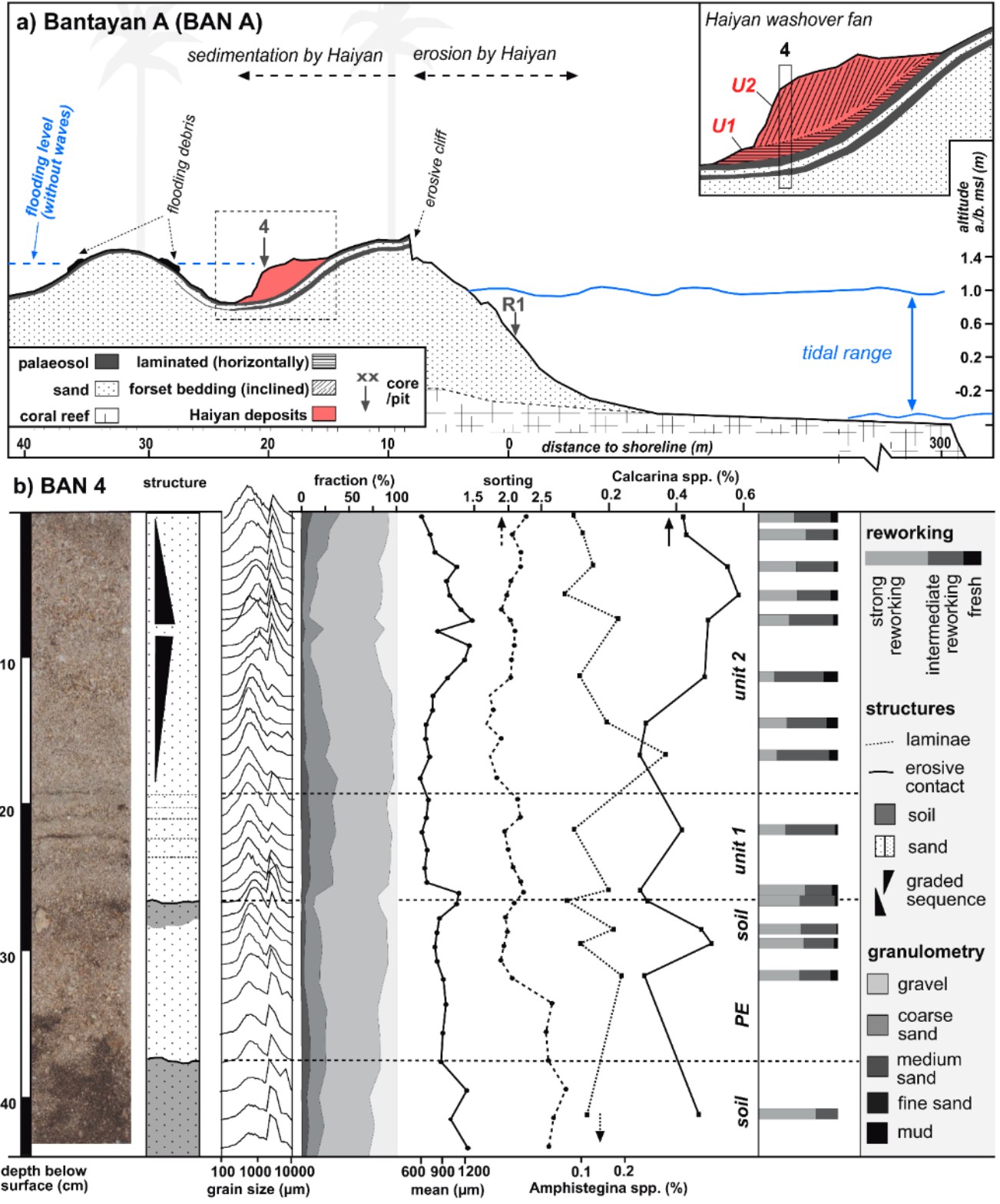

**Fig. 12: Typhoon Haiyan onshore deposits at site BAN A. a) A shore-perpendicular cross section (see Fig. 11 for location) reveals two sedimentary units within the washover fans, a flat and planar laminated unit at the base (U1), and steep, landward inclined lobes on top (U2). Below the pre-Haiyan soil, a second palaeosol is covered by a thin sand layer (PE), possibly a former extreme wave event. b) Sedimentary characteristics of core BAN 4. The 26 cm thick Haiyan deposits are clearly separated from older sediments by an initial soil formation. While the basal unit 1 is laminated, unit 2 is characterised by gradual coarsening, changing foraminifer composition (more *Calcarina* spp., less *Amphistegina* spp.) and a decreasing percentage of fresh foraminifer tests towards the top.**

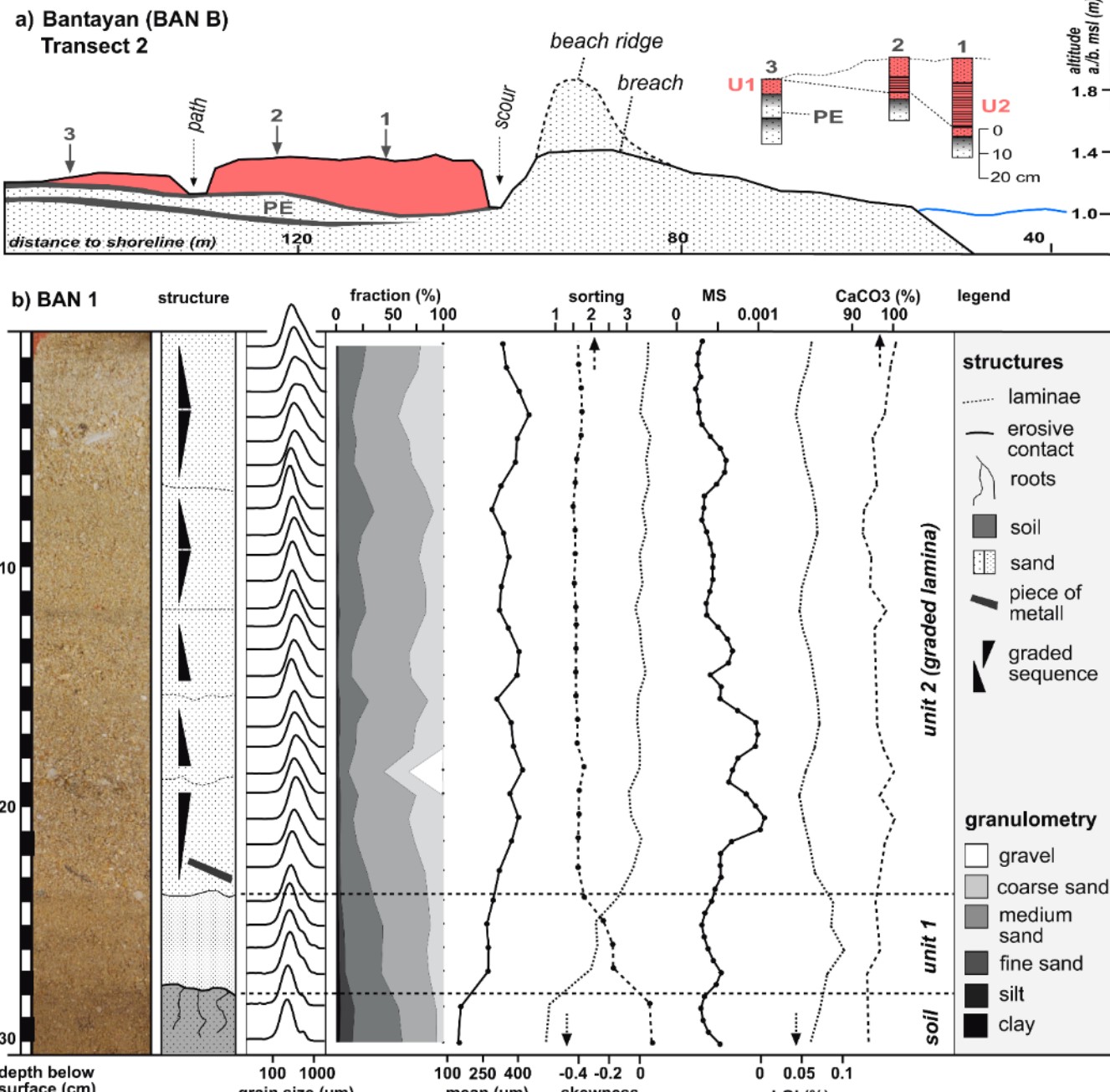

**Fig. 13: Sedimentary characteristics of Typhoon Haiyan deposits at site BAN B. a) Transect 2 (see Fig. 11 for location) reveals a landward thinning structure directly behind a storm-induced breach in the coastal barrier. A massive sand sheet at the base (U1) is overlain by a laminated section (U2) in the proximal part of the washover fan. b) BAN 1 is composed of two distinct units, a massive unit at the base (unit 1) and a laminated unit on top (unit 2).**

| | Coral ridges | Sand sheets | Washover fans |
|---|---|---|---|
| sedimentary structure | Coral rubble ridge on intertidal reef platform, 0.5-1.0 m high with two different units (at least uppermost due to Haiyan) | Normally graded to massive sand layer; <10 cm thick; thinning and fining in landward direction | Wedge-shaped sand lobes behind coastal barrier; up to 30 cm thick with planar and/or landward in-clined lamination |
| Locations | CAR (unit 1 + unit 2) | HER (unit 1)<br>TOL (unit 1)<br>BAN B (unit 1)<br>MOL (unit 1) | HER (unit 2)<br>TOL (unit 2)<br>BAN A (unit 1 + unit 2)<br>BAN B (unit 2) |
| Lateral extent | 10-20 m | 100-250 m | <50 m |
| Processes of transport & deposition | Storm-wave transport as traction load | Flooding pulse during early stage of the surge; mainly suspension load | Storm-wave trans-port during peak of the storm surge; mainly traction load |
| Sediment source | Mainly subtidal reef slope (coral rubble) with minor contribution from reef plat-form (fresh corals) | Mainly littoral zone; other source areas minor (poorly constrained) | Mainly littoral zone |
| Reference storm | Maragos et al., 1973; Scheffers et al., 2012 | Wang & Horwitz, 2007; Williams, 2010 | Williams, 2009; Sedgwick & Davis, 2003; Nott, 2006 |
| Reference tsunami | | Jankaew et al., 2008; Goto et al., 2011; Hawkes et al., 2007 | Atwater et al., 2013 |
| schematic sketch | CAR | TOL | TOL |

Haiyan deposit   source area   sediment in water column   deposition   erosion

Fig. 14: Compilation of sedimentary characteristics and inferred formation processes for the three different types of Haiyan deposits: coral ridges, sand sheets, and washover fans. The cited references report on storm or tsunami signatures with similar characteristics.