# Peer review of "Typhoon Haiyan's sedimentary record in coastal environments of the Philippines and its palaeotempestological implications"

_Natural Hazards and Earth System Sciences, 2016_

## Referee Comment (RC1) · P. Costa (Referee) · 19 Jul 2016

Dear Editor and authors, the work by Brill et al. presents an insight into the Typhoon Haiyan's sedimentary record in coastal environments of the Philippines and its palaeotempestological implications. I commend the authors on a well-written and interesting manuscript.

The authors addressed a topic with particular societal relevance due to the consequences of these catastrophic events for coastal areas. In this case, it is particularly relevant to say that the authors conducted extensive fieldwork and were also able to complement that with results derived from the application of some sedimentological proxies (grain-size, XRD and magnetic susceptibility). The data set gathered seems to

be solid and very interesting from a scientific point of view.

Overall, the manuscript has a clear structure and aims. However, in my opinion, several aspects should be addressed by the authors before the manuscript is accepted for publication, please see details below.

Although most of the issues I raise (please see below) are minor, I would like to stress that the authors need to be more consistent in terms of the vertical datum that they used. They should rewrite some of the numbered lists and make the text more easy to follow. They need to address more clearly the differences between tsunami and storm deposits and they should discuss transport modes and its implications for the depositional signature of the Typhoon Haiyan's sedimentary record in coastal environments of the Philippines (Jaffe et al., 2012 - Sed Geol). On top of this, they should stress that although local settings and sediment source are fundamental aspects that control storm deposit bed formation, there are a group of common features between the studied sites and that they share characteristics with deposits elsewhere (maybe adding a table summarizing these sedimentological features would help to the reader).

Other aspects: Abstract - the abstract is clear and well written. However, the authors need to clarify if they are studying 3 or 4 sites (they mention 4 sites here but mention 3 sites on page 3 line 10). I also suggest that the authors need to provide more comparisons with palaeotsunami data to sustain their sentence in line 20. In my opinion, that sentence should end "(...) typhoon signatures that can be used for palaeotempestological studies." the rest of the sentence should be deleted unless discussion is enriched with further topics on the comparison between tsunami and storm depositional signatures.

Introduction - Page 2, line 18, once we started talking about using geological record for several millenia we should also mention (and take in consideration) sea-level changes especially when we are using just a few specific study sites. - Page 2, line 25 - "naturaare" - spelling mistake - Page 2, line 27 - suggest you delete text up to line 30...

"Here, we report..." - Page 3 - I believe you should clarify or stress again the aims of your work, in particular at the end of the Introduction.

Study area - Page 3, line 10 - "three study areas"??? - Page 3, line 28 - when you refer to Samar please make reference to Figure or provide some clues about the specific location. - Page 4, line 5 - "three distinctive wave pulses" - Three sets of waves? How was this established? Was it measured? What was the Hs difference between the different pulses? Where any of these pulses related with infra-gravity waves? - Page 4, line 10 - i) and ii) and iii) - numbered lists were used intensively in this manuscript. I do not think they were used properly. Each numbered topic is very extensive and the reader is not guided properly. I suggest you rewrite all parts in the manuscript were you used numbered lists. Either you simplify the topics or you should write them as different sentences and start the sentences with "on the other hand" or "moreover" or etc. - Page 4, line 11 - "model-predicted". Throughout the manuscript you mention several times this but provide no details about modeled data. I strongly suggest you do that! Which model was used? What was the source data? What equations were used to calculate Hs, etc? etc, etc? - Page 4, line 13 - throughout the paper you refer to, at least 3, height (vertical datum) units (atl, msl, above mean low water and depth below surface). This makes it really hard for the reader. I strongly suggest you convert all to m above mean sea level! - Page 4, line 24 - please provide reference after "Philippine plate". - Page 4, line 29 - suggest you replace "originating" with "originated" and add "denser" to make the sentence ..."darker and denser minerals..." - Page 4, line 31 - Please see comment to page 4, line 10.

Methods - Page 5, line 12 - "along-shore perpendicular transects". So, cross-shore? What was the space between consecutive profiles? Did you created a DEM? - Page 5, line 16 - heights - Please see comment to page 4, line 13. - Page 5, line 21 - please replace "was" with "were". - Page 5, line 23 - this is a relevant aspect of the manuscript. Here, you suggest that in some locations you only used one core? Do you think this is enough for well supported interpretations? Especially, when later you

refer to all local specific conditions and lateral variations of the deposit!! - Page 5, line 13 - suggest you compare your approach with Quintela et al. (2016 - Quaternary International) methodology to identify allochthonous Foraminifera species within high-energy deposits.

Results - Page 6, lines 19, 26, 27, 28 - heights - Please see comment to page 4, line 13. - Page 6, line 23 - please refer to Figure 2 (?). - Page 7, line 3 - I believe you should provide/describe more grain-size data information. I suggest you add information on the D10, D90, sorting and unimodal or bimodal character of your samples. - Page 7, line 9 to 13 - I feel that in the discussion you should refer to the relationship between reworking and sediment concentration. Did you detected more reworked material in the basal sector of the storm layer or on the top? How was this correlated with grain-size? - Page 7, line 23 - again, the heights...what vertical datum did you used this time? - Page 8, line 2 - please refer to Figure. - Page 8, line 29 - I guess you should cite it as personal communication. - Page 8, line 30 - heights - Please see comment to page 4, line 13. - Page 9, line 16 - Rsubt was collected at approximately what depth? - page 10 - line 14 to 17 - the fact that the basal sector is slightly finer than the middle section is not just a consequence of the more erosive character of the initial stage of the event? The following phases benefited from a lowered coastal sector thus were capable of transporting coarser sediments farther inland. - Page 11, line 16 to 19 - this just reflects the dominance of the original (2nd cycle) sediment source. - Page 11, line 20 - I believe it is the first time you refer to principal components analysis, I suggest you refer to it in full. Line 12 - line 13 - this strongly suggests this area as the main sediment source.

Discussion - Page 12, line 27 - again the numbered list. - Page 12, line 30 - "normally graded or massive layers of sand". This implies totally different sediment transport modes (suspended grading and traction). I believe you should add a sentence here to comment on this and discuss reasons for the differences observed. - Page 13, line 3 - I suggest you add references from one of the several works conducted by Donnelly et

al. or Liu et al. in the eastern coast of the US. - Page 13, line 8 - I believe you should also mention infra-gravity waves. - Page 13, line 13 - very very interesting but why? Can you add a comment on this? - Page 13, line 16 - now it is important to know at what depth was your sample (Rsubt) retrieved!! - Page 13, line 25 to 29 - I suggest you rewrite this sentence. - Page 14 - line 1 - you must refer, for example, to the work of Komar and Wang (1984) or Komar (in Mange, 2007). - Page 14, line 6 to 9 - Agree with interpretation. - Page 14, line 18 to 20 - I accept your interpretation but I think formation of ridges implies a "continuum in time" more suitable with normal storm regime and a succession of events. - Page 14, line 21 - please see comment to page 4, line 10. - Page 14, line 27 and 28 - I think this partially contradicts statements above. I suggest you rewrite it. - Page 15, line 3 - please quantify the "remarkable amplification". - Page 15, line 4 - which models? - Page 15, line 8 to 14 - Reasoning perfectly reasonable - Page 15, line 15 - in fact, you can add that sediment source is always a decisive factor. - Page 15, line 24 to 27 - please rewrite this sentence. - Page 15, line 28 - is backwash really relevant for depositional imprints in storm events? Against gravity? - Page 16 - line 4 to 7 - here, you acknowledge that site-specific limits extrapolations of your conclusions. I agree and it really is hard to overcome this but, in my opinion, this field of science will progress will a multitude of sites, settings and events being studied. maybe you can add a sentence regarding future work. - Page 16, line 20 - you need to add a comment on the different settings studied by Hawkes et al. (2007) and Goto et al. (2011). - Page 16, line 20 to 27 - your conclusions are somewhat constrained because you did not compared tsunami and storm deposits in the same locations (e.g. Kortekaas and Dawson, 2007). - Page 16, line 28 to 31 - 2 units by one event is totally different from 2 units by more than one!! You need to discuss this!!

Conclusions - Page 17, line 2 - "local factors"... After so much work, it is important to stress the relevance of local conditions. In fact, I suggest you provide a geomorphological sketch (conceptual) model that describes accurately the initial pre-event conditions and the deposit after the event.

References - I suggest you add the above mentioned references.

Figures - The figures have very good quality, are well-designed and are informative.

Supplementary material - Useful.

I believe that in scientific terms the authors developed quality work that clearly deserves publication in NHESS, subject to very few minor changes. Regards Pedro J. M. Costa

---

## Author Comment (AC1) · 22 Aug 2016

Dear reviewer,

Thanks for the thorough and constructive comments to our submission. Integration of these suggestions will definitely help to further improve our manuscript. In the following we will address each of the comments separately (we attached a formated PDF-version of our reply as a supplement).

"Dear Editor and authors, the work by Brill et al. presents an insight into the Typhoon Haiyan's sedimentary record in coastal environments of the Philippines and its palaeotempestological implications. I commend the authors on a well-written and in-

teresting manuscript. The authors addressed a topic with particular societal relevance due to the consequences of these catastrophic events for coastal areas. In this case, it is particularly relevant to say that the authors conducted extensive fieldwork and were also able to complement that with results derived from the application of some sedimentological proxies (grain-size, XRD and magnetic susceptibility). The data set gathered seems to be solid and very interesting from a scientific point of view. Overall, the manuscript has a clear structure and aims. However, in my opinion, several aspects should be addressed by the authors before the manuscript is accepted for publication, please see details below. Although most of the issues I raise (please see below) are minor, I would like to stress that the authors need to be more consistent in terms of the vertical datum that they used. They should rewrite some of the numbered lists and make the text easier to follow."

**All height information will be presented in meters above mean sea level in the revised manuscript version. The respective sections will be rewritten for clarification (see also replies to specific comments below for more details).**

"They need to address more clearly the differences between tsunami and storm deposits and they should discuss transport modes and its implications for the depositional signature of the Typhoon Haiyan's sedimentary record in coastal environments of the Philippines (Jaffe et al., 2012 - Sed Geol)."

**We will add some more explanations concerning potential features that may allow to differentiate between tsunami and storm deposits in the discussion section of the revised version. We will provide sufficient details on potential tsunami and cyclone indicators (and the problematic that all of them are dependent on local setting) to allow the reader to follow our argumentation. However, in our opinion this discussion has already occurred in a large number of publications, and we therefore will also refer to the respective literature for more information (we will provide more references dealing with this topic) instead of going too much into detail.**

[Figure]

\# We also will enlarge the discussion of transport modes for the different storm signatures reported. In agreement with observations on the deposits of other tropical cyclones (e.g. Williams, 2009 at the US coast), we mainly attribute the formation of the slightly normally graded sand layers to settling from suspension during an early stage of the storm surge (at least at sites HER and TOL this initial flooding of the back-barrier areas was supported by infra-gravity waves). On the other hand, the laminated washover fans are interpreted to be the result of bedload transport over the coastal barrier, related to distinct storm waves during a later stage of the storm surge.

"On top of this, they should stress that although local settings and sediment source are fundamental aspects that control storm deposit bed formation, there are a group of common features between the studied sites and that they share characteristics with deposits elsewhere (maybe adding a table summarizing these sedimentological features would help to the reader)."

\# We will highlight this aspect in both the discussion section and the conclusions. This will also include a table listing the characteristics of the different storm features (including comparison with features of other storms and tsunamis) as suggested by the reviewer.

"The abstract is clear and well written. However, the authors need to clarify if they are studying 3 or 4 sites (they mention 4 sites here but mention 3 sites on page 3 line 10)."

\# We will align the information on the number of locations in the manuscript to 4 sites: Hernani (Samar), Tacloban (Leyte), Carbin/Molocaboc (Northern Negros), and Bantayan.

"I also suggest that the authors need to provide more comparisons with palaeotsunami data to sustain their sentence in line 20. In my opinion, that sentence should end "(...) typhoon signatures that can be used for palaeotempestological studies." the rest of the sentence should be deleted unless discussion is enriched with further topics on the comparison between tsunami and storm depositional signatures."

**We will slightly enlarge the discussion of tsunami signatures in the revised version of the manuscript (see reply to comment above) and, thus, will leave the sentence as is.**

"Page 2, line 18, once we started talking about using geological record for several millennia we should also mention (and take in consideration) sea-level changes especially when we are using just a few specific study sites."

**We absolutely agree with the reviewer that sea-level changes (and changes in palaeogeography) since the time prehistoric tsunamis or storms made landfall have to be considered when interpreting their geological records. We will add a short section to highlight the importance of this aspect.**

"Page 2, line 25 - "natura are" - spelling mistake"

**Will be corrected.**

"Page 2, line 27 - suggest you delete text up to line 30... "Here, we report...""

**We will delete the part of the mentioned section referring to a previous study on coastal boulders from the same area (May et al., 2015). However, we think the sentence concerning the potential relevance of the presented data for a general discrimination of tsunami and storm deposits is necessary and should not be deleted, since we believe that sediments of exceptional typhoons such as Haiyan definitely contribute to this discussion.**

"Page 3 - I believe you should clarify or stress again the aims of your work, in particular at the end of the Introduction."

**We will stress the aims of our study at the end of the introduction to make this aspect clearer for the reader. The revised section should read: "The major aims of this study are to (i) document Haiyan's impact on the sedimentology and geomorphology of heavily affected coastal areas by recording onshore and intertidal sedimentation, coastal erosion and geomorphological changes. Based on these data, sedimentary and geomorphological typhoon signatures typical for the study area shall be established. In**

addition, (ii) the spatial variability of these typhoon signatures due to site-specific characteristics such as the local topography, bathymetry, geology, and hydrodynamics as well as the exposure to the typhoon track shall be investigated. Finally, (iii) the potential of these modern typhoon deposits will be evaluated in respect of possible implications for the identification of prehistoric cyclones in the geological record."

"Page 3, line 10 - "three study areas"???"

**We will change to "four" to be in accordance with the information given earlier (see reply to first comment).**

"Page 3, line 28 - when you refer to Samar please make reference to Figure or provide some clues about the specific location."

**We will add a reference to Figure 1a, where the track of Typhoon Haiyan as well as the location of Samar are indicated.**

"Page 4, line 5 - "three distinctive wave pulses" - Three sets of waves? How was this established? Was it measured? What was the Hs difference between the different pulses? Where any of these pulses related with infra-gravity waves?"

**These three pulses of flooding with periods much longer than those of wind waves are based on eye-witness observations, so their heights (or the differences of heights between the pulses) are not well constrained. As mentioned on page 4, line 4, numerical models suggest that they may be the result of seiches (i.e. standing waves) in the semi-enclosed San Pedro Bay (Mori et al., 2014).**

"Page 4, line 10 - i) and ii) and iii) - numbered lists were used intensively in this manuscript. I do not think they were used properly. Each numbered topic is very extensive and the reader is not guided properly. I suggest you rewrite all parts in the manuscript were you used numbered lists. Either you simplify the topics or you should write them as different sentences and start the sentences with "on the other hand" or "moreover" or etc."

\# We will change the structure of this section according to the reviewer's suggestions.

"Page 4, line 11 - "model-predicted". Throughout the manuscript you mention several times this but provide no details about modeled data. I strongly suggest you do that! Which model was used? What was the source data? What equations were used to calculate Hs, etc? etc, etc?"

\# There are a number of different models that have been used. The data presented by Bricker et a. (2014), Mori et al. (2014), Cuadra et al. (2014), May et al. (2015b), Roeber and Bricker (2015), Kennedy et al. (2016), and Soria et al. (2016) are all based on models with different specifications. Although we agree that knowledge of models and parameters is important to evaluate the model output, providing all specifications in the manuscript would be a lengthy description that in our opinion would rather distract the reader. Since we refer to the original literature wherever we mention modelled data, interested readers can easily consult these articles for further information. While the detailed specifications of the models are in our opinion not required to understand the presented data, there are two main types of models that have been used to predict flooding levels, and which to discriminate is indeed important for the interpretation of our data: (1) numerical storm-surge models combined with phase-averaged wave models are routinely used to model surge heights for larger areas (Bricker et al., 2014; Cuadra et al., 2014; Mori et al., 2014; May et al., 2015; Soria et al., 2016); (2) numerical surge models with phase-resolved (boussinesq-type) wave models are required to reproduce the interactions of waves with the local topography, which may generate infra-gravity waves (Roeber and Bricker, 2015; Kennedy et al., 2016). So while we think it is sufficient to refer to the original literature for detailed model specifications, we will explicitly include the discrimination of phase-averaged and phase-resolved wave models in the revised version of the manuscript.

"Page 4, line 13 - throughout the paper you refer to, at least 3, height (vertical datum) units (atl, msl, above mean low water and depth below surface). This makes it really hard for the reader. I strongly suggest you convert all to m above mean sea level!"

\# We agree with the reviewer that the use of different height levels might be confusing for the reader. To allow for comparability between all sites, we will provide all height references for data presented in this research (topography, sediments, flood marks, etc.) in meters above or below mean sea level (above/below msl). However, since the same values relative to mean sea level may – depending on the elevation of the ground – have completely different implications for sedimentation, we will also provide the flow depth in meters above ground level in case of the measured flood marks. For describing the stratigraphies of sediment profiles, we will stick to meters below surface, since here a relation to sea level would be rather confusing. We will state this information explicitly in the methods section of the revised version.

"Page 4, line 24 - please provide reference after "Philippine plate"."

\# We will add Rangin et al. (1989) as a reference.

"Page 4, line 29 - suggest you replace "originating" with "originated" and add "denser" to make the sentence ..."darker and denser minerals...""

\# The sentence will be changed accordingly.

"Page 4, line 31 - Please see comment to page 4, line 10."

\# The structure of this sections will be changed according to the reviewer's suggestions.

"Page 5, line 12 - "along-shore perpendicular transects". So, cross-shore? What was the space between consecutive profiles? Did you created a DEM?"

\# Due to the limited time available at each study location during the survey, we measured only a single transect at most of the sites. Only on Carbin Reef (6 transects) and at BAN B (3 transects) several transects were measured (all of them are documented in the respective figures). Consequently, no DEMs were created as well.

"Page 5, line 16 - heights - Please see comment to page 4, line 13."

\# As already mentioned in our reply to page 4, line 13, all heights will be provided in

meters above mean sea level. Flow depth (so meters above ground surface) will be provided additionally in case of flood marks.

"Page 5, line 21 - please replace "was" with "were"."

**Will be corrected.**

"Page 5, line 23 - this is a relevant aspect of the manuscript. Here, you suggest that in some locations you only used one core? Do you think this is enough for well supported interpretations? Especially, when later you refer to all local specific conditions and lateral variations of the deposit!!"

**We indeed analyzed only a single sediment core for site BAN A. For all other locations (HER, TOL, BAN B), several samples collected at different distances to the shoreline were analyzed. We agree that there are lateral variations in terms of granulometry and faunal composition at the individual sites, which are of course not covered by the single core at BAN A. However, at site BAN A the lateral extension of the deposit is only ∼10-20 meters. We checked the lateral structure by means of trenches and the section sampled by BAN 4 is assumed to be representative for the entire washover fan. Especially for the comparison of BAN 4 with other sites the lack of lateral data should be negligible, since the differences between sediments from different sites are much more pronounced. Although some limitations must be expected for granulometry and faunal composition that vary laterally, we therefore assume that the results of this single core (i) represent the typical sediment composition at BAN A quite well, and (ii) can already document the main differences compared to the other locations.**

"Page 6, line 13 - suggest you compare your approach with Quintela et al. (2016 – Quaternary International) methodology to identify allochthonous Foraminifera species within high energy deposits."

**The methodological aspects of foraminifer determination, counting and taphonomy classification used in our study should be similar to those applied by Quintela et al.**

(2016). Particularly the argument that higher percentages of broken foraminifer tests are the result of high grain density in the traction-transport dominated parts of high energy flows may be of importance for discussing our foraminifer assemblages, and will be considered in the discussion section of the revised version.

"Page 6, lines 19, 26, 27, 28 - heights - Please see comment to page 4, line 13."

**All heights will be provided in meters above mean sea level and (additionally) as flow depth above surface.**

"Page 6, line 23 - please refer to Figure 2 (?)."

**A reference to figure 2 will be added.**

"Page 7, line 3 - I believe you should provide/describe more grain-size data information. I suggest you add information on the D10, D90, sorting and unimodal or bimodal character of your samples."

**We will complement the grain-size information for all sites, and provide data on mean, sorting, and modality for each site.**

"Page 7, line 9 to 13 - I feel that in the discussion you should refer to the relationship between reworking and sediment concentration. Did you detected more reworked material in the basal sector of the storm layer or on the top? How was this correlated with grain-size?"

**In case of HER 10, no clear vertical trend in foraminifer taphonomy or species composition could be detected. There is rather a slight correlation between coarser grain size and stronger reworking (Fig. 4). This is similar for core BAN 4, where strong reworking correlates with larger grain size as well (Fig. 12). However, since sediments first tend to become coarser towards the top of BAN 4 and fine afterwards, reworking is highest in the central section of BAN 4.**

"Page 7, line 23 - again, the heights...what vertical datum did you used this time?"

**All heights are provided in meters above mean sea level and, in case of flood marks, (additionally) as flow depth above surface.**

"Page 8, line 2 - please refer to Figure."

**We will insert a reference to figure 6.**

"Page 8, line 29 - I guess you should cite it as personal communication."

**We will cite the observation as "personal communication".**

"Page 8, line 30 - heights - Please see comment to page 4, line 13."

**The heights are provided in meters above/below mean sea level.**

"Page 9, line 16 - Rsubt was collected at approximately what depth?"

**The sample was collected at 0.5 m below mean sea level. We will add this information in the revised manuscript.**

"Page 10 - line 14 to 17 - the fact that the basal sector is slightly finer than the middle section is not just a consequence of the more erosive character of the initial stage of the event? The following phases benefited from a lowered coastal sector thus were capable of transporting coarser sediments farther inland."

**The proposed mechanism is a very plausible explanation for the observed stratigraphical pattern at this location, because both units are assumed to be deposited by similar processes, i.e. wave swash overtopping the coastal barrier. We will briefly address this aspect in the discussion section of the revised version.**

"Page 11, line 16 to 19 – this just reflects the dominance of the original (2nd cycle) sediment source."

**We agree that the mineralogy and geochemistry mainly indicate the differences between limestone and volcanic environments. We already address this topic in the discussion section (page 15, lines 15 ff in the original version of the manuscript).**

"Page 11, line 20 - I believe it is the first time you refer to principal components analysis, I suggest you refer to it in full."

\# The abbreviation PCA is already mentioned in the methods section. However, we agree that referring to it in full at this position might facilitate reading.

"Line 12 - line 13 - this strongly suggests this area as the main sediment source."

\# That is how we interpret this data in the discussion section as well. To make this implications already clear in section 4.5, we will add a brief explanation.

"Page 12, line 27 - again the numbered list."

\# We will remove the numbering to facilitate reading.

"Page 12, line 30 - "normally graded or massive layers of sand". This implies totally different sediment transport modes (suspended grading and traction). I believe you should add a sentence here to comment on this and discuss reasons for the differences observed."

\# This should read "normally graded to massive layers of sand". While slight normal grading could be detected for thicker layers (close to the coast) and especially those analyzed for vertical grain-size variations in the laboratory, the small thickness of the sand layers further inland did not allow for unambiguous identification of grading. We assume that even the thinner parts of the sand sheets might be normally graded. But since we cannot prove this (macroscopically their structure could be both massive and slightly graded), we prefer to describe them as "normally graded to massive".

"Page 13, line 3 - I suggest you add references from one of the several works conducted by Donnelly et al. or Liu et al. in the eastern coast of the US."

\# We will add Donnelly et al. (2006) as a reference from the US coast.

"Page 13, line 8 - I believe you should also mention infra-gravity waves."

\# Actually, the mentioned "long-wave phenomena" already include infra-gravity waves that can result e.g. from surf beat. To make this clearer, we will explicitly use the term "infra-gravity waves" at this position.

"Page 13, line 13 - very very interesting but why? Can you add a comment on this?"

\# The deposits described by Williams (2009) have actually a very similar structure as those described on the Philippines: a finer, graded sand layer formed during the initial inundation of the back-barrier plains, topped by washover deposits during a later stage of the storm surge. We therefore assume similar transport modes for our deposits, i.e. suspension settling for the graded sand sheet and bedload/traction for the formation of the washover lobes. The role of long-wave phenomena for the deposits presented in our study (infra-gravity waves at HER and seiches at TOL) is probably a contribution to higher and more extensive flood levels, but not significantly different sedimentation processes.

"Page 13, line 16 - now it is important to know at what depth was your sample (Rsubt) retrieved!!"

\# As mentioned before, the sample was collected at 0.5 m below mean sea level. It contrasts significantly in terms of faunal composition from the storm deposits, while the littoral reference samples from BAN reveal a similar granulometry and faunal composition with the typhoon deposits.

"Page 13, line 25 to 29 - I suggest you rewrite this sentence."

\# We will change the structure of this section to make it clearer for the reader.

"Page 14 - line 1 - you must refer, for example, to the work of Komar and Wang (1984) or Komar (in Mange, 2007)."

\# We will add Komar and Wang (1984) as a reference for density sorting.

"Page 14, line 18 to 20 - I accept your interpretation but I think formation of ridges

implies a "continuum in time" more suitable with normal storm regime and a succession of events."

**We agree that ridges might form during several successive events rather than single storms. In fact, we state the possibility of ridge formation by several typhoons (with significant growth of a pre-existing ridge during Haiyan) further down in this section.**

"Page 14, line 21 - please see comment to page 4, line 10."

**We will remove the numbering.**

"Page 14, line 27 and 28 - I think this partially contradicts statements above. I suggest you rewrite it"

**Since our evidence is not unambiguous without robust age data, we have to provide both possible explanations for the ridge formation. Of course these explanations partially contradict each other, because only one of them can reflect reality. We will rewrite the section to clarify this aspect.**

"Page 15, line 3 - please quantify the "remarkable amplification"."

**In the central part of the bay, water levels of more than 8 m above sea level were recorded, which is much higher than in Haiyan-affected areas not subject to infragravity waves (HER) or raised water levels related to shore configuration (TOL). We will add the value in the revised manuscript.**

"Page 15, line 4 - which models?"

**Here we refer to storm surge models combined with phase-averaged wave models (for details we refer to the original reference by Bricker et al., 2014) that do not account for the effects of infra-gravity waves (phase-resolved wave models). We will add this information in the revised version.**

"Page 15, line 15 - in fact, you can add that sediment source is always a decisive factor."

[Figure]

**We explicitly will add sediment source as a further decisive factor.**

"Page 15, line 24 to 27 - please rewrite this sentence."

**The sentence will be rewritten.**

"Page 15, line 28 – is backwash really relevant for depositional imprints in storm events? Against gravity?"

**Usually backwash is probably of minor or no importance during storms. However, sample HER 10 was collected close to a fluvial channel, where the backwash was not against gravity.**

"Page 16 - line 4 to 7 - here, you acknowledge that site-specific limits extrapolations of your conclusions. I agree and it really is hard to overcome this but, in my opinion, this field of science will progress will a multitude of sites, settings and events being studied. Maybe you can add a sentence regarding future work."

**We agree that the value of case studies is their contribution to the database of locally and regionally differences of storm and tsunami deposits. We already tried to address this aspect later in this section. However, we will modify our statement to highlight this message.**

"Page 16, line 20 - you need to add a comment on the different settings studied by Hawkes et al. (2007) and Goto et al. (2011)."

**The settings mentioned here are wide coastal plains or beach-ridge plains with a low topography that does not hinder lateral inundation and sediment transport due to steep slopes. Indeed, this is not true for the sites investigated by Hawkes et al. (2007), so we replaced the reference with observations on 2004 Tsunami deposits from a beach-ridge plain in Thailand by Jankaew et al. (2008). While most of the sites presented here have a steeper topography and are therefore not directly comparable, similar conditions are given at TOL. Nevertheless, in spite of high surge levels >5m, sand transport is limited to not more than ∼300 m (although the topography of the coastal plain would**

allow for much more extensive deposition).

"Page 16, line 20 to 27 - your conclusions are somewhat constrained because you did not compared tsunami and storm deposits in the same locations (e.g. Kortekaas and Dawson, 2007)."

**We agree that the conclusions are limited due to this fact. However, by comparing with tsunami deposits from sites with similar settings (similar flood levels and similar topography), we think our findings nevertheless add to the observation that sediment extent tends to be a discriminative feature.**

"Page 16, line 28 to 31 - 2 units by one event is totally different from 2 units by more than one!! You need to discuss this!!"

**We agree that both interpretations would have completely different implications. But since we are not able to prove one of the two possibilities (the formation of the washover fans at TOL could neither be proved by eyewitnesses nor by satellite images), we have to present both options for this location.**

"Conclusions - Page 17, line 2 - "local factors"... After so much work, it is important to stress the relevance of local conditions. In fact, I suggest you provide a geomorphological sketch (conceptual) model that describes accurately the initial pre-event conditions and the deposit after the event."

**The idea to present the main outcomes of the study in a conceptual figure is indeed reasonable. We decided to merge this figure with the table summarizing the characteristics of the storm features presented in this paper (see reply to an earlier comment). This figure will include schematic sketches for the formation of each storm feature (we attached a figure scetch at the end of our reply). It will, however, not provide a separate figure on the pre-event situation. This is poorly constrained and, thus, cannot be documented with sufficient detail.**

"I suggest you add the above mentioned references."

**The mentioned references will be included.**

"I believe that in scientific terms the authors developed quality work that clearly deserves publication in NHESS, subject to very few minor changes. Regards Pedro J. M. Costa"

Please also note the supplement to this comment:
http://www.nat-hazards-earth-syst-sci-discuss.net/nhess-2016-224/nhess-2016-224-AC1-supplement.pdf
* * *
| | Coral ridges | Sand sheets | Washover fans |
|---|---|---|---|
| **sedimentary structure** | Coral rubble ridge on intertidal reef platform, 0.5-1.0 m high with two different units (at least uppermost due to Haiyan) | Normally graded to massive sand layer; <10 cm thick; thinning and fining in landward direction | Wedge-shaped sand lobes behind coastal barrier; up to 30 cm thick with planar and/or landward inclined lamination |
| **Locations** | CAR (unit 1 + unit 2) | HER (unit 1)
TOL (unit 1)
BAN B (unit 1)
MOL (unit 1) | HER (unit 2)
TOL (unit 2)
BAN A (unit 1 + unit 2)
BAN B (unit 2) |
| **Lateral extent** | 10-20 m | 100-300 m | <50 m |
| **Processes of transport & deposition** | Storm-wave transport as traction load | Flooding pulse during early stage of the surge; mainly suspension load | Storm-wave trans-port during peak of the storm surge; mainly traction load |
| **Sediment source** | Mainly subtidal reef slope (coral rubble) with minor contribution from reef plat-form (fresh corals) | Mainly littoral zone; other source areas minor (poorly constrained) | Mainly littoral zone |
| **Reference storm** | Maragos et al., 1973; Scheffers et al., 2012; | Wang & Horwitz, 2007; Williams, 2010 | Williams, 2009; Sedgwick & Davis, 2003; Nott, 2006 |
| **Reference tsunami** | | Jankaew et al., 2008; Goto et al., 2011; Hawkes et al., 2007 | Atwater et al., 2013 |
| **schematic sketch** | CAR | TOL | TOL |

Legend: ▯ Haiyan deposit  ▨ source area  ⋮ sediment in water column  ⤺ deposition  ⇑ erosion

**Fig. 1.** summary figure

---

## Referee Comment (RC2) · B. Jaffe (Referee) · 27 Sep 2016

Dear Editor and authors,

The research presented in this paper is a valuable contribution to the documentation of the sedimentary record of storms. Documenting the sedimentary record of Haiyan is critical because it is a very large storm and there is a need for data on erosion and deposition for extreme storms. The scope of the field investigation is impressive and the wide variety of laboratory analyses performed create a large data set that can be used to discriminate storm deposits from tsunami and other high-energy deposits. The figures are informative and well done. I recommend that this paper be published after revision. I suggest possible ways to improve the paper below.

[Figure]

Perhaps because of the amount of data presented I found the paper hard to follow. I suggest two things to help the reader: (1) a table summarizing all the sites visited and their important characteristics [source sediment available for transport, topography, etc.] and Haiyan deposit metrics [inland extent, maximum thickness, number of layers, grain size, etc.], and (2) adding text in the introduction, or in a new section, about published reports on geometries and thickness/grain size trends for storm and tsunami deposits to give context for Haiyan deposits.

I am not entirely sure, but it appears that all the figures use distance along transect rather than distance from the shoreline. This is supported by the text on Page 6, Line 24, "shallow reef outcrops occur at 180 m transect length." I also measured the distance along transect at Tolosa (Fig. 5). TOL14 is about 120 m inland from the shoreline, but is plotted at about 145 m in Figure 3 and 5. I measured the distance along transect at Tolosa to be about 140 m. Do all the figures use distance along transect rather than distance from the shoreline? If so, they need to be corrected. Distance along transect is meaningless because a change in transect orientation results in a different distance along transect for the same distance from the shoreline, the physically meaningful parameter.

I disagree with the statement on Page 1, Line 20 in the Abstract that Haiyan deposits, "might also function as a benchmark example for a general discrimination between storm and tsunami deposits." The Haiyan deposits are part of a spectrum of possible storm deposits; however, because of the presence of surf beat creating a tsunami-like bore they may not be typical. If so, although valuable to illustrate the spectrum of possible storm deposits, they may be more atypical than typical and therefore not a "benchmark example for a general discrimination between storm and tsunami deposits."

A discussion of preservation potential of the Haiyan deposits would provide insight into whether the deposits observed in this study would be found in the geologic record.

Instead of using "geological imprint" in the first sentence of the Conclusions (Page 17, Line 2) use "deposits" because preservation of the deposits is not addressed. Preservation of storm deposits in environments investigated in this study is a rather large topic and worthy of another paper. But, although it might seem like semantics, the "geological imprint" is unknown at this time because whether the deposit will be preserved is unknown and how it will be altered as time passes is also unknown.

Same comment about using "geological legacy" in the next sentence (Page 17, Line 3). I suggest using "deposits" again.

Also in the conclusion is the statement, "the sandy onshore deposits left by Haiyan are very similar to those generated by tsunamis." Rather than give a qualitative qualifier of "very similar", which means different things to different people, list the similarities.

In the Abstract (Page 1, Line 3) and Conclusions (Page 17, Line 4) it would help the reader if you clarified what "Extended onshore sand sheets"/"extensive sand sheets" mean. Is there an inland distance that a sand sheet extends inland that you classify as "extended"/"extensive"? Perhaps it would be better to specify how far inland the sand sheets extend. How readers define extended/extensive will vary and it is better to be specific.

Shell fragments are present in the Haiyan deposit at some locations. Grain size is measured by laser diffraction and Camsizer and does not account for particle density or shape, both of which would be quite different for shells than other components and affect their settling velocity and transport in suspension. I suggest discussing how the presence of shells affect your interpretation of the grain size data. Woodruff et al. (2008) address the differences between settling velocities of shells and siliciclastic particles. Figure DR2 in the data repository summarizes their results. The citation for Woodruff et al. (2008) is: Woodruff, J.D., Donnelly, J.P., Mohrig, D., Geyer, W.R., 2008. Reconstructing relative flooding intensities responsible for hurricane-induced deposits from Laguna Playa Grande, Vieques, Puerto Rico. Geology 36, 391–394.

http://dx.doi.org/10.1130/G24731A.1.

A fining trend in modal grain size is reported for Hernani (page 7, lines 2 and 3; Figure 3). However, the mean grain size trends of Hernani are more complicated. If trends in modal grain size are reported, please discuss how mean grain size trends are different and justify why you assign a "fining landward" trend based on modal grain size.

Are the statistics in Figure 3 for grain size for the entire deposit? That is, are they averages for all the grain size data for deposit? Please clarify for the reader.

Please explain further how it was determined that sediment from the foreshore and deeper water are part of the Haiyan deposit (Page 132, Line 16). Are there grain sizes present in the Haiyan deposit that are not from the beach? Can this be sediment picked-up landward of the beach? Were there samples collected from the foreshore and nearshore close in time to when Haiyan impacted the Phillipines that have grain size data?

What is meant by "a rather normally graded structure of these sand sheets" (Page 13, Line 9)? This is important because grading of deposits may be a discriminator of storm versus tsunami deposition. Were the Haiyan deposit suspension graded, as has been observed for deposits formed by several recent tsunamis and for paleotsunami deposits? (for an explanation of suspension grading see: Jaffe, B.E., Buckley, M.L., Richmond, B.M., Strotz, L., Etienne, S., Clark, K., and Gelfenbaum, G., 2011, Flow speed estimated by inverse modeling of sandy sediment deposited by the 29 September 2009 tsunami near Satitoa, east Upolu, Samoa, Earth-Science Reviews, v. 107, p. 23-37, doi:10.1016/j.earscirev.2011.03.009.)

Missing reference for Haiyan surf beat: Roeber, V and Bricker, J., 2015, Destructive tsunami-like wave generated by surf beat over a coral reef during Typhoon Haiyan, Nature Communications (6), DOI: 10.1038/ncomms8854

Other comments:

Page 2, Line 8- Suggest changing "coastal disasters in the immediate past" to "recent coastal disasters".

Page 2, Line 15- Suggest ending the sentence after "records" and change the next part of the sentence to a new sentence "This discrepancy is great because cyclones usually follow an inverse power law (Corral et al., 2010).

Page 2, Line 18- Suggest changing "even events of the highest magnitudes" to "large events".

Page 2, Line 25- Suggest changing "naturaare particularly" to "are".

Page 3, Line 16- Suggest changing "the significance of seasonality" to "seasonal variability".

Page 4, Lines 10-18 and later in the paper as well- (i), (ii), (iii) are not needed and are distracting.

Page 4, Line 31- Suggest changing "typically shows" to "is characterized by".

Page 4, Line 31- Again, (i), (ii), (iii) are not needed in this paragraph.

Page 4, Line 32- The fetch over the Pacific, not the narrow shelf, is the reason that Eastern Samar has high swell waves.

Page 5, Lines 13 and 14- The times of day for the DGSP are not relevant and should be omitted.

Page 5, Line 15- Suggest changing "were recorded by leveling" to "were documented by measuring elevations of".

Page 5, Line 20- Suggest deleting "directly".

Page 5, Line 23- Define what you mean by representative. Typical thickness? Typical sediment grain size? Typical structure?

Page 5, Line 27- Chemical formula contain subscripts for the number of atoms for

elements.

Page 6, Line 17- Is Barangay capitalized?

Page 6, Line 19- Suggest changing "the two tropical storms/depressions recorded between Haiyan and this field survey on January 19th and February 1st respectively" to "the two tropical storms/depressions hitting the Philippines on January 19th and February 1st, respectively, which is after Haiyan and before this field survey".

Page 7, Line 6- Are the values for grain size in ")"thinning and fining landward from 8 cm and a mean of 570 $\mu$m at 130 m from the shoreline (HER 8) to only 3 mm and a mean of 223 $\mu$m at 260 m (HER 3) (Fig. 3)." for the mode or mean? It appears from Figure 3 that they are for the mode, but I am not sure because this for Unit 1 and it is not clear what is shown in Figure 3.

Page 7, Line 21- Delete "According to".

Page 7, Line 22- Suggest changing "bushes, Haiyan" to "are evidence that Haiyan".

Page 7, Line 30- Suggest changing "Pre and post-typhoon" to "Pre- and post-typhoon".

Page 8, Line 9 and later in the text- The use of the word "profiles" is not standard. Suggest changing "profiles TOL 7-14" to "trenches TOL 7-14". This suggestion applies everywhere in the text where "profile" is used to describe a trench.

Page 8, Line 15- Specify what "slightly inclined" means.

Page 8, Line 21- What is meant by "moderate flooding". I have no idea what is moderate. Please be specific by giving an spatial extent and/or a water depth.

Page 8, Line 25- How thin are the sand patches?

Page 9, Line 1- Delete "single". It is not needed.

Page 9, Line 1- Suggest changing "in either direction" to "crest elevation".

Page 12, Line 30- Suggest changing "show comparably large inland extents exceeding

100 m" to "extend at least 100 m inland".

Page 13, Line 13- Suggest changing "dedicated" to "attributed".

Page 16, Line 21- Suggest changing "confined" to "indicated".

Page 25, Figure 3- Be consistent with line types in each panel. Sorting is a different line type for HER 3-9 than for TOL 3-14 and BAN 1-3. A minor point, but why not make it easier on the reader to compare panels? Also, a solid line is used for both the mode and mean in different panels. Why not use a solid line for the mean and another line type for the mode?

Page 25, Figure 3- Why does the thickness scale for TOL 3-14 start at -2? This makes it difficult to determine the thickness of the more landward deposts. Whay not start the scale at 0 to make it easy to determine the thickness of landward deposits?

Page 25, Figure 3- Why is there a vertical dashed line at 40 m in the TOL 3-14 panel? Please explain this line in the caption.

Page 25 Figure 3 caption- The mean grain size of HER 3-8 doesn't monotonically fine landward. See earlier comment on mode versus mean and description/definition of landward fining.

Page 29, Figure 6- For consistency, add the column that indicates grading by the shaded triangles.

Page 29, Figure 6 caption- The transect is not coast-perpindicular.

Page 33, Figure 10 caption- The transect is only shore-perpindicular fpr tremcjes 1 and 2, not trench 3.
* * *
[Figure]

[Figure]

**Figure DR2: Grain size versus settling velocity for LPG sediment.** Mean settling velocities ($w_s$) measured for siliciclastics (black circles) and shell material (gray circles). Vertical error bars indicate 1σ range for $w_s$ and horizontal error bars indicate ranges of grain diameters in each bin size. Comparisons between the actual mean settling velocities measured for LPG siliciclastics sediments and values predicted by Ferguson and Church (2004) for naturally shaped quartz sands (black dotted line) reveal an excellent fit and support using the relationship for analyses in this study.

**Fig. 1.** Figure showing difference in settling velocity of shells and siliciclastic material (Woodruff et al., 2008)

---

## Author Comment (AC2) · 16 Oct 2016

Dear reviewer,

Thanks for the thorough and constructive comments to our submission. We think integration of these suggestions will help to further improve our manuscript. In the following we will address each of the comments separately.

Dear Editor and authors, The research presented in this paper is a valuable contribution to the documentation of the sedimentary record of storms. Documenting the sedimentary record of Haiyan is critical because it is a very large storm and there is a need for data on erosion and deposition for extreme storms. The scope of the field in-

vestigation is impressive and the wide variety of laboratory analyses performed create a large data set that can be used to discriminate storm deposits from tsunami and other high-energy deposits. The figures are informative and well done. I recommend that this paper be published after revision. I suggest possible ways to improve the paper below.

Main comments:

Perhaps because of the amount of data presented I found the paper hard to follow. I suggest two things to help the reader: (1) a table summarizing all the sites visited and their important characteristics [source sediment available for transport, topography, etc.] and Haiyan deposit metrics [inland extent, maximum thickness, number of layers, grain size, etc.],

Response: The suggested compilation of the most important storm deposit characteristics at each investigated site is reasonable. This was also suggested by reviewer 1. We will provide this summarizing information as a new figure 14, which will combine information in table form with conceptual figures for the generation of the different sediment types (i.e. sand sheets, washover fans, and coral ridges).

and (2) adding text in the introduction, or in a new section, about published reports on geometries and thickness/grain size trends for storm and tsunami deposits to give context for Haiyan deposits.

Response: To provide more context for typical storm and tsunami signatures we will add references for the geometry, grain size, thickness and sediment sources of well-investigated, modern analogues in the introduction section: "Although all potential discrimination criteria have been observed for both tsunami and storm deposits from a global perspective (Shanmugam, 2012), local comparisons assign several decimeter thick, laminated deposits with inland extents of a few tens to hundreds of meters that often show foreset bedding and tend to thin and fine landwards to typical storm signatures, while tsunami deposits tend to be rather thin, are composed of a few layers with massive or normally graded structure, and may extend inland for several kilometers

(Tuttle, 2004; Morton et al., 2007; Switzer and Jones, 2008; Goto et al., 2011)." Like-wise, in response to a comment of reviewer 1, we will add information on discrimination criteria between tsunami and storm deposits in the discussion section.

I am not entirely sure, but it appears that all the figures use distance along transect rather than distance from the shoreline. This is supported by the text on Page 6, Line 24, "shallow reef outcrops occur at 180 m transect length." I also measured the distance along transect at Tolosa (Fig. 5). TOL14 is about 120 m inland from the shoreline, but is plotted at about 145 m in Figure 3 and 5. I measured the distance along transect at Tolosa to be about 140 m. Do all the figures use distance along transect rather than distance from the shoreline? If so, they need to be corrected. Distance along transect is meaningless because a change in transect orientation results in a different distance along transect for the same distance from the shoreline, the physically meaningful parameter.

Response: The trenches are indeed plotted against distance along transects in all figures (only flood marks at TOL are already plotted with distance to shoreline). Since we agree that the distance perpendicular to the shore is the physically meaningful parameter, we will adjust all distances in figures 2, 3, and 5 to be consistent with distance to shoreline. The new numbers for distance to shoreline will also be implemented in the text.

I disagree with the statement on Page 1, Line 20 in the Abstract that Haiyan deposits, "might also function as a benchmark example for a general discrimination between storm and tsunami deposits." The Haiyan deposits are part of a spectrum of possible storm deposits; however, because of the presence of surf beat creating a tsunami-like bore they may not be typical. If so, although valuable to illustrate the spectrum of possible storm deposits, they may be more atypical than typical and therefore not a "benchmark example for a general discrimination between storm and tsunami deposits."

Response: We agree that the expression "benchmark" example might be misleading. What we want to express is that the deposits formed by Haiyan describe an extreme case of storm deposition that should be considered for the discrimination of cyclone and tsunami deposits. A similar observation could be made for boulders along the coast of Samar that were moved during Haiyan (May et al., 2015): due to the exceptional hydrodynamic conditions related to Haiyan's storm surge (i.e. the generation of surf beat), much larger boulders than typically related to storms could be moved, which is a valuable information for the discrimination of storms and tsunamis in the geological record, although Haiyan might rather be an atypical example. We will modify the section to: "As these sediments and landforms were generated by one of the strongest storms ever recorded, they not only provide a recent reference for typhoon signatures that can be used for palaeotempestological and palaeotsunami studies in the region, but might also increase the existing spectrum of possible cyclone deposits. Although a rather atypical example for storm deposition due to the impact of infragravity waves, it nevertheless provides a valuable reference for an extreme case that should be considered when discriminating between storm and tsunami deposits in general."

A discussion of preservation potential of the Haiyan deposits would provide insight into whether the deposits observed in this study would be found in the geologic record. Instead of using "geological imprint" in the first sentence of the Conclusions (Page 17, Line 2) use "deposits" because preservation of the deposits is not addressed. Preservation of storm deposits in environments investigated in this study is a rather large topic and worthy of another paper. But, although it might seem like semantics, the "geological imprint" is unknown at this time because whether the deposit will be preserved is unknown and how it will be altered as time passes is also unknown. Same comment about using "geological legacy" in the next sentence (Page 17, Line3). I suggest using "deposits" again.

Response: The reviewer is of course right. The terms "geological record" or "legacy" imply information on the preservation of the deposits that we do not – and within the

frame of this paper cannot – present. Meanwhile we collected some data about the preservation of some of the documented deposits, but including this new data and discussing it would be too much for a single paper. Therefore, we will strictly use the term "deposits" in the revised version of the manuscript.

Also in the conclusion is the statement, "the sandy onshore deposits left by Haiyan are very similar to those generated by tsunamis." Rather than give a qualitative qualifier of "very similar", which means different things to different people, list the similarities.

Response: At this point we will only list the major similarities between Haiyan induced sand sheets and washover fans on the one hand, and the respective tsunami features on the other hand: sedimentary structure, sediment sources, and granulometry. For more details we will refer to the new overview figure (Fig. 14), which summarizes the main characteristics of each type of deposition during Haiyan, and refers to references reporting similar deposits generated by tsunamis.

In the Abstract (Page 1, Line 3) and Conclusions (Page 17, Line 4) it would help the reader if you clarified what "Extended onshore sand sheets"/"extensive sand sheets" mean. Is there an inland distance that a sand sheet extends inland that you classify as "extended"/"extensive"? Perhaps it would be better to specify how far inland the sand sheets extend. How readers define extended/extensive will vary and it is better to be specific.

Response: We agree that the description is subjective and should be replaced by a number. We will replace the phrase by "Onshore sand sheets reaching 100-250 m inland..." in the revised version of the manuscript.

Shell fragments are present in the Haiyan deposit at some locations. Grain size is measured by laser diffraction and Camsizer and does not account for particle density or shape, both of which would be quite different for shells than other components and affect their settling velocity and transport in suspension. I suggest discussing how the presence of shells affect your interpretation of the grain size data. Woodruff et

al. (2008) address the differences between settling velocities of shells and siliciclastic particles. Figure DR2 in the data repository summarizes their results. The citation for Woodruff et al. (2008) is: Woodruff, J.D., Donnelly, J.P., Mohrig, D., Geyer, W.R., 2008. Reconstructing relative flooding intensities responsible for hurricane-induced deposits from Laguna Playa Grande, Vieques, Puerto Rico. Geology 36, 391–394.

Response: Shell fragments settle significantly slower compared to siliciclastic particles of the same size. Thus, larger shell fragments are found in a matrix of siliciclastic sand with much smaller grain diameter. This will make deposits with a large percentage of shells (i.e. parts of the deposits at BAN) appear coarser than they would be for siliciclastic grains only, without being related to any changes in flow dynamics. For storm deposits at TOL (no carbonates) and HER (nearly 100% carbonates without platy shell particles) no significant effects are expected. We will address this aspect in the methods section by adding "It should be noted that both approaches do not consider differences in particle shape and density, which significantly influence the settling velocity of the grains (Woodruff et al., 2008). This is particularly important for the interpretation of granulometric variations in storm deposits with a significant percentage of shells"). In addition, it will be addressed when discussing the granulometry of storm deposits at BAN in section 5.1.1: "On the other hand, the coarsening trend could just be an artefact of the reduced settling velocity of platy shell fragments, which are particularly abundant in this section of BAN 4 (Fig. 7), compared to more spherical grains (Woodruff et al., 2008)".

A fining trend in modal grain size is reported for Hernani (page 7, lines 2 and 3; Figure 3). However, the mean grain size trends of Hernani are more complicated. If trends in modal grain size are reported, please discuss how mean grain size trends are different and justify why you assign a "fining landward" trend based on modal grain size.

Response: We will add a description of the trend in mean grain size to allow comparison. Since a definition of landward fining based on modal grain size is indeed not common, we changed the sentence to read "While mean grain size does not show any

fining trend in the deposit, modal grain size decreases from 1.3 cm to 220 $\mu$m along the same section (Figs 3, 4a)". Although it may not be adequate to assign a fining trend based on modal grain size, the mode data at least indicate that there are changes in particle size along the transect (even though these do not affect the mean).

Are the statistics in Figure 3 for grain size for the entire deposit? That is, are they averages for all the grain size data for deposit? Please clarify for the reader.

Response: The data presented in the original version of the manuscript are based on mean grain size and sorting for the entire storm deposit. We will adjust the figure to show both average data for the deposit, as well as trends for individual units where appropriate. All trend lines will be labeled accordingly.

Please explain further how it was determined that sediment from the foreshore and deeper water are part of the Haiyan deposit (Page 132, Line 16). Are there grain sizes present in the Haiyan deposit that are not from the beach? Can this be sediment picked-up landward of the beach? Were there samples collected from the foreshore and nearshore close in time to when Haiyan impacted the Philippines that have grain size data?

Response: Unfortunately, reference samples were only collected and analyzed from the beach. Additional samples from the foreshore, deeper water, and terrestrial environments were not collected. The interpretation that minor proportions of the Haiyan deposits are derived from sources different from the beach are based on the differences between the beach reference samples and the typhoon deposits in terms of granulometry and faunal composition (species and abrasion). This other sediment sources could in principle be areas seaward of the littoral zone, but also areas landward of the storm deposits. In response to this comment and a similar comment made by reviewer 1 we will change the section to read: "Nevertheless, obvious differences in the granulometry and faunal composition of the sand sheets and modern beach sand (Fig. 8b, S4) may indicate also minor contributions of sediments from other source areas (the

foreshore, deeper water, or landward areas), as reported by Pilarczyk et al. (2016) for deposits of Typhoon Haiyan from Tanauan (Leyte) and Basey (Samar). Alternatively, at least the differences in foraminifer taphonomy may reflect alteration of the sediments due to wearing and fracturing of foraminifer tests during transport in high energy flows (Quintela et al., 2016)."

What is meant by "a rather normally graded structure of these sand sheets" (Page 13, Line 9)? This is important because grading of deposits may be a discriminator of storm versus tsunami deposition. Were the Haiyan deposit suspension graded, as has been observed for deposits formed by several recent tsunamis and for paleotsunami deposits? (for an explanation of suspension grading see: Jaffe, B.E., Buckley, M.L., Richmond, B.M., Strotz, L., Etienne, S., Clark, K., and Gelfenbaum, G., 2011, Flow speed estimated by inverse modeling of sandy sediment deposited by the 29 September 2009 tsunami near Satitoa, east Upolu, Samoa, Earth-Science Reviews, v. 107, p. 23-37, doi:10.1016/j.earscirev.2011.03.009.)

Response: This should read "mostly normally graded structure". While slight normal grading could be detected for thicker layers (close to the coast), the small thickness of the sand layers further inland did not allow for unambiguous identification of grading. We assume that even the thinner parts of the sand sheets might be normally graded, but cannot prove this by measurement data. However, regardless of normal grading can be observed or not, it is hard to say if they are suspension graded or not. Particularly, since the assignment of most normally graded sections within sand sheets is not based on laboratory analyzes (because they were not sampled in higher resolution). Only at BAN B sand sheets were sampled by pushcores and high-resolution data are available. Although the grain size distributions are not multimodal, improved sorting with decreasing grain size cannot be observed. A clear assignment of suspension-grading is therefore not possible.

Missing reference for Haiyan surf beat: Roeber, V and Bricker, J., 2015, Destructive tsunami-like wave generated by surf beat over a coral reef during Typhoon Haiyan,

Nature Communications (6), DOI: 10.1038/ncomms8854.

Response: We meanwhile had already realized the missing reference and included it in the paper and reference list.

Other comments:

Page 2, Line 8- Suggest changing "coastal disasters in the immediate past" to "recent coastal disasters".

Response: Okay, will be changed accordingly.

Page 2, Line 15- Suggest ending the sentence after "records" and change the next part of the sentence to a new sentence "This discrepancy is great because cyclones usually follow an inverse power law (Corral et al., 2010).

Response: Okay, will be changed accordingly.

Page 2, Line 18- Suggest changing "even events of the highest magnitudes" to "large events".

Response: Okay, will be changed accordingly.

Page 2, Line 25- Suggest changing "are particularly" to "are".

Response: Okay, will be changed accordingly.

Page 3, Line 16- Suggest changing "the significance of seasonality" to "seasonal variability".

Response: Okay, will be changed accordingly.

Page 4, Lines 10-18 and later in the paper as well- (i), (ii), (iii) are not needed and are distracting.

Response: As already mentioned in our reply to reviewer 1, all numberings will be removed in the revised version of the manuscript.

Page 4, Line 31- Suggest changing "typically shows" to "is characterized by".

Response: Okay, will be changed accordingly.

Page 4, Line 31- Again, (i), (ii), (iii) are not needed in this paragraph.

Response: Okay, see reply to comment above.

Page 4, Line 32- The fetch over the Pacific, not the narrow shelf, is the reason that Eastern Samar has high swell waves.

Response: We agree that the fetch is clearly the dominating factor for swell generation. Therefore we will delete the narrow shelf as an additional argument in the revised version of the manuscript.

Page 5, Lines 13 and 14- The times of day for the DGSP are not relevant and should be omitted.

Response: We agree that this information is not required to understand the presented data. We will delete it in the revised version of the manuscript.

Page 5, Line 15- Suggest changing "were recorded by levelling" to "were documented by measuring elevations of".

Response: Okay, will be changed accordingly.

Page 5, Line 20- Suggest deleting "directly".

Response: Okay, will be changed accordingly.

Page 5, Line 23- Define what you mean by representative. Typical thickness? Typical sediment grain size? Typical structure?

Response: Push cores were taken from deposits with sedimentary structures typical for the respective sites. This was supposed to guarantee sampling of all stratigraphical units documented at a site (and documentation of the differences between units). Since grain size and thickness of individual units vary in lateral direction, the push cores can

only reflect part of this spectrum.

Page 5, Line 27- Chemical formula contain subscripts for the number of atoms for elements

Response: This might be a formatting problem in the PDF version. We will, however, take care that the formula are shown with correct formatting.

Page 6, Line 17- Is Barangay capitalized?

Response: Okay, will be written in capital letters in the revised version.

Page 6, Line 19- Suggest changing "the two tropical storms/depressions recorded between Haiyan and this field survey on January 19th and February 1st respectively" to "the two tropical storms/depressions hitting the Philippines on January 19th and February 1st, respectively, which is after Haiyan and before this field survey".

Response: Okay, will be changed accordingly.

Page 7, Line 6- Are the values for grain size in ")"thinning and fining landward from 8 cm and a mean of 570 _m at 130 m from the shoreline (HER 8) to only 3 mm and a mean of 223 _m at 260 m (HER 3) (Fig. 3)." for the mode or mean? It appears from Figure 3 that they are for the mode, but I am not sure because this for Unit 1 and it is not clear what is shown in Figure 3.

Response: The values are indeed for the mode, which is as well shown in figure 3. We will clarify both text (replace mean by mode) and figure 3 (state the respective units grain size trends are plotted for) in the revised version of the manuscript.

Page 7, Line 21- Delete "According to".

Response: Okay, will be changed accordingly.

Page 7, Line 22- Suggest changing "bushes, Haiyan" to "are evidence that Haiyan".

Response: Okay, will be changed accordingly.

Page 7, Line 30- Suggest changing "Pre and post-typhoon" to "Pre- and post-typhoon".

Response: Okay, will be changed accordingly.

Page 8, Line 9 and later in the text- The use of the word "profiles" is not standard. Suggest changing "profiles TOL 7-14" to "trenches TOL 7-14". This suggestion applies everywhere in the text where "profile" is used to describe a trench.

Response: Okay, will be changed accordingly.

Page 8, Line 15- Specify what "slightly inclined" means.

Response: The laminae are dipping landwards with an angle of ∼10-15°. We will add this information in the revised version of the manuscript.

Page 8, Line 21- What is meant by "moderate flooding". I have no idea what is moderate. Please be specific by giving a spatial extent and/or a water depth.

Response: Since information on flooding was inferred from eyewitness observations, exact (i.e. measured) values for flooding extent and water levels cannot be provided. However, based on the observations, estimations for flooding extent (not more than a few 10s of meters) and water levels (not more than ∼3 m above msl) will be stated in the revised version of the manuscript.

Page 8, Line 25- How thin are the sand patches?

Response: The documented sand deposits do not exceed a maximum thickness of 10 cm; most of them are in the range of 1-3 cm thickness.

Page 9, Line 1- Delete "single". It is not needed.

Response: Okay, will be changed accordingly.

Page 9, Line 1- Suggest changing "in either direction" to "crest elevation".

Response: Okay, will be changed accordingly.

Page 12, Line 30- Suggest changing "show comparably large inland extents exceeding 100 m" to "extend at least 100 m inland".

Response: Okay, will be changed accordingly.

Page 13, Line 13- Suggest changing "dedicated" to "attributed".

Response: This section will be changed in response to a comment of reviewer 1. In this context the mentioned expression will be removed anyway.

Page 16, Line 21- Suggest changing "confined" to "indicated".

Response: Okay, will be changed accordingly.

Page 25, Figure 3- Be consistent with line types in each panel. Sorting is a different line type for HER 3-9 than for TOL 3-14 and BAN 1-3. A minor point, but why not make it easier on the reader to compare panels? Also, a solid line is used for both the mode and mean in different panels. Why not use a solid line for the mean and another line type for the mode?

Response: We agree with the reviewer that a consistent style for all panels would facilitate reading the figure. The line styles for mean, mode and sorting will be homogenized in the revised version of the manuscript.

Page 25, Figure 3- Why does the thickness scale for TOL 3-14 start at -2? This makes it difficult to determine the thickness of the more landward deposits. Why not start the scale at 0 to make it easy to determine the thickness of landward deposits?

Response: We see the point and will adjust the scale for thickness.

Page 25, Figure 3- Why is there a vertical dashed line at 40 m in the TOL 3-14 panel? Please explain this line in the caption.

Response: The vertical line is a drawing artefact that has no important meaning and will be deleted.

Page 25 Figure 3 caption- The mean grain size of HER 3-8 doesn't monotonically fine landward. See earlier comment on mode versus mean and description/definition of landward fining.

Response: It is true that the fining trend at TOL is restricted to the mode but does not apply to the mean. We will adjust the figure caption to match this observations.

Page 29, Figure 6- For consistency, add the column that indicates grading by the shaded triangles.

Response: Okay, will be added in the revised version.

Page 29, Figure 6 caption- The transect is not coast-perpendicular.

Response: The figure caption will be changed to read "Transect illustrating the succession of typhoon deposits in landward direction".

Page 33, Figure 10 caption- The transect is only shore-perpendicular for trenches 1 and 2, not 3.

Response: The figure caption will be changed to read "onshore sediments of Haiyan were investigated in three trenches (MOL 1–3) along a landward transect (T1)".

Please also note the supplement to this comment:
http://www.nat-hazards-earth-syst-sci-discuss.net/nhess-2016-224/nhess-2016-224-AC2-supplement.pdf
* * *

---

## Author Response (AR1)

Reply to comments of reviewer #1:

Dear reviewer,

Thanks for the thorough and constructive comments to our submission. The integration of these suggestions will definitely help to further improve our manuscript. In the following we will address each of the comments separately (author replies in blue). All references to line numbers are based on the revised version with track changes.

*Dear Editor and authors, the work by Brill et al. presents an insight into the Typhoon Haiyan's sedimentary record in coastal environments of the Philippines and its palaeotempestological implications. I commend the authors on a well-written and interesting manuscript. The authors addressed a topic with particular societal relevance due to the consequences of these catastrophic events for coastal areas. In this case, it is particularly relevant to say that the authors conducted extensive fieldwork and were also able to complement that with results derived from the application of some sedimentological proxies (grain-size, XRD and magnetic susceptibility). The data set gathered seems to be solid and very interesting from a scientific point of view. Overall, the manuscript has a clear structure and aims. However, in my opinion, several aspects should be addressed by the authors before the manuscript is accepted for publication, please see details below.*

*Although most of the issues I raise (please see below) are minor, I would like to stress that the authors need to be more consistent in terms of the vertical datum that they used. They should rewrite some of the numbered lists and make the text easier to follow.*

All height information are presented in meters above mean sea level in the revised manuscript version. The respective sections have been rewritten for clarification (see also replies to specific comments below for more details).

*They need to address more clearly the differences between tsunami and storm deposits and they should discuss transport modes and its implications for the depositional signature of the Typhoon Haiyan's sedimentary record in coastal environments of the Philippines (Jaffe et al., 2012 - Sed Geol).*

We added some more explanations concerning potential features that may allow to differentiate between tsunami and storm deposits in the discussion section of the revised version. There, we provide sufficient details on potential tsunami and cyclone indicators (and the problematic that all of them are dependent on local setting) to allow the reader to follow our argumentation (page 17, line 16 to page 18, line 6). However, in our opinion this discussion has already occurred in a large number of publications, and we therefore will also refer to the respective literature for more information (we will provide more references dealing with this topic) instead of going too much into detail.

We also enlarged the discussion of transport modes for the different storm signatures reported (page 14, line 10-21; page 14, line 29 ff). In agreement with observations on the deposits of other tropical cyclones (e.g. Williams, 2009 at the US coast), we mainly attribute the formation of the slightly normally graded sand layers to settling from suspension during an early stage of the storm surge (at least at sites HER and TOL this initial flooding of the back-barrier areas was supported by infra-gravity waves). On the other hand, the laminated washover fans are interpreted to be the result of bedload transport over the coastal barrier, related to distinct storm waves during a later stage of the storm surge.

*On top of this, they should stress that although local settings and sediment source are fundamental aspects that control storm deposit bed formation, there are a group of common features between the studied sites and that they share characteristics with deposits elsewhere (maybe adding a table summarizing these sedimentological features would help to the reader).*

We have highlighted this aspect in both the discussion section (page 17, line 31 ff) and the conclusions (page 18, line 31). This does also include a new figure (figure 14) listing the

characteristics of the different storm features (including comparison with features of other storms and tsunamis) as suggested by the reviewer.

**Abstract**

The abstract is clear and well written. However, the authors need to clarify if they are studying 3 or 4 sites (they mention 4 sites here but mention 3 sites on page 3 line 10).

We aligned the information on the number of locations in the manuscript to 4 sites: Hernani (Samar), Tacloban (Leyte), Carbin/Molocaboc (Northern Negros), and Bantayan.

I also suggest that the authors need to provide more comparisons with palaeotsunami data to sustain their sentence in line 20. In my opinion, that sentence should end "(...) typhoon signatures that can be used for palaeotempestological studies." the rest of the sentence should be deleted unless discussion is enriched with further topics on the comparison between tsunami and storm depositional signatures.

We slightly enlarged the discussion of tsunami signatures in the revised version of the manuscript (see reply to comment above) and, thus, will leave the sentence as is.

**Introduction**

Page 2, line 18, once we started talking about using geological record for several millennia we should also mention (and take in consideration) sea-level changes especially when we are using just a few specific study sites.

We absolutely agree with the reviewer that sea-level changes (and changes in palaeogeography) since the time prehistoric tsunamis or storms made landfall have to be considered when interpreting their geological records. We added a short section to highlight the importance of this aspect (page 2, line 21).

Page 2, line 25 - "natura are" - spelling mistake

Was corrected.

Page 2, line 27 - suggest you delete text up to line 30... "Here, we report..."

We deleted the part of the mentioned section referring to a previous study on coastal boulders from the same area (May et al., 2015). However, we think the sentence concerning the potential relevance of the presented data for a general discrimination of tsunami and storm deposits is necessary and should not be deleted, since we believe that sediments of exceptional typhoons such as Haiyan definitely contribute to this discussion.

Page 3 - I believe you should clarify or stress again the aims of your work, in particular at the end of the Introduction.

We stress the aims of our study at the end of the introduction to make this aspect clearer for the reader (page 3, line 4-10). The revised section reads: "The major aims of this study are to (i) document Haiyan's impact on the sedimentology and geomorphology of heavily affected coastal areas by recording onshore and intertidal sedimentation, coastal erosion and geomorphological changes. Based on these data, sedimentary and geomorphological typhoon signatures typical for the study area shall be established. In addition, (ii) the spatial variability of these typhoon signatures due to site-specific characteristics such as the local topography,

bathymetry, geology, and hydrodynamics as well as the exposure to the typhoon track shall be investigated. Finally, (iii) the potential of these modern typhoon deposits will be evaluated in respect of possible implications for the identification of prehistoric cyclones in the geological record."

**Study area**

Page 3, line 10 - "three study areas"???

We changed to "four" to be in accordance with the information given earlier (see reply to first comment).

Page 3, line 28 - when you refer to Samar please make reference to Figure or provide some clues about the specific location.

We added a reference to Figure 1a, where the track of Typhoon Haiyan as well as the location of Samar are indicated.

Page 4, line 5 - "three distinctive wave pulses" - Three sets of waves? How was this established? Was it measured? What was the Hs difference between the different pulses? Where any of these pulses related with infra-gravity waves?

These three pulses of flooding with periods much longer than those of wind waves are based on eye-witness observations, so their heights (or the differences of heights between the pulses) are not well constrained. As mentioned on page 4, line 4, numerical models suggest that they may be the result of seiches (i.e. standing waves) in the semi-enclosed San Pedro Bay (Mori et al., 2014).

Page 4, line 10 - i) and ii) and iii) - numbered lists were used intensively in this manuscript. I do not think they were used properly. Each numbered topic is very extensive and the reader is not guided properly. I suggest you rewrite all parts in the manuscript were you used numbered lists. Either you simplify the topics or you should write them as different sentences and start the sentences with "on the other hand" or "moreover" or etc.

We changed the structure of this section according to the reviewer's suggestions.

Page 4, line 11 - "model-predicted". Throughout the manuscript you mention several times this but provide no details about modeled data. I strongly suggest you do that! Which model was used? What was the source data? What equations were used to calculate Hs, etc? etc, etc?

There are a number of different models that have been used. The data presented by Bricker et a. (2014), Mori et al. (2014), Cuadra et al. (2014), May et al. (2015b), Roeber and Bricker (2015), Kennedy et al. (2016), and Soria et al. (2016) are all based on models with different specifications. Although we agree that knowledge of models and parameters is important to evaluate the model output, providing all specifications in the manuscript would be a lengthy description that in our opinion would rather distract the reader. Since we refer to the original literature wherever we mention modelled data, interested readers can easily consult these articles for further information.
While the detailed specifications of the models are in our opinion not required to understand the presented data, there are two main types of models that have been used to predict flooding levels, and which to discriminate is indeed important for the interpretation of our data: (1) numerical storm-surge models combined with phase-averaged wave models are routinely used to model surge heights for larger areas (Bricker et al., 2014; Cuadra et al., 2014; Mori et al., 2014; May et al., 2015; Soria et al., 2016); (2) numerical surge models with phase-resolved (boussinesq-type) wave models are required to reproduce the interactions of waves with the local topography, which may generate infra-gravity waves (Roeber and Bricker, 2015; Kennedy

et al., 2016). So while we think it is sufficient to refer to the original literature for detailed model specifications, we explicitly included the discrimination of phase-averaged and phase-resolved wave models in the revised version of the manuscript (page 4, line 9-22).

Page 4, line 13 - throughout the paper you refer to, at least 3, height (vertical datum) units (atl, msl, above mean low water and depth below surface). This makes it really hard for the reader. I strongly suggest you convert all to m above mean sea level!

We agree with the reviewer that the use of different height levels might be confusing for the reader. To allow for comparability between all sites, we now provide all height references for data presented in this research (topography, sediments, flood marks, etc.) in meters above or below mean sea level (above/below msl). However, since the same values relative to mean sea level may – depending on the elevation of the ground – have completely different implications for sedimentation, we also provide the flow depth in meters above ground level in case of the measured flood marks. For describing the stratigraphies of sediment profiles, we will stick to meters below surface, since here a relation to sea level would be rather confusing. We state this information explicitly in the methods section of the revised version (page 5, lines 27-28).

Page 4, line 24 - please provide reference after "Philippine plate".

We added Rangin et al. (1989) as a reference.

Page 4, line 29 - suggest you replace "originating" with "originated" and add "denser" to make the sentence ..."darker and denser minerals..."

The sentence was changed accordingly.

Page 4, line 31 - Please see comment to page 4, line 10.

The structure of this sections was changed according to the reviewer's suggestions.

*Methods*

Page 5, line 12 - "along-shore perpendicular transects". So, cross-shore? What was the space between consecutive profiles? Did you created a DEM?

Due to the limited time available at each study location during the survey, we measured only a single transect at most of the sites. Only on Carbin Reef (6 transects) and at BAN B (3 transects) several transects were measured (all of them are documented in the respective figures). Consequently, no DEMs were created as well.

Page 5, line 16 - heights - Please see comment to page 4, line 13.

As already mentioned in our reply to page 4, line 13, all heights are provided in meters above mean sea level in the revised version. Flow depth (so meters above ground surface) is provided additionally in case of flood marks.

Page 5, line 21 - please replace "was" with "were".

Was corrected.

Page 5, line 23 - this is a relevant aspect of the manuscript. Here, you suggest that in some locations you only used one core? Do you think this is enough for well supported interpretations? Especially, when later you refer to all local specific conditions and lateral variations of the deposit!!

We indeed analyzed only a single sediment core for site BAN A. For all other locations (HER, TOL, BAN B), several samples collected at different distances to the shoreline were analyzed. We agree that there are lateral variations in terms of granulometry and faunal composition at the individual sites, which are of course not covered by the single core at BAN A. However, at site BAN A the lateral extension of the deposit is only ~10-20 meters. We checked the lateral structure by means of trenches and the section sampled by BAN 4 is assumed to be representative for the entire washover fan. Especially for the comparison of BAN 4 with other sites the lack of lateral data should be negligible, since the differences between sediments from different sites are much more pronounced. Although some limitations must be expected for granulometry and faunal composition that vary laterally, we therefore assume that the results of this single core (i) represent the typical sediment composition at BAN A quite well, and (ii) can already document the main differences compared to the other locations.

Page 6, line 13 - suggest you compare your approach with Quintela et al. (2016 – Quaternary International) methodology to identify allochthonous Foraminifera species within high energy deposits.

The methodological aspects of foraminifer determination, counting and taphonomy classification used in our study should be similar to those applied by Quintela et al. (2016). Particularly the argument that higher percentages of broken foraminifer tests are the result of high grain density in the traction-transport dominated parts of high energy flows may be of importance for discussing our foraminifer assemblages, and is considered in the discussion section of the revised version (page 14, line 19-21).

**Results**

Page 6, lines 19, 26, 27, 28 - heights - Please see comment to page 4, line 13.

All heights are provided in meters above mean sea level and (additionally) as flow depth above surface.

Page 6, line 23 - please refer to Figure 2 (?).

A reference to figure 2 was added.

Page 7, line 3 - I believe you should provide/describe more grain-size data information. I suggest you add information on the D10, D90, sorting and unimodal or bimodal character of your samples.

We complemented the grain-size information for all sites, and provide data on mean, sorting, and modality for each site.

Page 7, line 9 to 13 - I feel that in the discussion you should refer to the relationship between reworking and sediment concentration. Did you detected more reworked material in the basal sector of the storm layer or on the top? How was this correlated with grain-size?

In case of HER 10, no clear vertical trend in foraminifer taphonomy or species composition could be detected. There is rather a slight correlation between coarser grain size and stronger reworking (Fig. 4). This is similar for core BAN 4, where strong reworking correlates with larger grain size as well (Fig. 12). However, since sediments first tend to become coarser towards the top of BAN 4 and fine afterwards, reworking is highest in the central section of BAN 4.

Page 7, line 23 - again, the heights...what vertical datum did you used this time?

All heights are provided in meters above mean sea level and, in case of flood marks, (additionally) as flow depth above surface.

Page 8, line 2 - please refer to Figure.

We inserted a reference to figure 6.

Page 8, line 29 - I guess you should cite it as personal communication.

We now cite the observation as "personal communication".

Page 8, line 30 - heights - Please see comment to page 4, line 13.

The heights are provided in meters above/below mean sea level.

Page 9, line 16 - Rsubt was collected at approximately what depth?

The sample was collected at 0.5 m below mean sea level. We added this information in the revised manuscript.

Page 10 - line 14 to 17 - the fact that the basal sector is slightly finer than the middle section is not just a consequence of the more erosive character of the initial stage of the event? The following phases benefited from a lowered coastal sector thus were capable of transporting coarser sediments farther inland.

The proposed mechanism is a very plausible explanation for the observed stratigraphical pattern at this location, because both units are assumed to be deposited by similar processes, i.e. wave swash overtopping the coastal barrier. We briefly address this aspect in the discussion section of the revised version (page 15, line 11-16).

Page 11, line 16 to 19 – this just reflects the dominance of the original (2nd cycle) sediment source.

We agree that the mineralogy and geochemistry mainly indicate the differences between limestone and volcanic environments. We already address this topic in the discussion section (page 15, lines 15 ff in the original version of the manuscript).

Page 11, line 20 - I believe it is the first time you refer to principal components analysis, I suggest you refer to it in full.

The abbreviation PCA is already mentioned in the methods section. However, we agree that referring to it in full at this position might facilitate reading.

Line 12 - line 13 - this strongly suggests this area as the main sediment source.

That is how we interpret this data in the discussion section as well. To make this implications already clear in section 4.5, we added a brief explanation.

**Discussion**

Page 12, line 27 - again the numbered list.

We removed the numbering to facilitate reading.

Page 12, line 30 - "normally graded or massive layers of sand". This implies totally different sediment transport modes (suspended grading and traction). I believe you should add a sentence here to comment on this and discuss reasons for the differences observed.

This should read "normally graded to massive layers of sand". While slight normal grading could be detected for thicker layers (close to the coast) and especially those analyzed for

vertical grain-size variations in the laboratory, the small thickness of the sand layers further inland did not allow for unambiguous identification of grading. We assume that even the thinner parts of the sand sheets might be normally graded. But since we cannot prove this (macroscopically their structure could be both massive and slightly graded), we prefer to describe them as "normally graded to massive".

Page 13, line 3 - I suggest you add references from one of the several works conducted by Donnelly et al. or Liu et al. in the eastern coast of the US.

We added Donnelly et al. (2006) as a reference from the US coast.

Page 13, line 8 - I believe you should also mention infra-gravity waves.

Actually, the mentioned "long-wave phenomena" already include infra-gravity waves that can result e.g. from surf beat. To make this clearer, we explicitly use the term "infra-gravity waves" at this position.

Page 13, line 13 - very very interesting but why? Can you add a comment on this?

The deposits described by Williams (2009) have actually a very similar structure as those described on the Philippines: a finer, graded sand layer formed during the initial inundation of the back-barrier plains, topped by washover deposits during a later stage of the storm surge. We therefore assume similar transport modes for our deposits, i.e. suspension settling for the graded sand sheet and bedload/traction for the formation of the washover lobes. The role of long-wave phenomena for the deposits presented in our study (infra-gravity waves at HER and seiches at TOL) is probably a contribution to higher and more extensive flood levels, but not significantly different sedimentation processes.

Page 13, line 16 - now it is important to know at what depth was your sample (Rsubt) retrieved!!

As mentioned before, the sample was collected at 0.5 m below mean sea level. It contrasts significantly in terms of faunal composition from the storm deposits, while the littoral reference samples from BAN reveal a similar granulometry and faunal composition with the typhoon deposits.

Page 13, line 25 to 29 - I suggest you rewrite this sentence.

We changed the structure of this section to make it clearer for the reader.

Page 14 - line 1 - you must refer, for example, to the work of Komar and Wang (1984) or Komar (in Mange, 2007).

We added Komar and Wang (1984) as a reference for density sorting.

Page 14, line 6 to 9 - Agree with interpretation.

Page 14, line 18 to 20 - I accept your interpretation but I think formation of ridges implies a "continuum in time" more suitable with normal storm regime and a succession of events.

We agree that ridges might form during several successive events rather than single storms. In fact, we state the possibility of ridge formation by several typhoons (with significant growth of a pre-existing ridge during Haiyan) further down in this section.

Page 14, line 21 - please see comment to page 4, line 10.

We removed the numbering.

Page 14, line 27 and 28 - I think this partially contradicts statements above. I suggest you rewrite it

Since our evidence is not unambiguous without robust age data, we have to provide both possible explanations for the ridge formation. Of course these explanations partially contradict each other, because only one of them can reflect reality. We rewrote the section to clarify this aspect.

Page 15, line 3 - please quantify the "remarkable amplification".

In the central part of the bay, water levels of more than 8 m above sea level were recorded, which is much higher than in Haiyan-affected areas not subject to infra-gravity waves (HER) or raised water levels related to shore configuration (TOL). We added the value in the revised manuscript.

Page 15, line 4 - which models?

Here we refer to storm surge models combined with phase-averaged wave models (for details we refer to the original reference by Bricker et al., 2014) that do not account for the effects of infra-gravity waves (phase-resolved wave models). We added this information in the revised version.

Page 15, line 8 to 14 - Reasoning perfectly reasonable

Page 15, line 15 - in fact, you can add that sediment source is always a decisive factor.

We now explicitly mention sediment source as a further decisive factor.

Page 15, line 24 to 27 - please rewrite this sentence.

The sentence was rewritten.

Page 15, line 28 – is backwash really relevant for depositional imprints in storm events? Against gravity?

Usually backwash is probably of minor or no importance during storms. However, sample HER 10 was collected close to a fluvial channel, where the backwash was not against gravity.

Page 16 - line 4 to 7 - here, you acknowledge that site-specific limits extrapolations of your conclusions. I agree and it really is hard to overcome this but, in my opinion, this field of science will progress will a multitude of sites, settings and events being studied. Maybe you can add a sentence regarding future work.

We agree that the value of case studies is their contribution to the database of locally and regionally differences of storm and tsunami deposits. We already tried to address this aspect later in this section. However, we modified our statement to highlight this message.

Page 16, line 20 - you need to add a comment on the different settings studied by Hawkes et al. (2007) and Goto et al. (2011).

The settings mentioned here are wide coastal plains or beach-ridge plains with a low topography that does not hinder lateral inundation and sediment transport due to steep slopes. Indeed, this is not true for the sites investigated by Hawkes et al. (2007), so we replaced the reference with observations on 2004 Tsunami deposits from a beach-ridge plain in Thailand by Jankaew et al. (2008). While most of the sites presented here have a steeper topography and are therefore not directly comparable, similar conditions are given at TOL. Nevertheless, in spite of high surge levels >5m, sand transport is limited to not more than ~300 m (although the topography of the coastal plain would allow for much more extensive deposition).

Page 16, line 20 to 27 - your conclusions are somewhat constrained because you did not compared tsunami and storm deposits in the same locations (e.g. Kortekaas and Dawson, 2007).

We agree that the conclusions are limited due to this fact. However, by comparing with tsunami deposits from sites with similar settings (similar flood levels and similar topography), we think our findings nevertheless add to the observation that sediment extent tends to be a discriminative feature.

Page 16, line 28 to 31 - 2 units by one event is totally different from 2 units by more than one!! You need to discuss this!!

We agree that both interpretations would have completely different implications. But since we are not able to prove one of the two possibilities (the formation of the washover fans at TOL could neither be proved by eyewitnesses nor by satellite images), we have to present both options for this location.

Conclusions - Page 17, line 2 - "local factors"... After so much work, it is important to stress the relevance of local conditions. In fact, I suggest you provide a geomorphological sketch (conceptual) model that describes accurately the initial pre-event conditions and the deposit after the event.

The idea to present the main outcomes of the study in a conceptual figure is very reasonable. We decided to merge this figure with the table summarizing the characteristics of the storm features presented in this paper (see reply to an earlier comment). This figure (figure 149 includes schematic sketches for the formation of each storm feature (coral ridge, sand sheet, washover fan). It, however, does not provide a separate figure on the pre-event situation. This is poorly constrained and, thus, cannot be documented with sufficient detail.

**References**

I suggest you add the above mentioned references.

The mentioned references will be included.

**Figures**

The figures have very good quality, are well-designed and are informative.

**Supplementary material**

Useful.

I believe that in scientific terms the authors developed quality work that clearly deserves publication in NHESS, subject to very few minor changes. Regards Pedro J. M. Costa

Reply to comments of reviewer #2:

Dear reviewer,

Thanks for the thorough and constructive comments to our submission. We think integration of these suggestions will help to further improve our manuscript. In the following we will address each of the comments separately (author replies in blue). References to lines always refer to the revised version with track changes.

*Dear Editor and authors,*
*The research presented in this paper is a valuable contribution to the documentation of the sedimentary record of storms. Documenting the sedimentary record of Haiyan is critical because it is a very large storm and there is a need for data on erosion and deposition for extreme storms. The scope of the field investigation is impressive and the wide variety of laboratory analyses performed create a large data set that can be used to discriminate storm deposits from tsunami and other high-energy deposits. The figures are informative and well done. I recommend that this paper be published after revision. I suggest possible ways to improve the paper below.*

**Main comments:**

*Perhaps because of the amount of data presented I found the paper hard to follow. I suggest two things to help the reader: (1) a table summarizing all the sites visited and their important characteristics [source sediment available for transport, topography, etc.] and Haiyan deposit metrics [inland extent, maximum thickness, number of layers, grain size, etc.],*

The suggested compilation of the most important storm deposit characteristics at each investigated site is very reasonable. This was also suggested by reviewer 1. We now provide this summarizing information as a new figure 14, which combines information in table form with conceptual figures for the generation of the different sediment types (i.e. sand sheets, washover fans, and coral ridges).

*and (2) adding text in the introduction, or in a new section, about published reports on geometries and thickness/grain size trends for storm and tsunami deposits to give context for Haiyan deposits.*

To provide more context for typical storm and tsunami signatures we added references for the geometry, grain size, thickness and sediment sources of well-investigated, modern analogues in the introduction section (page 2, line 24-29): "Although all potential discrimination criteria have been observed for both tsunami and storm deposits from a global perspective (Shanmugam, 2012), local comparisons assign several decimeter thick, laminated deposits with inland extents of a few tens to hundreds of meters that often show foreset bedding and tend to thin and fine landwards to typical storm signatures, while tsunami deposits tend to be rather thin, are composed of a few layers with massive or normally graded structure, and may extend inland for several kilometers (Tuttle, 2004; Morton et al., 2007; Switzer and Jones, 2008; Goto et al., 2011)." Likewise, in response to a comment of reviewer 1, we added information on discrimination criteria between tsunami and storm deposits in the discussion (section 5.3).

*I am not entirely sure, but it appears that all the figures use distance along transect rather than distance from the shoreline. This is supported by the text on Page 6, Line 24, "shallow reef outcrops occur at 180 m transect length." I also measured the distance along transect at Tolosa (Fig. 5). TOL14 is about 120 m inland from the shoreline, but is plotted at about 145 m in Figure 3 and 5. I measured the distance along transect at Tolosa to be about 140 m. Do all the figures use distance along transect rather than distance from the shoreline? If so, they need to be corrected. Distance along transect is meaningless because a change in transect orientation results in a different distance along transect for the same distance from the shoreline, the physically meaningful parameter.*

The trenches are indeed plotted against distance along transects in all figures (only flood marks at TOL are already plotted with distance to shoreline). Since we agree that the distance

perpendicular to the shore is the physically meaningful parameter, we adjusted all distances in figures 2, 3, and 5 to be consistent with distance to shoreline. The new numbers for distance to shoreline have also been implemented in the text.

*I disagree with the statement on Page 1, Line 20 in the Abstract that Haiyan deposits, "might also function as a benchmark example for a general discrimination between storm and tsunami deposits." The Haiyan deposits are part of a spectrum of possible storm deposits; however, because of the presence of surf beat creating a tsunamilike bore they may not be typical. If so, although valuable to illustrate the spectrum of possible storm deposits, they may be more atypical than typical and therefore not a "benchmark example for a general discrimination between storm and tsunami deposits."*

We agree that the expression "benchmark" example might be misleading. What we want to express is that the deposits formed by Haiyan describe an extreme case of storm deposition that should be considered for the discrimination of cyclone and tsunami deposits. A similar observation was made for boulders along the coast of Samar that were moved during Haiyan (May et al., 2015): due to the exceptional hydrodynamic conditions related to Haiyan's storm surge (i.e. the generation of surf beat), much larger boulders than typically related to storms could be moved, which is a valuable information for the discrimination of storms and tsunamis in the geological record, although Haiyan might rather be an atypical example. We modified the section to: "As these sediments and landforms were generated by one of the strongest storms ever recorded, they not only provide a recent reference for typhoon signatures that can be used for palaeotempestological and palaeotsunami studies in the region, but might also increase the existing spectrum of possible cyclone deposits. Although a rather atypical example for storm deposition due to the impact of infragravity waves, it nevertheless provides a valuable reference for an extreme case that should be considered when discriminating between storm and tsunami deposits in general" (page 1, 19-23).

*A discussion of preservation potential of the Haiyan deposits would provide insight into whether the deposits observed in this study would be found in the geologic record. Instead of using "geological imprint" in the first sentence of the Conclusions (Page 17, Line 2) use "deposits" because preservation of the deposits is not addressed. Preservation of storm deposits in environments investigated in this study is a rather large topic and worthy of another paper. But, although it might seem like semantics, the "geological imprint" is unknown at this time because whether the deposit will be preserved is unknown and how it will be altered as time passes is also unknown. Same comment about using "geological legacy" in the next sentence (Page 17, Line3). I suggest using "deposits" again.*

The reviewer is of course right. The terms "geological record" or "legacy" imply information on the preservation of the deposits that we do not – and within the frame of this paper cannot – present. Meanwhile we collected some data about the preservation of some of the documented deposits, but including this new data and discussing it would be too much for a single paper. Therefore, we strictly use the term "deposits" in the revised version of the manuscript.

*Also in the conclusion is the statement, "the sandy onshore deposits left by Haiyan are very similar to those generated by tsunamis." Rather than give a qualitative qualifier of "very similar", which means different things to different people, list the similarities.*

At this point we will only list the major similarities between Haiyan induced sand sheets and washover fans on the one hand, and the respective tsunami features on the other hand: sedimentary structure, sediment sources, and granulometry (page 19, line 11-12). For more details we refer to the new overview figure (Fig. 14), which summarizes the main characteristics of each type of deposition during Haiyan, and refers to references reporting similar deposits generated by tsunamis.

*In the Abstract (Page 1, Line 3) and Conclusions (Page 17, Line 4) it would help the reader if you clarified what "Extended onshore sand sheets"/"extensive sand sheets" mean. Is there an inland distance that a sand sheet extends inland that you classify as "extended"/"extensive"? Perhaps it would be better to specify how far inland the sand sheets extend. How readers define extended/extensive will vary and it is better to be specific.*

We agree that the description is subjective and should be replaced by a number. We replaced the phrase by "Onshore sand sheets reaching 100-250 m inland…" in the revised version of the manuscript.

*Shell fragments are present in the Haiyan deposit at some locations. Grain size is measured by laser diffraction and Camsizer and does not account for particle density or shape, both of which would be quite different for shells than other components and affect their settling velocity and transport in suspension. I suggest discussing how the presence of shells affect your interpretation of the grain size data. Woodruff et al. (2008) address the differences between settling velocities of shells and siliciclastic particles. Figure DR2 in the data repository summarizes their results. The citation for Woodruff et al. (2008) is: Woodruff, J.D., Donnelly, J.P., Mohrig, D., Geyer, W.R., 2008. Reconstructing relative flooding intensities responsible for hurricane-induced deposits from Laguna Playa Grande, Vieques, Puerto Rico. Geology 36, 391–394.*

Shell fragments settle significantly slower compared to siliciclastic particles of the same size. Thus, larger shell fragments are found in a matrix of siliciclastic sand with much smaller grain diameter. This will make deposits with a large percentage of shells (i.e. parts of the deposits at BAN) appear coarser than they would be for siliciclastic grains only, without being related to any changes in flow dynamics. For storm deposits at TOL (no carbonates) and HER (nearly 100% carbonates without platy shell particles) no significant effects are expected. We address this aspect in the methods section (page 6, line 10-12) by adding "It should be noted that both approaches do not consider differences in particle shape and density, which significantly influence the settling velocity of the grains (Woodruff et al., 2008). This is particularly important for the interpretation of granulometric variations in storm deposits with a significant percentage of shells". In addition, it is addressed when discussing the granulometry of storm deposits at BAN in section 5.1.1 (page 15, line 11-16): "On the other hand, the coarsening trend could just be an artefact of the reduced settling velocity of platy shell fragments, which are particularly abundant in this section of BAN 4 (Fig. 7), compared to more spherical grains (Woodruff et al., 2008)".

*A fining trend in modal grain size is reported for Hernani (page 7, lines 2 and 3; Figure 3). However, the mean grain size trends of Hernani are more complicated. If trends in modal grain size are reported, please discuss how mean grain size trends are different and justify why you assign a "fining landward" trend based on modal grain size.*

We added a description of the trend in mean grain size to allow comparison. Since a definition of landward fining based on modal grain size is indeed not common, we changed the sentence to read "While mean grain size does not show any fining trend in the deposit, modal grain size decreases from 1.3 cm to 220 μm along the same section (Figs 3, 4a)". Although it may not be adequate to assign a fining trend based on modal grain size, the mode data at least indicate that there are changes in particle size along the transect (even though these do not affect the mean).

*Are the statistics in Figure 3 for grain size for the entire deposit? That is, are they averages for all the grain size data for deposit? Please clarify for the reader.*

The data presented in the original version of the manuscript are based on mean grain size and sorting for the entire storm deposit. We adjusted the figure to show both average data for the deposit, as well as trends for individual units where appropriate. All trend lines will be labeled accordingly.

*Please explain further how it was determined that sediment from the foreshore and deeper water are part of the Haiyan deposit (Page 132, Line 16). Are there grain sizes present in the Haiyan deposit that are not from the beach? Can this be sediment picked-up landward of the beach? Were there samples collected from the foreshore and nearshore close in time to when Haiyan impacted the Philippines that have grain size data?*

Unfortunately, reference samples were only collected and analyzed from the beach. Additional samples from the foreshore, deeper water, and terrestrial environments were not collected.

The interpretation that minor proportions of the Haiyan deposits are derived from sources different from the beach are based on the differences between the beach reference samples and the typhoon deposits in terms of granulometry and faunal composition (species and abrasion). These other sediment sources could in principle be areas seaward of the littoral zone, but also areas landward of the storm deposits. In response to this comment and a similar comment made by reviewer 1 we changed the section (page 14, line 16-21) to read: "Nevertheless, obvious differences in the granulometry and faunal composition of the sand sheets and modern beach sand (Fig. 8b, S4) may indicate also minor contributions of sediments from other source areas (the foreshore, deeper water, or landward areas), as reported by Pilarczyk et al. (2016) for deposits of Typhoon Haiyan from Tanauan (Leyte) and Basey (Samar). Alternatively, at least the differences in foraminifer taphonomy may reflect alteration of the sediments due to wearing and fracturing of foraminifer tests during transport in high energy flows (Quintela et al., 2016)."

*What is meant by "a rather normally graded structure of these sand sheets" (Page 13, Line 9)? This is important because grading of deposits may be a discriminator of storm versus tsunami deposition. Were the Haiyan deposit suspension graded, as has been observed for deposits formed by several recent tsunamis and for paleotsunami deposits? (for an explanation of suspension grading see: Jaffe, B.E., Buckley, M.L., Richmond, B.M., Strotz, L., Etienne, S., Clark, K., and Gelfenbaum, G., 2011, Flow speed estimated by inverse modeling of sandy sediment deposited by the 29 September 2009 tsunami near Satitoa, east Upolu, Samoa, Earth-Science Reviews, v. 107, p. 23-37, doi:10.1016/j.earscirev.2011.03.009.)*

This should read "mostly normally graded structure". While slight normal grading could be detected for thicker layers (close to the coast), the small thickness of the sand layers further inland did not allow for unambiguous identification of grading. We assume that even the thinner parts of the sand sheets might be normally graded, but cannot prove this by measurement data. However, regardless of normal grading can be observed or not, it is hard to say if they are suspension graded or not. Particularly, since the assignment of most normally graded sections within sand sheets is not based on laboratory analyzes (because they were not sampled in higher resolution). Only at BAN B sand sheets were sampled by pushcores and high-resolution data are available. Although the grain size distributions are not multimodal, improved sorting with decreasing grain size cannot be observed. A clear assignment of suspension-grading is therefore not possible.

*Missing reference for Haiyan surf beat: Roeber, V and Bricker, J., 2015, Destructive tsunami-like wave generated by surf beat over a coral reef during Typhoon Haiyan, Nature Communications (6), DOI: 10.1038/ncomms8854.*

The missing reference was included in the paper and reference list.

**Other comments:**

*Page 2, Line 8- Suggest changing "coastal disasters in the immediate past" to "recent coastal disasters".*

Okay, was changed accordingly.

*Page 2, Line 15- Suggest ending the sentence after "records" and change the next part of the sentence to a new sentence "This discrepancy is great because cyclones usually follow an inverse power law (Corral et al., 2010).*

Okay, was changed accordingly.

*Page 2, Line 18- Suggest changing "even events of the highest magnitudes" to "large events".*

Okay, was changed accordingly.

*Page 2, Line 25- Suggest changing "are particularly" to "are".*

Okay, was changed accordingly.

*Page 3, Line 16- Suggest changing "the significance of seasonality" to "seasonal variability".*

Okay, was changed accordingly.

*Page 4, Lines 10-18 and later in the paper as well- (i), (ii), (iii) are not needed and are distracting.*

All numberings were removed in the revised version of the manuscript.

*Page 4, Line 31- Suggest changing "typically shows" to "is characterized by".*

Okay, was changed accordingly.

*Page 4, Line 31- Again, (i), (ii), (iii) are not needed in this paragraph.*

Okay, see reply to comment above.

*Page 4, Line 32- The fetch over the Pacific, not the narrow shelf, is the reason that Eastern Samar has high swell waves.*

We agree that the fetch is clearly the dominating factor for swell generation. Therefore we deleted the narrow shelf as an additional argument in the revised version of the manuscript.

*Page 5, Lines 13 and 14- The times of day for the DGSP are not relevant and should be omitted.*

We agree that this information is not required to understand the presented data. We deleted it in the revised version of the manuscript.

*Page 5, Line 15- Suggest changing "were recorded by levelling" to "were documented by measuring elevations of".*

Okay, was changed accordingly.

*Page 5, Line 20- Suggest deleting "directly".*

Okay, was changed accordingly.

*Page 5, Line 23- Define what you mean by representative. Typical thickness? Typical sediment grain size? Typical structure?*

Push cores were taken from deposits with sedimentary structures typical for the respective sites. This was supposed to guarantee sampling of all stratigraphical units documented at a site (and documentation of the differences between units). Since grain size and thickness of individual units vary in lateral direction, the push cores can only reflect part of this spectrum.

*Page 5, Line 27- Chemical formula contain subscripts for the number of atoms for elements*

We checked this. The formula are shown with correct formatting now.

*Page 6, Line 17- Is Barangay capitalized?*

Okay, is written in capital letters in the revised version.

*Page 6, Line 19- Suggest changing "the two tropical storms/depressions recorded between Haiyan and this field survey on January 19th and February 1st respectively" to "the two tropical storms/depressions hitting the Philippines on January 19th and February 1st, respectively, which is after Haiyan and before this field survey".*

*Okay, was changed accordingly.*

*Page 7, Line 6- Are the values for grain size in ")"thinning and fining landward from 8 cm and a mean of 570 _m at 130 m from the shoreline (HER 8) to only 3 mm and a mean of 223 _m at 260 m (HER 3) (Fig. 3)." for the mode or mean? It appears from Figure 3 that they are for the mode, but I am not sure because this for Unit 1 and it is not clear what is shown in Figure 3.*

The values are indeed for the mode, which is as well shown in figure 3. We clarified this in both text (replace mean by mode) and figure 3 (state the respective units grain size trends are plotted for) in the revised version of the manuscript.

*Page 7, Line 21- Delete "According to".*

Okay, was changed accordingly.

*Page 7, Line 22- Suggest changing "bushes, Haiyan" to "are evidence that Haiyan".*

Okay, was changed accordingly.

*Page 7, Line 30- Suggest changing "Pre and post-typhoon" to "Pre- and post-typhoon".*

Okay, was changed accordingly.

*Page 8, Line 9 and later in the text- The use of the word "profiles" is not standard. Suggest changing "profiles TOL 7-14" to "trenches TOL 7-14". This suggestion applies everywhere in the text where "profile" is used to describe a trench.*

Okay, was changed accordingly.

*Page 8, Line 15- Specify what "slightly inclined" means.*

The laminae are dipping landwards with an angle of ~10-15°. We added this information in the revised version of the manuscript.

*Page 8, Line 21- What is meant by "moderate flooding". I have no idea what is moderate. Please be specific by giving a spatial extent and/or a water depth.*

Since information on flooding was inferred from eyewitness observations, exact (i.e. measured) values for flooding extent and water levels cannot be provided. However, based on the observations, estimations for flooding extent (not more than a few 10s of meters) and water levels (not more than ~3 m above msl) are stated in the revised version of the manuscript.

*Page 8, Line 25- How thin are the sand patches?*

The documented sand deposits do not exceed a maximum thickness of 10 cm; most of them are in the range of 1-3 cm thickness.

*Page 9, Line 1- Delete "single". It is not needed.*

Okay, was changed accordingly.

*Page 9, Line 1- Suggest changing "in either direction" to "crest elevation".*

Okay, was changed accordingly.

*Page 12, Line 30- Suggest changing "show comparably large inland extents exceeding 100 m" to "extend at least 100 m inland".*

Okay, was changed accordingly.

*Page 13, Line 13- Suggest changing "dedicated" to "attributed".*

This section was changed in response to a comment of reviewer 1. In this context the mentioned expression was removed anyway.

*Page 16, Line 21- Suggest changing "confined" to "indicated".*

Okay, was changed accordingly.

*Page 25, Figure 3- Be consistent with line types in each panel. Sorting is a different line type for HER 3-9 than for TOL 3-14 and BAN 1-3. A minor point, but why not make it easier on the reader to compare panels? Also, a solid line is used for both the mode and mean in different panels. Why not use a solid line for the mean and another line type for the mode?*

We agree with the reviewer that a consistent style for all panels would facilitate reading the figure. The line styles for mean, mode and sorting were homogenized in the revised version of the manuscript.

*Page 25, Figure 3- Why does the thickness scale for TOL 3-14 start at -2? This makes it difficult to determine the thickness of the more landward deposits. Why not start the scale at 0 to make it easy to determine the thickness of landward deposits?*

We see the point and adjusted the scale for thickness.

*Page 25, Figure 3- Why is there a vertical dashed line at 40 m in the TOL 3-14 panel? Please explain this line in the caption.*

The vertical line is a drawing artefact that has no important meaning and was deleted.

*Page 25 Figure 3 caption- The mean grain size of HER 3-8 doesn't monotonically fine landward. See earlier comment on mode versus mean and description/definition of landward fining.*

It is true that the fining trend at TOL is restricted to the mode but does not apply to the mean. We adjusted the figure caption to match this observations.

*Page 29, Figure 6- For consistency, add the column that indicates grading by the shaded triangles.*

Okay, is added in the revised version.

*Page 29, Figure 6 caption- The transect is not coast-perpendicular.*

The figure caption was changed to read "Transect illustrating the succession of typhoon deposits in landward direction".

*Page 33, Figure 10 caption- The transect is only shore-perpendicular for trenches 1 and 2, not 3.*

[revised manuscript text omitted]

a) inundation limit

300 m

11/13/2013

N

b) 3.28 3.15 2.90 A limit of sand transport 2.80 14 13 12 11 10 3.42 wall 9 8 7 6 5 4 3 15 3.43 4.70 4.49

30 m pre-Haiyan 11/11/2013

30 m

N

A

c)

d)

e)                                                                    *Tolosa (TOL)*

A' ← deposition →                                    erosion →

4.7   4.5                                              altitude (m a./b. msl)

3.2              3.4                          3.4
2.8              2.8

                                             5  4                erosive scarp
                                         6
                                                    3
       14          13   12      11 10  9  8  7                              post-Haiyan

280        120    *distance to shoreline*   80        40        0

flow direction   x.x water level in m above msl   xx core   xx pit/trench

[revised manuscript text omitted]